# Efficient and accurate search in petabase-scale sequence repositories

Mikhail Karasikov[1,2,3,6], Harun Mustafa[1,2,3,6], Daniel Danciu[1,3], Oleksandr Kulkov[1,3], Marc Zimmermann[1,3], Christopher Barber[1,3], Gunnar Rätsch[1,2,3,4,5 ✉] & André Kahles[1,2,3 ✉]

The amount of biological sequencing data available in public repositories is growing rapidly, forming a critical resource for biomedicine. However, making these data efficiently and accurately full-text searchable remains challenging. Here we build on efficient data structures and algorithms for representing large sequence sets[1–6]. We present MetaGraph, a methodological framework that enables us to scalably index large sets of DNA, RNA or protein sequences using annotated de Bruijn graphs. Integrating data from seven public sources[7–13], we make 18.8 million unique DNA and RNA sequence sets and 210 billion amino acid residues across all clades of life—including viruses, bacteria, fungi, plants, animals and humans—full-text searchable. We demonstrate the feasibility of a cost-effective full-text search in large sequence repositories (67 petabase pairs (Pbp) of raw sequence) at an on-demand cost of around US$100 for small queries up to 1 megabase pairs (Mbp) and down to US$0.74 per queried Mbp for large queries. We show that the highly compressed representation of all public biological sequences could fit on a few consumer hard drives (total cost of around US$2,500), making it cost-effective to use and readily transportable for further analysis. We explore several practical use cases to mine existing archives for interesting associations, demonstrating the use of our indexes for integrative analyses, and illustrating that such capabilities are poised to catalyse advancements in biomedical research.

For more than a decade, innovation in high-throughput DNA sequencing (that is, transforming information stored in DNA into human- and machine-readable sequences) has propelled research in the biomedical domain and led to an exponential growth in worldwide sequencing capacity[14,15]. A large proportion of these data is deposited in publicly funded repositories, such as the European Nucleotide Archive (ENA) maintained by the European Molecular Biology Laboratory's European Bioinformatics Institute (EMBL-EBI)[16], the Sequence Read Archive (SRA) maintained by the National Center for Biotechnology Information (NCBI)[7] and the DDBJ Sequence Read Archive (DRA) maintained by the DNA Data Bank of Japan[17]. Although the rate of data deposition has slowed in recent years, the number of sequenced nucleotides contained in the ENA still doubles roughly every 45 months and currently comprises more than $10.8 \times 10^{16}$ nucleotide bases (108 Pbp)[18] of raw sequencing data, of which $6.7 \times 10^{16}$ nucleotide bases (67 Pbp) are publicly available (as of 11 January 2025; Supplementary Fig. 13).

The classical pattern for accessing sequencing data in such large repositories is to identify relevant records using descriptive metadata and to retrieve a copy or slice of the data for further processing. Downloading the records or accessing them on a cloud platform for analysis often requires considerable resources. For the longest time, these petabase-scale raw sequencing data remained inaccessible for full-text search (that is, the task of retrieving all datasets containing a sequence similar to a given sequence query), substantially limiting its potential for

future research. Consequently, we propose a highly scalable approach to index petabase-size repositories of raw biological sequencing data, transforming them into a highly compressed representation that is portable and yet fully accessible for downstream analysis. Inspired by recent advances in sequence algorithms and data structures[1,2,19–23], we focused on specific technical developments for higher compression, as well as faster exact search and accurate alignment[4,6] to demonstrate that searching public sequence archives as a whole is not only theoretically possible, but practically feasible at high accuracy and affordable costs.

Indexing petabases of biological sequencing data gives rise to a variety of technical challenges, for which some recent solutions have been proposed. Naturally, a focus is on making genetic variation in large cohorts, especially in humans, accessible for research and clinical use. Recent frameworks for variation graphs, such as VG[24], and methods for compressing haplotypes or other graph paths[25] have improved variation-aware alignment and variant calling[23,26]. However, these methods show strong limitations in the scalability of the input size and sequence variability. A second focus of algorithmic work is on taxonomic classification and read annotation, where the task is to match a given query sequence against a (large) set of known reference sequences or taxonomic labels. The classic approach for alignments against a collection of assembled genomes is BLAST[27]. Although the tool is heavily used and has gradually improved in speed[28] over the past 30 years, it still lacks the scalability required for high-throughput

[1]Biomedical Informatics Group, Department of Computer Science, ETH Zurich, Zurich, Switzerland. [2]Swiss Institute of Bioinformatics, Zurich, Switzerland. [3]Medical Informatics, University Hospital Zurich, Zurich, Switzerland. [4]Department of Biology, ETH Zurich, Zurich, Switzerland. [5]AI Center, ETH Zurich, Zurich, Switzerland. [6]These authors contributed equally: Mikhail Karasikov, Harun Mustafa. ✉e-mail: Gunnar.Raetsch@inf.ethz.ch; Andre.Kahles@inf.ethz.ch

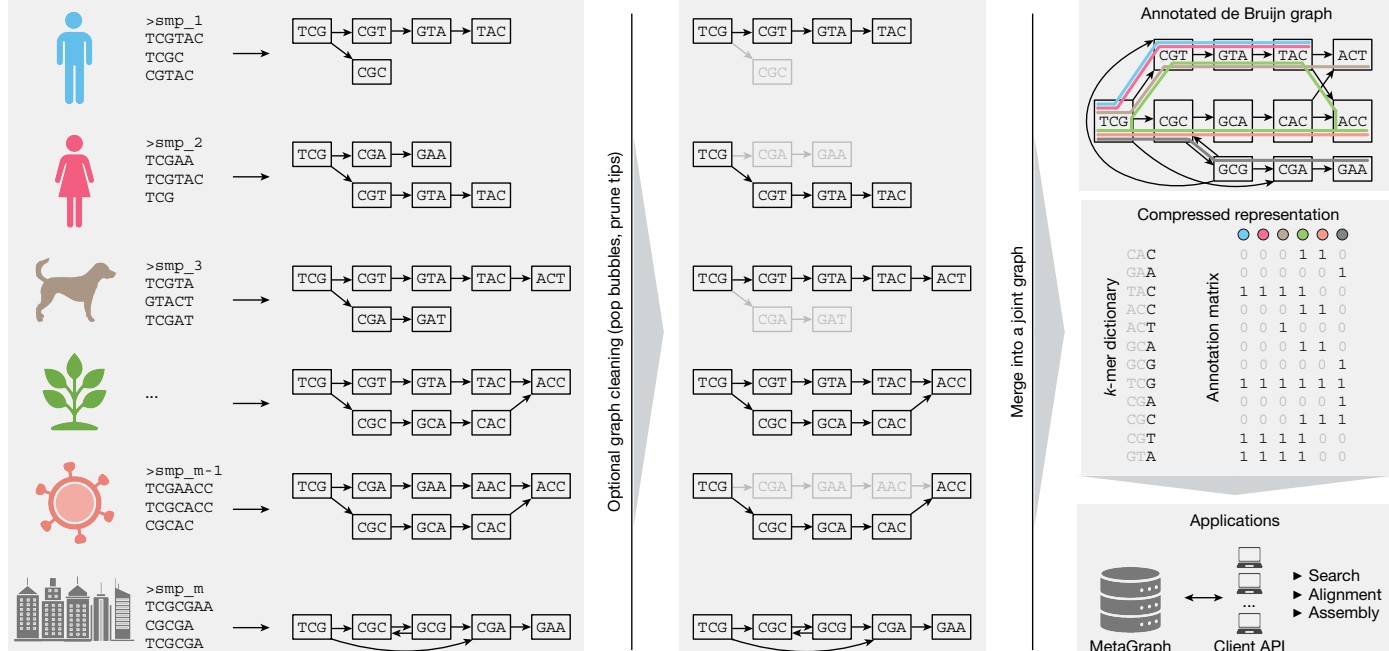

**Fig. 1 | The MetaGraph framework.** Schematic of graph construction and representation. Individual sequencing samples (left) are assembled into graphs (sample graphs), which are then cleaned to remove spurious paths and misassemblies (middle). Next, the sample graphs are merged to form a MetaGraph index (top right), consisting of the compressed *k*-mer dictionary index and a compressed annotation matrix (middle right; the grey colour indicates parts that can be efficiently compressed and do not have to be stored explicitly). The index is then used as the basis for downstream applications, such as sequence search, assembly and other queries (bottom right). Created in BioRender. Kahles, A. (2025) https://BioRender.com/3w8zkl3.

searches in petabase-scale archives. With approaches like Centrifuge[29], promising speed-up has at least been gained for querying large sets of assembled genomes. A third technical focus is on the sequence or experiment discovery problem: querying a sequence of interest (such as a transcript) against all samples available in a sequence repository. Current approaches for experiment discovery can be grouped into three categories: (1) sketching techniques for approximate set similarity and containment queries, which produce small hash-based summaries of the input data and then use these summaries (that is, sketches) to estimate the distances between query sequence sets and a target[30,31]; (2) approximate membership query data structures (such as Bloom filters) for approximately querying individual short sequences against sequence sets[22,32–34]; and (3) methods for the exact representation of annotated de Bruijn graphs, also called coloured de Bruijn graphs, storing additional metadata as annotations of its nodes or edges[19], such as Mantis[20], Fulgor[35,36] and others[22,37,38]. All methods approach experiment discovery by matching short substrings of a fixed length *k*, called *k*-mers, from query sequences against those stored in the index. Although some of them are efficiently scalable, such as the recently presented PebbleScout approach[39], this comes at the cost of lacking support for sensitive alignment and a comparatively high false-positive rate[34,39].

A major challenge faced by all the methods discussed is finding a desirable balance between efficient operation on petabase-scale sequence collections and support for a versatile array of query operations, such as exact *k*-mer lookup and sequence-to-graph alignment for experiment discovery. To bridge this evident gap and to demonstrate the practical feasibility of economical and accurate full-text indexing of biological sequence repositories, we present MetaGraph, a versatile framework for the indexing and analysis of biological sequence libraries at the petabase scale.

## Efficiency, modularity and extensibility

Building on recent advances in the field[1,21] and by developing additional approaches for sequence analysis[3–6,40], we devised the MetaGraph framework, which forms the core of this work and scales from use on single desktop computers up to distributed compute clusters. Notably, MetaGraph can index biological sequences of all kinds, such as raw DNA-sequencing and RNA-sequencing (RNA-seq) reads, assembled genomes and amino acid sequences. A MetaGraph index consists of an annotated sequence graph (Fig. 1 (top right)) that has two main components: the first is a *k*-mer dictionary representing a de Bruijn graph (Fig. 1 (middle right)). The *k*-mers stored in this dictionary serve as elementary tokens in all operations on the MetaGraph index. The second component encodes any metadata as a relationship between *k*-mers and categorical features (called annotation labels) such as sample IDs, geographical locations[8], quantitative or positional information. This relationship is represented as a sparse matrix (annotation matrix; Fig. 1 (middle right)). We use various techniques to represent the de Bruijn graph and the annotation matrix in a highly compressed form[1,3–5] (Methods). MetaGraph supports using different graph and annotation representations interchangeably, adapting to different storage requirements and analysis tasks, and allowing easy adoption of new algorithmic developments. Consequently, we made certain design choices when developing MetaGraph: (1) the use of succinct data structures and efficient representation schemes for extremely high scalability; (2) efficient algorithmic use of succinct data structures (such as preferring batched operations); and (3) modular open-source-licensed architecture supporting several graph and annotation representations and enabling the addition of new algorithms with little code overhead. These strengths allow MetaGraph to easily adapt to the rapid methodological advancements for sequence set representation[35,37,41].

## Scalable multi-sample index construction

Without loss of generality, we will assume that the indexing is applied to read sets that stem from sequencing biological samples, where each annotation label encodes a sample ID. Thus, the annotation matrix encodes the samples to which each *k*-mer belongs. Our indexing workflow proceeds in three steps: (1) data preprocessing; (2) graph

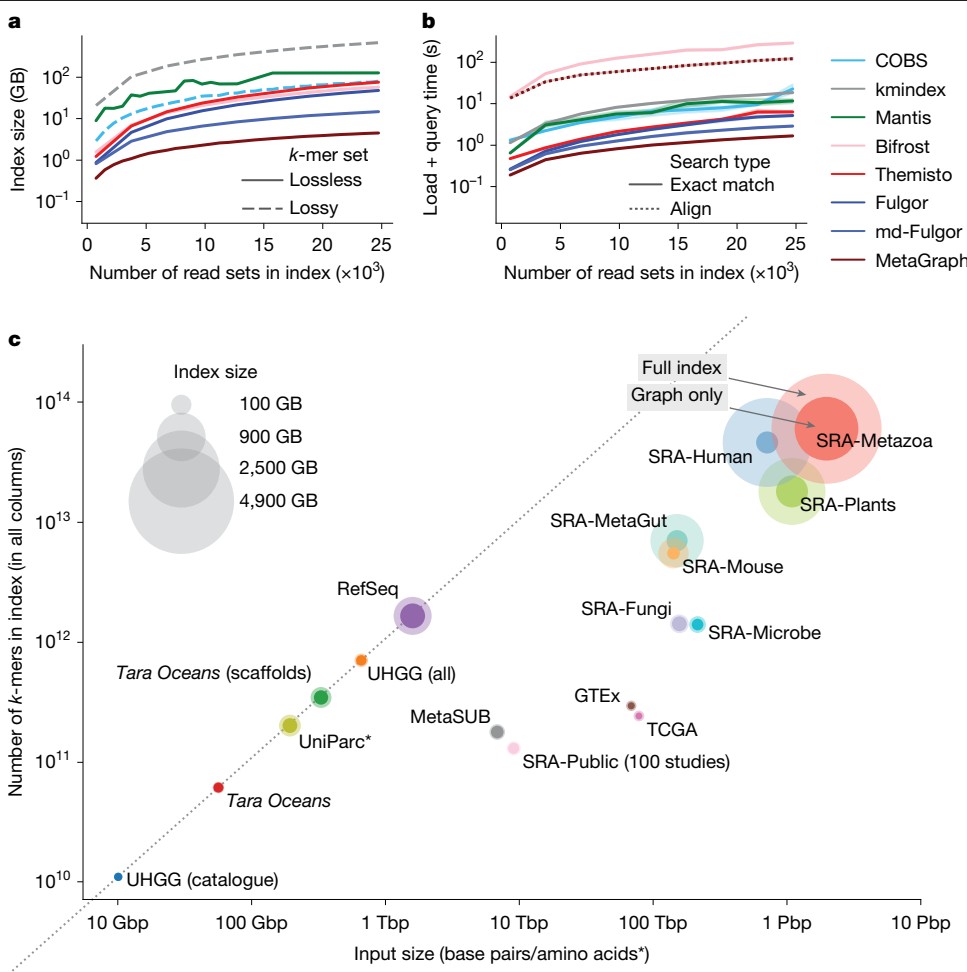

**Fig. 2 | Scalable indexing with MetaGraph. a**, The size of evaluated index data structures for representing a set of microbial whole-genome sequencing (WGS; BIGSI dataset) experiments of increasing size, shown for both lossy indexing methods COBS and kmindex, and for lossless Mantis, Bifrost, Themisto, Fulgor and MetaGraph with the SuccinctDBG and RowDiff<Multi-BRWT> compression schemes to encode the graph and the annotation, respectively. The dashed lines indicate lossy methods. **b**, The times for querying human gut metagenome AMPLICON sequencing reads (SRA: DRR067889) against indexes constructed with MetaGraph and other state-of-the-art tools from sets of microbial WGS experiments of increasing size. All curves show the performance of exact $k$-mer matching, except for the dotted MetaGraph curve, which shows the query performance with the more sensitive search strategy involving alignment. **c**, Overview of all MetaGraph indexes. For all datasets, we show the total number of input characters on the $x$ axis and index size (given as the total number of unique $k$-mers) on the $y$ axis. The marker size represents the size of the index. The solid portion of each marker represents the fraction of the total size taken by the graph and the translucent portion represents the fraction taken by the annotation (Table 1). Asterisk indicates that the inputs of the UniParc dataset are amino acid sequences, not base pairs.

construction; and (3) annotation construction. (1) Data preprocessing involves the construction of separate de Bruijn graphs (sample graphs) from raw input samples (Fig. 1 (left)). We optionally apply a subsequent cleaning step to each sample graph to reduce the impact of potential sequencing errors (Fig. 1 (middle)). Recently, the Logan project provided preprocessed and cleaned data of the Nucleotide archives[13] as a public resource which covers this first step. (2) In graph construction, all of the sample graphs obtained in the first stage are merged into a single joint de Bruijn graph (Fig. 1 (top right)). (3) In annotation construction, we build the columns of the annotation matrix to indicate the membership of different $k$-mers in their respective sample graphs (Fig. 1 (middle right)). Finally, graph and annotation are compressed into representations that are best suited for their target application (Methods).

## First-in-class scalability of MetaGraph

We evaluated MetaGraph's scalability against other state-of-the-art indexing tools[20,33–38] on subsets of increasing size up to 25,000 read sets randomly drawn from the BIGSI collection of microbial genomic read sets[32] (Fig. 2a). The space taken by MetaGraph indexes is 3–150×

smaller than that of the other evaluated approaches, despite some of them[33,34] using a lossy compression approach (while Mantis[20], Bifrost[38], Themisto[37], Fulgor[35,36] and MetaGraph provide lossless representations of the indexed $k$-mer sets, COBS[33] and kmindex[34] both use probabilistic data structures that lead to false-positive matches).

However, the exceptional space efficiency of MetaGraph does not compromise query time. To evaluate that, we queried amplicon reads from a human gut metagenome sample (DDBJ: DRR067889) and found that MetaGraph offers highly competitive query times (Fig. 2b) compared with existing raw sequencing data indexing methods, despite using substantially less space. To achieve this, we devised several efficient algorithms to identify matching paths in the de Bruijn graph with their corresponding annotations and a batch query algorithm (Methods and Extended Data Fig. 1e), exploiting the presence of $k$-mers shared between individual queries by forming a fast intermediate query subgraph, increasing throughput up to 32-fold (Supplementary Fig. 9) for repetitive queries (such as sequencing read sets). In addition to exact $k$-mer matching (Extended Data Fig. 1c), we developed more-sensitive sequence-to-graph alignment[5,6] algorithms that identify the closest matching path in the graph (Methods and Extended Data Fig. 1d).

While exact *k*-mer matching can be viewed as a simplified version of semi-global alignment, it becomes less accurate as the data become more complex, necessitating sequence-to-graph alignment. With sufficient sequencing depth, we find that the more sensitive alignment substantially improves search accuracy (Extended Data Fig. 2).

## Building petabase-scale indexes

To demonstrate the practical feasibility of accurate, cost-effective search across entire sequence archives, we used MetaGraph to index a substantial part of the public NCBI SRA, including DNA and RNA, together with restricted-access human cohorts and UniParc amino acid sequences. Our indexed datasets range from large RNA-seq cohorts (The Cancer Genome Atlas (TCGA)[42], Genotype Tissue Expression (GTEx)[43]) and vast archives of genome sequencing records from the SRA[7] (comprising 2,347,037 of microbial, fungal, plant and metazoa samples) to large, highly diverse whole-metagenome sequencing (WMGS) cohorts (MetaSUB[8]; all public SRA human gut metagenome samples) and collections of reference and assembled sequences (Reference Sequence (RefSeq)[9], Unified Human Gastrointestinal Genome (UHGG)[10], *Tara Oceans*[11]). Our MetaGraph index of the complete UniParc dataset[12] demonstrates the straightforward use of protein sequences as input data. For selected datasets, we generated indexes preserving *k*-mer counts or coordinates using the concept of counting de Bruijn graphs[5]. The key statistics of all datasets presented in this work and the MetaGraph indexes constructed from them are provided in Table 1 and visualized in Fig. 2c. These indexes form a valuable community resource, as they succinctly summarize large sequencing datasets while supporting a variety of sequence queries against them. Indexes of public sequencing data are available at S3 Bucket s3://metagraph. When processing raw sequencing data, we applied moderate input cleaning (Methods), which led to considerable reductions in the required storage space without a substantial effect on index completeness and query sensitivity. In total, we processed 4.8 Pbp of raw input sequences (around 2.5 petabytes (PB) of gzipped FASTQ files), making these data fully and efficiently searchable by sequence.

To estimate the total size of an index for all publicly available sequences, we indexed a random subset of 100 studies from the entire SRA with MetaGraph (9.6 terabase pairs (Tbp)), which resulted in an index of size 32 gigabytes (GB) (Table 1). The entire public sequence repository has 534,254 studies and 66,912.1 Pbp (as of 11 January 2025); the size of a complete index is therefore estimated to be 223.3 terabytes (TB) (Table 1) (note that this would fit on a few consumer hard drives, at a total cost of around US$2,500). We repeated this estimate for an additional set of 47 chunks, each comprising 100 random studies, using preprocessed samples from the set of Logan contigs[13] (Methods), resulting in a cumulative index size of 1.1 TB. Using the same extrapolation rules above, we estimate a complete index size of 172.9 TB. Lastly, we indexed a much larger subset of the Logan contigs (over 16 million accessions), confirming our estimated compression ratios (Table 1).

## Data redundancy drives compression ratio

MetaGraph's compression performance greatly depends on the properties of the indexed sequence sets. As input data often has different formats (such as FASTA, FASTQ, SRA), we measured the final compression ratio in characters per byte (that is, the average number of input characters per byte of the MetaGraph index), making it comparable across different datasets. We decompose the compression ratio into two factors: data redundancy and indexing efficiency, shown in the last two columns of Table 1. Intuitively, redundancy shows the amount of data duplication within a sample, while indexing efficiency reflects data duplication across different samples and increases for sets of related samples.

While the GTEx[43] and TCGA[42] cohorts total over 100 TB of raw compressed RNA-seq data, they show only limited diversity. Our MetaGraph index represents each in around 10 GB space, thereby achieving the highest compression ratios among all indexed datasets (up to 7,416 bp per byte). Even when adding *k*-mer counts, the final compression ratio is around 1,000 bp per byte.

The medium complexity range mainly consists of whole-genome sequencing read sets, showing less redundancy. Notably, our representation of the SRA-Microbe dataset takes only 57 GB, which is 28× smaller than the 1.6 TB BIGSI index[32] and 2.2× smaller than the smallest Fulgor[36] representation (Supplementary Fig. 7).

On the opposite end, we selected the MetaSUB cohort[8], containing 4,220 environmental metagenome samples comprising 7.2 Tbp, and the SRA-MetaGut cohort, containing all human gut metagenome samples available on SRA at this time, comprising around 156 Tbp. These inputs span very diverse organism populations and include many rare sequences, yielding lower compression ratios (140–155 bp per byte), but still compact indexes (46.7 GB for MetaSUB; 1,111 GB for SRA-Meta-Gut). Lastly, collections of assembled genomes and protein sequences exhibit the greatest diversity and only minimal data redundancy, with intersample similarity reflecting evolutionary distance; despite this, MetaGraph compactly indexes them.

## Highly accurate, near-complete indexes

When performing an experiment discovery query with a given sequence, some true labels may not be retrieved due to *k*-mers discarded by our per-sample cleaning workflow. In this light, we evaluated the accuracy of experiment discovery on our SRA-derived MetaGraph indexes. We validated our cleaning and alignment approaches by showing that the individual sample graphs accurately represent their respective input read sets, with around 82% of our query reads realigning back to their respective sample graphs with at least 80% sequence identity (Extended Data Fig. 3a). When mapping these reads to their corresponding annotated graph indexes, we similarly observe that 75–95% of the query reads retrieve their true labels with at least 75% sequence identity (Fig. 3a).

Moreover, we evaluated the robustness of our mapping algorithms to sequence variation using altered versions of the query reads with varying mutation rates. Generally, we observe that the experiment discovery accuracy difference between exact *k*-mer matching and sequence-to-graph alignment increases with increasing sequence variation (Fig. 3b). Especially crucial for functional sequencing data, such as RNA-seq, is MetaGraph's ability to encode per-sample count information. As an example, we provide the expression of transcript SFTPB-207 (ENST00000519937.6) in a sample from the GTEx cohort (Fig. 3c). Even though our representation compresses the original data 1,000-fold, we can closely reproduce the coverage profile generated from aligning the original raw sequences to the reference using STAR. As all count information is encoded per sample, tissue specificity is preserved. For this example, we find the same surfactant transcript mainly expressed in the lung, and partially in the testis, but not elsewhere (Extended Data Fig. 3b), as expected.

## Exploring the gut resistome and phageome

To showcase MetaGraph's use in real-time omics analysis on large-scale datasets, we queried the full CARD antimicrobial resistance (AMR) database[44] and all bacteriophages from RefSeq release 218 (ref. 45) against the 241,384 human gut microbiome samples in our SRA-MetaGut database (Methods). Using classical approaches, this analysis would have required access to hundreds of terabytes of raw sequencing data. With compressed sequence representations, it can be done on a single compute node in about an hour.

We recover strong associations between the *Escherichia* λ phage ev017 and the *Escherichia coli* β-lactamase gene, two *Klebsiella* phages and carbapenem-resistant β-lactamases from *Klebsiella pneumoniae*, and λ ev017 and the RNA antibiotic efflux pumps (Fig. 4a). Moreover, we studied trends in antibiotic resistance over time across various

**Table 1 | Summary of the datasets and constructed indexes**

| Type | Dataset | Class | Tbp | Number of labels | Index size | Compression | Redundancy | Efficiency |
|---|---|---|---|---|---|---|---|---|
| | | | | | | (char. per byte) | (char. per rel.) | (rel. per byte) |
| Assembled sequence | UHGG (catalogue) | D | 0.01 | 4,644 | 3.2 GB | 3.5 | 1.0 | 3.5 |
| | UHGG (all) | D | 0.71 | 286,997 | 27.3 GB | 26.0 | 1.0 | 25.9 |
| | *Tara Oceans* (scaffolds) | D | 0.36 | 318,205,057 | 110.2 GB | 3.2 | 1.0 | 3.1 |
| | *Tara Oceans* with coordinates[a] | D | 0.06 | 34,815 | 14.6 GB | 4.2 | 1.0 | 4.2 |
| | RefSeq with coordinates[a] | D | 1.70 | 85,375 | 508.9 GB | 3.3 | 1.0 | 3.3 |
| | UniParc | P | 0.21 | 543,904,874 | 124.6 GB | 1.7 | 1.0 | 1.6 |
| RNA-seq | GTEx | R | 71.2 | 9,759 | 9.6 GB | 7,416 | 241.2 | 30.7 |
| | GTEx with counts[b] | R | 71.2 | 9,759 | 76.3 GB | 934 | 241.2 | 3.9 |
| | TCGA | R | 81.2 | 11,095 | 11.1 GB | 7,288 | 334.4 | 21.8 |
| | TCGA with counts[b] | R | 81.2 | 11,095 | 81.2 GB | 1,001 | 320.4 | 3.1 |
| WMGS | MetaSUB | D | 7.2 | 4,220 | 46.7 GB | 155 | 40.5 | 3.8 |
| | SRA-MetaGut | D | 155.8 | 241,384 | 1,111.3 GB | 140 | 22.2 | 6.3 |
| SRA subsets | SRA-Microbe | D | 221.1 | 446,506 | 57.1 GB | 3,870 | 157.6 | 24.5 |
| | SRA-Fungi | D | 162.1 | 121,900 | 80.5 GB | 2,013 | 113.9 | 17.7 |
| | SRA-Plants | D | 1,109.2 | 531,714 | 1,844.1 GB | 602 | 61.5 | 9.8 |
| | SRA-Metazoa (human) | D | 725.4 | 436,494 | 3,402.1 GB | 213 | 15.7 | 13.5 |
| | SRA-Metazoa (mouse) | D | 146.6 | 57,938 | 291.6 GB | 503 | 26.6 | 18.9 |
| | SRA-Metazoa (without Human) | D | 1,999.5 | 805,239 | 5,366.5 GB | 373 | 33.3 | 11.2 |
| SRA-Public | 100-study subset | D+R | 9.6 | 5,184 | 32.0 GB | 300 | 73.3 | 4.1 |
| | Extrapolation to all (11 Jan 2025) | D+R | 66,912.1 | 33,337,531 | 223.3 TB[c] | 300[c] | 73.3[c] | 4.1[c] |
| | 4,700-study Logan contig set | D+R | 405.1 | 227,290 | 1.1 TB | 385 | 56.1 | 6.9 |
| | Extrapolation to all (11 Jan 2025) | D+R | 66,912.1 | 33,337,531 | 172.9 TB[d] | 385[d] | 56.1[d] | 6.9[d] |
| | Logan contigs (subset) | D+R | 16,450.4 | 16,764,975 | 42.9 TB | 384 | 62.2 | 6.2 |

Each index is assigned to one of three classes: DNA (D), RNA (R) or protein (P). The compression ratio is measured in characters (char.) per byte (base pairs or amino acids of input sequences per byte of the MetaGraph index), which corresponds to the input/output ratio if 1 character takes 1 byte. The last two columns show the two factors of the compression ratio: redundancy of the data (due to sequencing depth, sequencing errors and genome complexity/repetitiveness) and indexing efficiency, that is, how many (*k*-mer, label) relationships (rel.) are encoded per byte of the MetaGraph index.

[a]Indexes encode *k*-mer coordinates (they are therefore fully lossless).

[b]In addition to the membership of *k*-mers to different labels, indexes also represent their respective counts.

[c]The index sizes of SRA-Public are estimated from a 100-study subset (described in Supplementary Tables 3 and 4).

[d]The index sizes of SRA-Public are estimated from indexing more than 16 million samples pre-processed in the Logan project[13] (extrapolated from the Logan subset).

continents from whole-genome sequencing samples. We find significant strong growth trends over time in resistance against diaminopyrimidines in Africa, antiseptics and fluoroquinolones in Oceania, and against cephamycin and one of the 'last resort' antibiotics tigecycline[46] (a glycylcycline) in South America (Fig. 4b).

Demonstrating MetaGraph's interactive abilities, we provide public example scripts (https://github.com/ratschlab/metagraph_paper_resources/tree/master/notebooks) querying the full CARD database against an index of 4,220 WMGS samples from the MetaSUB cohort. We reproduce a ranking of cities based on the average number of AMR markers per sample (Extended Data Fig. 4b), consistent with the analysis performed on the raw data using orthogonal strategies[8]. We also show other exploratory analyses, linking sample metadata, such as surface material at the sampling location, to the query results (Extended Data Fig. 4c).

## Survey of back-splicing in GTEx and TCGA

MetaGraph indexes can easily express transcriptome features that are difficult to represent in a classical linear coordinate system. One example is back-splice junctions (BSJs) that connect the donor of an exon site to the acceptor site of a preceding exon, forming circular RNAs, which have been found to commonly occur also in humans[47] (Fig. 4d). When systematically querying all 4,052,768 possible intragenic BSJs using the

GENCODE annotation (v.38) and the hg38 reference genome (Methods), we found a total of 1,113 and 2,093 candidates perfectly matching to the GTEx and TCGA indexes, respectively, that did not match the reference genome or transcriptome, and had a coverage of all *k*-mers of at least 10 in at least one sample. While such features are hard to map using linear RNA-seq aligners (Extended Data Fig. 4e), it is a simple task using our graph index. After further filtering (Methods), we found several BSJs to be recurrently detectable in a large fraction of GTEx tissues and TGCA cancer types (Fig. 4c), where testis, pancreas, blood, muscle, skin and brain tissues showed the largest relative fraction in GTEx, and oesophageal carcinoma (ESCA), ovarian serous cystadenocarcinoma (OV), glioblastoma multiforme (GBM) and stomach adenocarcinoma (STAD) the largest relative fractions in TCGA. When overlapping our candidate set to a set of BSJ derived from recent long read data[48] that we also aligned against the GTEx and TCGA indexes, we found large overlaps of 660 and 880 BSJs (Fig. 4d), respectively, which had high tissue specificity (Extended Data Fig. 4f). Subsetting to the COSMIC Cancer Gene Census and comparing the TCGA and GTEx cohorts suggests a differential use of BSJ between cancer and normal tissues (Extended Data Fig. 4f).

## Constructing a community resource

All indexes constructed from public-access data are available in AWS under s3://metagraph and can be downloaded for large-scale local

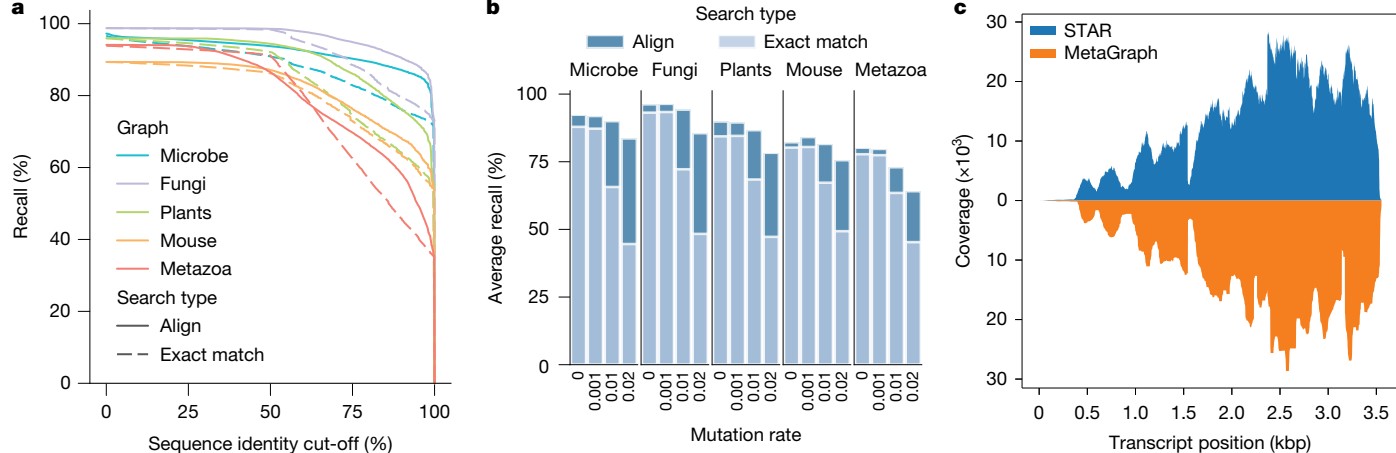

**Fig. 3 | MetaGraph completeness and accuracy. a**, Complementary estimator of the cumulative distribution function (ECDF) curve, showing the fraction of reads that are retrieving the correct label from the index (label recall, *y* axis) at different sequence identity cut-offs (*x* axis) for a range of indexed SRA datasets. When calculating recall at a given sequence identity cut-off, a read mapped to a graph sequence (that is, the spelling of a graph walk) with a given label is only considered if the percentage of the read's nucleotides exactly matching the sequence is above the cut-off. We define the realignability, or average recall, of a graph–query pair as the area under this curve. **b**, Realignability of each graph for increasing error rates in the query (an evaluation of realignability at higher error rates is shown in Supplementary Fig. 5). **c**, Coverage plot for the human transcript SFTPB-207 in GTEx sample SRR599154 determined by STAR alignment against the hg38 reference genome (top, blue) and retrieved from the MetaGraph GTEx index with counts (bottom, orange).

search and analyses or directly accessed from the cloud. Instructions for access are provided at GitHub (https://github.com/ratschlab/metagraph-open-data). To make our indexes accessible for online queries, we developed a search engine for biological sequences—MetaGraph Online (Methods)—which currently hosts a large subset of the various indexes constructed and presented in this work, ranging from SRA samples to several collections of assembled and reference genomes (Supplementary Fig. 12). The service is publicly available for web and API queries (https://metagraph.ethz.ch/search). It uses the MetaGraph framework for hosting the indexes and performing queries on them in real time, showing the result of a query practically immediately. However, the service has limited throughput; thus, for large-scale searches, the indexes provided in the cloud should be used instead.

## The cost and accuracy of global search

To estimate the total cost of exact search and alignment against an entire sequence repository (such as the ENA or SRA), we indexed 47 random subsets of 100 studies each from the Logan contig set[13] using MetaGraph (Methods), leading to performance-optimized (35.6 GB median) and space-optimized (23.2 GB median) indexes. On average, indexing required 0.2 core-hours per sample, which would cost US$0.0025 per sample on Amazon AWS. We created an additional index from raw read sets across 100 random studies sampled from the entire SRA with MetaGraph, using our sample cleaning strategy (Table 1). The construction cost per sample was US$0.028, where more than 90% of the effort went into preprocessing the raw sequencing data (Supplementary Table 4). We then estimated the throughput of *k*-mer matching and alignment for different query sizes (Methods). We assume on-demand querying, where we download each index from cloud storage into local RAM before querying. For each query size, we choose a cost-optimal Amazon AWS cloud compute node, considering the indexes' data transfer and querying times. We estimated costs by querying reads against our random subset indexes, then scaling up to the entire SRA (33,337,531 public accessions, 67 Pbp as of 11 January 2025) by multiplying the costs by 252.6 (Fig. 5a and Methods). This realistically estimates the costs of searching against the whole set of publicly available sequences. Figure 5a shows the scaled query price per kilobase pair (kbp) for different query sizes. For large queries, for which index loading is a minor cost, queries become as cheap as US$0.74 per Mbp for exact *k*-mer matching and US$18.02 per Mbp for

alignment. For small queries, the index loading time dominates, but a single query ranging from 1 to 10 kbp can still be done at a price as low as US$100 even for search with alignment against the entire SRA. We find that these results are robust against changes to the composition of indexed samples. Even querying complex metagenomes incurs the same cost (Supplementary Fig. 17). Next, to assess how our search sensitivity depends on the mutation rate and how it is affected by the cleaning during preprocessing, we searched our 2 Mbp query set (18,889 reads) against the index of 100 random studies from which they originated (Fig. 5b), as well as the same reads with added random mutations at rates of 0.01 and 0.05. We observe a high recall for the original reads without mutations, reaching 96% at a 0.5 sequence identity cut-off for both *k*-mer matching and alignment, and 81% and 85% at a sequence identity cut-off of 0.75 for *k*-mer matching and alignment, respectively. For mutated reads, alignment substantially improves on the exact matching strategy. At a mutation rate of 0.05, it retrieves around 58% instead of 0.5% of the reads at a 0.75 sequence identity cut-off. Finally, despite using a lossless index, even completely random sequences can occur by chance in any read set (with probabilities dropping exponentially with read length) and may accumulate when querying millions of samples. To estimate the expected number of matches that a random query sequence would encounter within the index of all publicly available samples for different search strategies, we generated a set of random sequences and queried them against the same 100-study index, then scaled it to the 33,337,531 public accessions in the SRA to 11 January 2025 (Fig. 5c). We also derived a theoretical model that predicts the number of random accession matches in the entire SRA (Methods).

## Discussion

Here we set out to demonstrate the practical feasibility of indexing entire sequencing archives, such as the EBI's ENA or the NCBI's SRA, with the goal of making them accessible for accurate and cost-efficient full-text search. The major challenge of this task is to hold pace with the constantly growing amount of biological sequencing data. As a solution, we present MetaGraph—a highly scalable and modularized framework designed to index and analyse very large collections of biological sequencing data.

In total, we have processed almost 5 Pbp of public SRA/ENA data and transformed them into compressed indexes that are accessible for

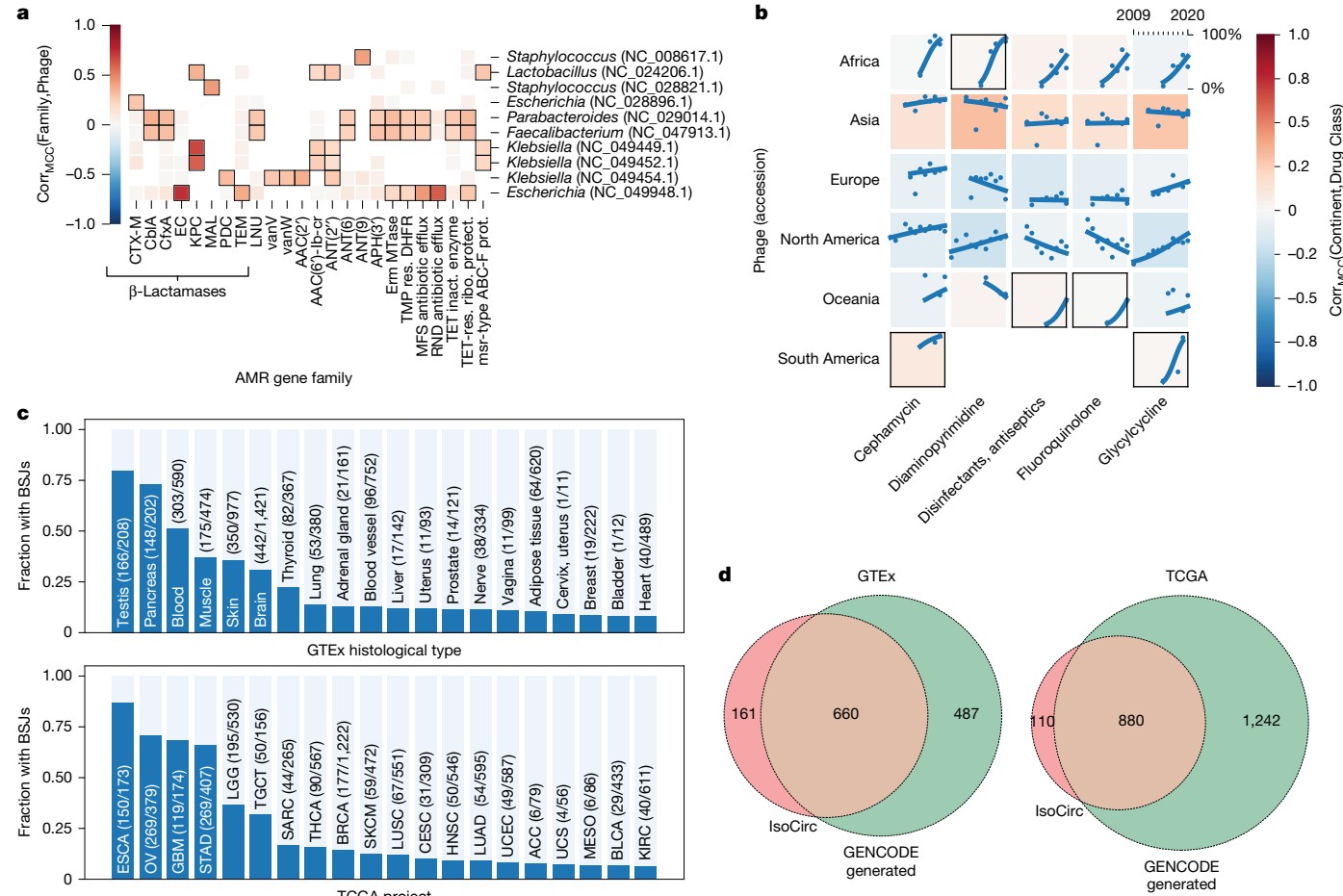

**Fig. 4 | MetaGraph aids biological discovery. a**, Associations between AMR gene families and RefSeq bacteriophages (phages) discovered in publicly available human gut microbiomes deposited to the NCBI SRA from 2008 to 2022. Squares with a black border indicate a statistically significant association and missing squares indicate hypotheses that were not tested due to an insufficient number of matching samples. inact., inactivation; prot., protein; protect., protection; res., resistant; ribo., ribosomal. **b**, The growth in the prevalence of resistance to antibiotics in human gut microbiomes across six continents from 2010 to 2020. Each point represents the normalized proportion of samples from a continent matching genes conferring resistance in a given year, and each line represents a binomial regression fit. Squares with a black

border indicate regressions with significantly high fit scores and missing squares indicate experiments with an insufficient number of samples. Corr$_{MCC}$, Matthews correlation coefficient. **c**, The relative fraction of GTEx (top) and TCGA (bottom) samples per tissue (*x* axis) that carry recurrent BSJs. The per-tissue ratio is computed as samples that carry at least 1 BSJ (dark blue) over all samples of that tissue (light blue)—both on the *y* axis. The absolute numbers are given in parentheses. **d**, The overlap between BSJ predicted from GENCODE and found through MetaGraph with the junction set experimentally provided through IsoCirc for GTEx (left) and TCGA (right) cohorts. Only junctions sufficiently highly expressed in the respective cohort are counted.

full-text search, reducing the input data size up to a factor of around 7,400 for highly redundant sequence sets and a factor of about 300 on average (as in our random 100-study subset). This not only makes the data better accessible, but also makes the data easily transportable across analysis sites. Specifically, we have indexed various collections of DNA and RNA sequences, including a substantial portion of all publicly available whole-genome sequencing samples from the NCBI SRA. In particular, we have indexed a substantial portion of all microbe, fungi, plant, human and human gut wmetagenome and a substantial part of the metazoan samples from the SRA, which together alone make up 2.6 Pbp in 1,903,327 read sets. Moreover, we indexed several other diverse and biologically relevant datasets, from reference genomes to raw metagenomic reads. Finally, we indexed over 16 million public SRA samples that were pre-assembled into contigs within the Logan project[13]. The MetaGraph indexes require orders of magnitude less storage than the original gzip-compressed FASTQ inputs and provide insights into the biological structure and composition of their respective input data (Table 1), while demonstrating consistently high realignability (Fig. 3).

Owing to the sheer size of the data, and because our indexes perform optimally when present in main memory, we have split the input into

batches. This enables us to arbitrarily scale the index representation and enable efficient parallel search, while only marginally reducing compression performance. This partitioning induces a natural growth strategy for a whole database of biological sequences: newly deposited samples are indexed at regular intervals to form index chunks. The already indexed samples can optionally be repartitioned, improving inter-sample similarity and thus further improving compression. On all datasets (except for assembled references), we apply a strategy of moderate cleaning to not accumulate sequencing noise in the final indexes. Although this slightly reduces our query sensitivity (Supplementary Fig. 8), we generally observe a tenfold reduction in index size.

We provide all indexes generated from public data as an open resource to the community that can be accessed within the compute cloud for queries. By additionally providing a web service for interactive queries at MetaGraph Online, we allow life-science researchers and other communities easy and cost-efficient access to the sequencing data for exploration and search. Although this still represents only a fraction of all publicly available sequencing data, our effort was carried out by a single research laboratory, demonstrating general feasibility and serving as a proof of concept.

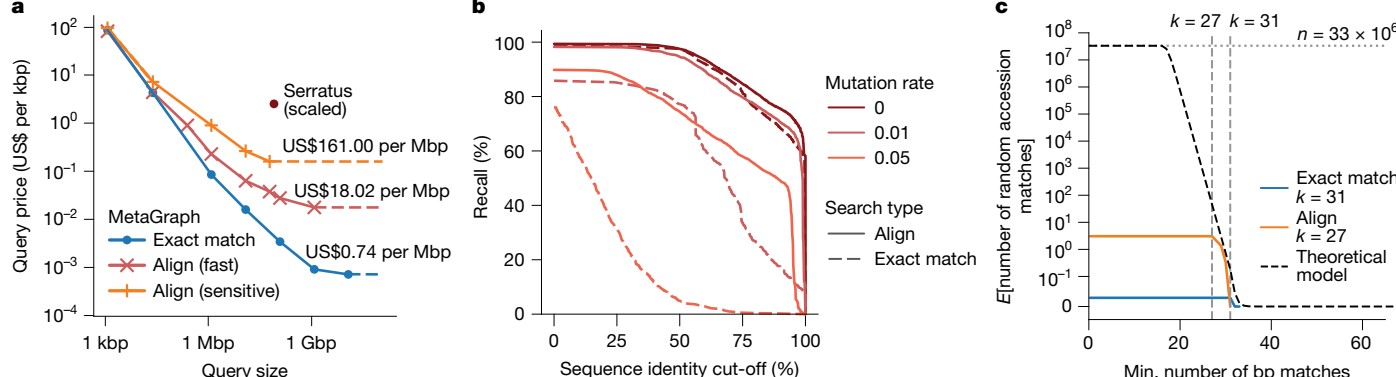

**Fig. 5 | Cost and accuracy of MetaGraph indexes of the SRA. a**, The cloud compute costs for sequence queries of increasing size calculated by mapping query sets with reads of length 100–250 randomly selected from the SRA-Public (100 studies) data against 47 100-random-study indexes of Logan contigs[13], and then scaled up to the whole set of publicly available sequences (33,337,531 public accessions as of 11 January 2025). We use the minimum cost attainable when querying both the performance- and space-optimized index format (the costs of both formats are shown in Supplementary Fig. 16). For comparison, we plotted the estimated cost of the Serratus alignment query[60] if it were performed on the 11 January 2025 SRA snapshot. **b**, Query sequence recall to the 100-study index at different mutation rates for exact matches and alignments. We use the Fast alignment profile for the 0.0 mutation rate and the Sensitive profile for the 0.01 and 0.05 mutation rates. When calculating recall at a given sequence identity cut-off, a read mapped to a graph sequence (that is, the spelling of a graph walk) with a given label is only considered if the percentage of the read's nucleotides exactly matching the sequence is above the cut-off. **c**, The expected numbers of random matches of a sequence of 100–250 bp length to the whole set of public sequences (as of 11 January 2025) reported by different search approaches. The dotted horizontal line represents the total number of public SRA accessions, and the dashed vertical lines represent the $k$-mer size (31) for exact matching and the seed length (27) used for alignment. The theoretical model curve represents the theoretically expected number of matches of a random string of length 130 bp to the SRA ($y$ axis) where at least one $k$-mer of a given length ($x$ axis) is found exactly. A comparison to other tools is shown in Supplementary Fig. 14.

During the completion of this work, the Logan resource[13] became available. In this seminal effort, all read datasets published in the SRA up to December 2023 were transformed into $k$-mer spectra (discarding singleton $k$-mers), reducing their data size 25-fold for contigs and 6-fold for unitigs. The resulting Logan contigs are analogous to MetaGraph's preprocessed sequence sets and can be used directly to build MetaGraph indexes. Recently, Logan Search[49] has become available, presenting a website with online search capabilities for all Logan contigs. A direct comparison of query cost and accuracy is beyond the scope of this work.

Based on a representative random sample taken from SRA, we provide an extrapolated cost to index the entire public archive and show the expected query sensitivity and precision (Fig. 5). These results show that continuous indexing is technically feasible for institutions such as the EBI or the NCBI.

The results presented in this study are an important milestone in computational genomics, demonstrating the feasibility of accurate and cost-effective search in petabase-scale biological sequence repositories and making them more accessible for exploration and computational analyses—a current, pressing problem of high practical relevance. We have demonstrated that it is feasible to accurately and cost-efficiently search for and align nucleotide sequences in all available raw and assembled sequencing data across the tree of life. What was deemed to be very challenging a few years ago, such as indexing and searching in a few thousand read sets, now is tractable and can easily be done on a modern laptop. We envision MetaGraph to serve as a versatile and modular framework that enables researchers to perform large-scale comparative analyses in genomics and medicine using typical academic compute clusters, making public datasets truly open and interactively accessible. We envision that MetaGraph indexes could facilitate large-scale learning tasks on biological sequences, such as training large language models. For example, MetaGraph indexes can act as a database to efficiently generate sequences for model training, including both previously observed and de novo sequence recombinations.

Lastly, MetaGraph has been developed in the context of a highly active research community. Its design as a modular framework enables us to benefit from future technological improvements. Many new exciting approaches for representing $k$-mer sets[35,50], the use of approximate-membership-query data structures[34,51–53], dynamically changing the $k$-mer set[54], improved annotation compression[55,56], as well as alternative alignment and seeding approaches[37,57–59] will be interesting avenues to explore for future extensions of MetaGraph.

## Limitations

The quality of any sequence index can only be as good as the indexed input data. The presence of technical sequence noise, for example, can excessively increase index sizes. We have attempted to remove sequencing noise in a computationally efficient manner, at the cost of removing potentially useful sequence information. Consequently, our search indexes are lossy in the sense that they only represent a user-chosen subset of a sample's $k$-mers losslessly, as opposed to losslessly representing the input sequences or their complete $k$-mer sets. On the other hand, our indexing of assembled sequences (RefSeq[9], UHGG[10] and *Tara Oceans*[11]) is completely lossless, encoding genomic coordinates and avoiding any sequence cleaning[5]. A second limitation of our approach is the use of static data structures for maximal query performance and minimal memory consumption, which makes it hard to directly extend an existing index with additional samples. Although we have shown that this is not an issue for the presented use cases, there are situations in which extending a large index with a few samples is desirable. In such situations, either the index has to be fully reconstructed or the delta needs to be indexed and queried separately from the existing pre-constructed index. Third, all $k$-mer based indexes show limited sensitivity when querying noisy (such as nanopore sequencing) or distantly homologous sequences. Lastly, MetaGraph has been optimized for DNA, RNA and amino acid alphabets. Many recent sequencing technologies can also record sequence modifications, resulting in much larger alphabet sizes. Although MetaGraph can handle such inputs in principle, current search algorithms are not optimized for this setting.

## Conclusions

We are confident that the approaches presented here can be used and integrated into the infrastructure of large data repositories, such as

ENA and SRA, to make all sequencing data stored in these repositories searchable, thereby providing a small, yet efficient prototype of a 'Google for DNA'. Notably, the cost of providing this whole service would be relatively small compared with the price paid to generate these data in the first place and the cost of storing it in the ENA and SRA.

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

# Methods

## Representing de Bruijn graphs

According to the formal definition of a de Bruijn graph, every pair of $k$-mers (short contiguous subsequences of length $k$) with an overlap of $k-1$ characters are connected with a directed edge. It can be seen that in the de Bruijn graph, the edges can always be derived from the nodes alone, and hence, it is sufficient to store only the $k$-mer set. These $k$-mers are used as elementary indexing tokens in MetaGraph.

MetaGraph provides several data structures for storing $k$-mer sets, which are used as a basis to implement different representations of the de Bruijn graph abstraction. In addition to the simple hash table, the $k$-mers may be stored in an indicator bitmap[61] (a binary vector represented as a succinct bitmap of size $|\varSigma|^k$ indicating which $k$-mers are present in the set) or in the BOSS table[1] (a data structure proposed by Bowe, Onodera, Sadakane and Shibuya for storing a set of $k$-mers succinctly; see Supplementary Fig. 3 and the corresponding Supplementary Table 2 for an example BOSS graph and table). We call our de Bruijn graph implementations based on these data structures HashDBG, BitmapDBG and SuccinctDBG, respectively. For detailed descriptions and the properties of these representations, refer to Supplementary Section A.3.1 and Supplementary Table 1. All these data structures support exact membership queries, and they map $k$-mers to positive indexes from 1 to $n$, where $n$ is the number of $k$-mers in the represented set (or zero if the queried $k$-mer does not belong to the set). While HashDBG is mostly used internally (for example, for batched sequence search), SuccinctDBG typically exhibits the best compression performance and, therefore, serves as the default compressed representation.

## Scalable $k$-mer enumeration and counting

To count and de-duplicate $k$-mers, MetaGraph uses the following approach to generate the so-called $k$-mer spectrum: the input $k$-mers are appended to a list, which is sorted and de-duplicated every time it reaches the allocated space limit or after all input $k$-mers have been processed. During de-duplication, the $k$-mer counts are summed up to maintain the total count of each unique $k$-mer. We call this approach SortedSet. Moreover, MetaGraph offers the SortedSetDisk approach, which implements a similar algorithm in external memory. Using a pre-allocated fixed-size buffer limits memory usage and allows for constructing virtually arbitrarily large $k$-mer spectra but requires a larger amount of disk I/O. Lastly, MetaGraph supports passing precomputed outputs from the KMC3 (ref. 62) $k$-mer counting tool (fork karasikov/ KMC commit b163688) as an input to make use of its exceptionally efficient counting algorithm and filters. Once the entire $k$-mer spectrum is obtained and all $k$-mers are sorted, they are converted into the final data structure to construct the target graph representation.

## Extracting contigs and unitigs from graphs

All sequences encoded in the graph (or any defined subgraph) can be extracted from it and stored in FASTA format through graph traversal[63,64]. The graph is fully traversed and its paths, formed by consecutive overlapping $k$-mers, are converted into sequences (contigs) that are returned as a result of this operation. Each $k$-mer of the graph (or subgraph) appears in the assembled contigs exactly once. Thus, the resulting set of sequences is a disjoint node cover of the traversed graph.

MetaGraph provides efficient parallel algorithms for sequence extraction and distinguishes two main types of traversal: (1) traversal in contig mode extends a traversed path until no further outgoing edge is present or if all the next outgoing edges have already been traversed; while (2) traversal in unitig mode only extends a path if its last node has a single outgoing edge, and this edge is the single edge incoming to its target node. This definition of a unitig matches the one described previously[65].

## Basic, canonical and primary graphs

When indexing raw reads sequenced from unknown strands, we supplement each sequence with its reverse complement, which is then indexed along with the original sequence. As a result, the de Bruijn graph accumulates each $k$-mer in both orientations. Such graphs (which we call canonical) can be represented by storing only one orientation of each $k$-mer and simulating the full canonical graph on-the-fly (for example, for querying outgoing edges, return not only the edges outgoing from the source $k$-mer, but also all edges incoming to its reverse complement $k$-mer).

Storing only canonical $k$-mers (that is, the lexicographically smallest of the $k$-mer and its reverse complement) effectively reduces the size of the graph by up to two times. However, this cannot be efficiently used with the succinct graph representation based on the BOSS table. The BOSS table, by design, requires that each $k$-mer in it has other $k$-mers overlapping its prefix and suffix of length $k-1$ (at least one incoming and one outgoing edge in the de Bruijn graph). However, it is often the case that among two consecutive $k$-mers in a read, only one of them is canonical. Thus, storing only canonical $k$-mers in the BOSS table would often require adding several extra dummy $k$-mers for each real $k$-mer (Supplementary Fig. 4), which makes this approach memory inefficient. We overcome this issue by constructing primary graphs, where the word primary reflects the traversal order, as described in the next paragraph.

When traversing a canonical de Bruijn graph, we can additionally apply the constraint that only one of the orientations of a given $k$-mer is called. More precisely, the traversal algorithm works as usual, but never visits a $k$-mer if its reverse complement has already been visited. Whichever orientation of the forward or reverse complement $k$-mer is visited first is considered to be the primary $k$-mer of the pair (an example illustration is shown in Supplementary Fig. 6). This results in a set of sequences, which we call primary (primary contigs or primary unitigs). Note that the traversal order of the graph may change the set of primary sequences extracted from it, but it may never change the total number of $k$-mers in these sequences (primary $k$-mers). This is relevant when extracting primary contigs with multiple threads since the node traversal order may differ between runs. We call graphs constructed from primary sequences primary graphs. In contrast to the common approach in which only canonical $k$-mers are stored, primary de Bruijn graphs can be efficiently represented succinctly using the BOSS table, and effectively enable us to reduce the size of the graph part of the MetaGraph index by up to two times.

## Graph cleaning

When a graph is constructed from raw sequencing data, it might contain a considerable number of $k$-mers resulting from sequencing errors (erroneous $k$-mers). These $k$-mers do not occur in the biological sequences and make up spurious paths in the graph, which one may desire to prune off. True-signal $k$-mers may also originate from contaminant organisms in the biological sample. Pruning the graph to discard either or both of these classes of undesirable $k$-mers is called graph cleaning.

MetaGraph provides routines for graph cleaning and $k$-mer filtering based on the assumption that $k$-mers with relatively low abundance (low $k$-mer counts) in the input data were probably generated due to sequencing errors or contamination and should therefore be dropped. To identify potentially undesirable $k$-mers, we use an algorithm proposed previously[65]. In MetaGraph, we adapted and scaled up this algorithm to work not only for small but also for very large graphs (up to trillions of nodes).

In brief, the decision to filter out a $k$-mer is based on the median abundance of the unitig to which this $k$-mer belongs. That is, $k$-mers with low abundance are preserved if they are situated in a unitig with sufficiently many (more precisely, at least 50%) highly abundant (solid) $k$-mers.

Then the entire unitig is considered solid and is kept in the graph. All solid unitigs (which may also be concatenated into contigs called clean contigs) are extracted from the graph and output in FASTA format. Connected unitigs (those with non-zero degree) that are discarded due to lack of abundance typically originate from sequencing errors, and their removal is traditionally called bubble popping[66]. Optionally, all tips (that is, unitigs where the last node has no outgoing edges) that are shorter than a given cut-off (typically $2k$) are discarded as well. Afterwards, a new graph can be constructed from these clean contigs, which we call a cleaned graph.

The abundance threshold for solid unitigs can be set either manually or computed automatically from the full $k$-mer spectrum. It is assumed that $k$-mers with an abundance of at most 3 are likely to be generated by sequencing errors and that all erroneous $k$-mers follow a negative binomial distribution. After fitting a negative binomial distribution to these low-abundance $k$-mers, the abundance threshold is set to the 99.9th percentile of this distribution[65]. Finally, in case the chosen threshold leads to preserving less than 20% of the total coverage, the automatic estimation procedure is deemed unsuccessful and a pre-defined value (typically 2) is used as a fallback threshold instead.

## Constructing a joint graph from multiple samples

According to our workflow, when indexing multiple read sets (especially when indexing vast collections of raw sequencing data), the recommended workflow for constructing a joint de Bruijn graph from the input samples consists of the following three steps. First, we independently construct a de Bruijn graph from each input read set. As each graph is constructed from a single read set (or sample), we call these graphs sample graphs. If desired, these sample graphs are independently cleaned with the graph cleaning procedure described above. Then, each sample graph is decomposed into a set of (clean) contigs, either by extracting the contigs directly or as a result of the graph cleaning procedure. Finally, a new de Bruijn graph is constructed from all these contigs, which is then annotated to represent the relation between the $k$-mers and the input samples. As this graph represents the result of merging all sample graphs, we refer to it as the joint de Bruijn graph. In practice, the size of the contigs extracted from sample graphs is up to 100 times smaller than the raw input, which makes the construction of the joint de Bruijn graph by this workflow much more efficient compared with constructing it directly from the original raw read sets.

## Graph annotations

Once a de Bruijn graph is constructed, it can already be used to answer $k$-mer membership queries, that is, to check whether a certain $k$-mer belongs to the graph or not. However, the de Bruijn graph alone can encode no additional metadata (such as sample ID, organism, chromosome number, expression level or geographical location). Thus, we supplement the de Bruijn graph with another data structure called an annotation matrix. Each column of the annotation matrix $A \in \{0,1\}^{n \times m}$, where $n$ is the number of $k$-mers in the graph and $m$ is the number of annotation labels, is a bit vector indicating which $k$-mers possess a particular property:

$$A_j^i := \begin{cases} 1 & k\text{-mer } i \text{ is in relation with attribute } j, \\ 0 & \text{otherwise} \end{cases}$$

Without loss of generality, we will assume that the annotation matrix encodes the membership of $k$-mers to different samples, that is, encodes sample IDs. In this case, $A_j^i = 1$ indicates that $k$-mer $i$ appears in sample $j$ (note that the same $k$-mer may also appear in multiple samples).

The number of rows of the annotation matrix corresponds to the number of $k$-mers indexed in the de Bruijn graph, and the annotation matrix can therefore be of enormous size, containing up to $10^{12}$ rows and $10^9$ columns. However, this matrix is typically extremely sparse and can therefore be efficiently compressed.

## Representing graph annotations in MetaGraph

Independent of the choice of graph representation, a variety of methods are provided in MetaGraph for compressing annotation matrices to accommodate different query types. These different matrix representation schemes can be split into row-major and column-major. The row-major representations (such as RowSparse[4], RowFlat[67,68]) enable fast row queries but have a poor performance of column queries. By contrast, the column-major representations (such as ColumnCompressed, Multi-BRWT[3]) provide fast access to individual columns of the annotation matrix but typically have a poorer row query performance. Despite being mathematically equivalent, up to the transposition of the represented matrix, these schemes are in fact algorithmically different due to the number of rows of the annotation matrix typically being orders of magnitude larger than the number of columns. A detailed description of these and other representation schemes is provided in Supplementary Information A.4. The size and query time benchmarks, respectively, of the compression methods used in our representations are provided in Supplementary Figs. 1 and 2.

## The RowDiff compression technique

Due to the nature of de Bruijn graphs and the fact that adjacent nodes ($k$-mers) usually originate from the same sequences, it turns out that, in practice, adjacent nodes in the graph are likely to carry identical or similar annotations. The RowDiff compression technique[4,5] exploits this regularity by replacing the annotations at nodes with their relative differences. This enables us to substantially sparsify the annotation matrix and, therefore, considerably improve its compressibility. Notably, the transformed annotation matrix can still be represented with any other available scheme, including those described above. MetaGraph provides a scalable implementation of this technique with efficient construction algorithms, which allow applying it to virtually arbitrarily large annotation matrices. The algorithm essentially consists of two parts. First, for each node with at least one outgoing edge, it picks one of the edges and marks its target node as a successor. Second, it replaces the original annotations at nodes with their differences from the annotations at their assigned successor nodes. This delta-like transform is applied to all nodes in the graph except for a small subset of them (called anchors), which keep their original annotation unchanged and serve to end every path composed of successors and break the recursion when reconstructing the original annotations (for example, in cycles).

## Counting de Bruijn graphs

Finally, MetaGraph supports generalized graph annotations for representing quantitative information such as $k$-mer positions and their abundances in input sources, encoded with non-binary matrices[5].

## Annotation construction

The typical workflow for constructing an annotation matrix for a large input set consists of the following steps. After the joint de Bruijn graph has been constructed from the input sequences, we iterate over the different samples (corresponding to the different annotation labels) in parallel and map all $k$-mers of each sample to the joint graph, generating a single annotation column. To avoid the mapping of identical $k$-mers multiple times and to prevent the processing of erroneous $k$-mers ($k$-mer with sequencing errors), we use the unitigs extracted from the cleaned sample graphs instead of the raw sequences when annotating the graph. This substantially reduces the annotation construction time, especially when the joint graph is represented with SuccinctDBG, for which the traversal to an adjacent $k$-mer is several times faster than a $k$-mer lookup performed from scratch (Supplementary Information A.3.2).

Once an annotation matrix has been constructed (typically in the ColumnCompressed representation), it can be transformed to any other representation to achieve the desired trade-offs between the representation size and the performance of the required operations. In particular, for sequence search, the recommended workflow is to apply the RowDiff transform on the annotation matrix and then convert the sparsified columns to the Multi-BRWT or RowSparse representation, depending on the desired speed versus memory trade-off.

## Dynamic index augmentation and batch updates

Generally, there are three strategies for extending a fully constructed MetaGraph index (a joint de Bruijn graph and its corresponding annotation matrix). First, the batch of new sequences can be indexed separately and that second MetaGraph index can be hosted on the same or on a different server. Then, these two indexes can be queried simultaneously, as it is done for a distributed MetaGraph index.

Second, the graph can be updated directly if it is represented using a dynamic data structure that supports dynamic updates (for example, SuccinctDBG (dynamic)). Then, the annotation matrix needs to be updated accordingly. This approach allows making instant changes. However, it does not enable large updates because of the limited performance of dynamic data structures[69].

Finally, for large updates, the existing index can be reconstructed entirely. For the reconstruction, the index is first decomposed into contig buckets, where each bucket stores contigs extracted from the subgraph induced by the respective annotation column. Then, these buckets are augmented with the new data (either by adding the new sequences directly or by pre-constructing sample graphs from the new sequences and adding contigs extracted from them), and a new MetaGraph index is constructed from these augmented buckets. Notably, this approach uses a non-redundant set of contigs and does not require processing raw data from scratch again. Furthermore, instead of extracting contigs from the old index, it is also possible to use the inputs initially used to construct the old index (for example, the contigs extracted from sample graphs), which can substantially simplify the process.

## Speeding up $k$-mer matching

For higher $k$-mer matching throughput, we implemented several techniques to speed up this procedure. First, when mapping $k$-mers to a primary graph (defined above), each $k$-mer may generally have to be searched twice (first that $k$-mer and then its reverse complement). Nevertheless, if a $k$-mer has been found, there is no need to search for its reverse complement. In fact, it is guaranteed in that case that the reverse complement $k$-mer would be missing in the graph. However, if a certain $k$-mer from the query is missing in the graph but its reverse complement is found, it is likely that, for the next $k$-mer from the query sequence, which is adjacent to the current one, the same applies. Thus, in such cases, we directly start the search for the next $k$-mer by querying the graph with the reverse complement and checking for the original $k$-mer only if that reverse complement $k$-mer is not found in the graph.

When the graph is represented with the BOSS table, indexing $k$-mer ranges in the BOSS table (as described in Supplementary Information A.3.2) greatly speeds up $k$-mer lookups, especially relevant when querying short sequences or arbitrary sequences against a primary graph.

Another optimization consists of querying the annotation matrix in batches, which improves cache locality and removes possible row duplications. To go further and speed up $k$-mer mapping as well, we developed the batch query algorithm described below in detail.

## Batched sequence search

To increase the throughput of sequence search for large queries (for example, sets of sequencing reads or long sequences), we have designed an additional batch query algorithm schematically shown in Extended Data Fig. 1e. The algorithm exploits possible query set redundancy: the presence of $k$-mers shared between individual queries. More precisely, query sequences are processed in batches and an intermediate batch graph is constructed from each batch. This batch graph is then effectively intersected with the large joint graph from the MetaGraph index. The result of this intersection operation forms a relatively small subgraph of the joint graph, which we call a query graph. It is represented in a fast-to-query uncompressed format (Hash-DBG). In practice, this intersection is performed as follows. First, the batch graph is traversed (step 2 in Extended Data Fig. 1e) to extract a non-redundant set of contigs that are afterwards mapped against the joint graph through exact $k$-mer matching (step 3) and the respective annotations are extracted from the compressed index accordingly to construct the query graph with its respective annotations representing the intersection of the batch graph with the full MetaGraph index (step 4). All of the query sequences from the current batch are then queried against this query graph (step 5). Depending on the structure of the query data, this algorithm achieves a 10- to 100-fold speedup compared to unbatched queries.

## Sequence search with alignment

For cases in which the sensitivity of sequence search through exact $k$-mer matching is insufficient, we developed several approaches for aligning sequences to the MetaGraph index, a process known as sequence-to-graph alignment[5,6,23,70–76]. Note that each approach has its target use cases and the choice should be made based on the particular application and the problem setting.

Each alignment algorithm takes a classical seed-and-extend approach[27,71–74]. Given an input sequence, the seeds are composed by joining consecutive $k$-mer matches within the graph's unitigs (called unitig maximal exact matches, or Uni-MEMs[72,77]). Although, by default, this restricts the seeds to be at least of length $k$, representing the graph with the BOSS table allows for relaxing this restriction by mapping arbitrarily short sequences to suffixes of the $k$-mers indexed in the graph (as described in Supplementary Information A.3.2).

Each seed is extended in the graph forwards and backwards to produce a complete local alignment. Similarly to how GraphAligner[78,79] builds on Myers' algorithm[80], our extension algorithm is a generalization of the Smith–Waterman–Gotoh local alignment algorithm with affine gap penalties[81]. The user can choose to report multiple alignments for each query, which may be found if seeds to multiple locations in the graph are discovered.

We now describe the extension algorithm in more detail. Given a seed, let $s = s_1 \cdots s_{k'}$ denote the suffix of the query sequence starting from the first character of the seed. We use a dynamic programming table to represent the scores of the best partial alignments. More precisely, each node $v$ has three corresponding integer score vectors $\mathbf{S}_v$, $\mathbf{E}_v$, and $\mathbf{F}_v$ of size equal to the query length $\ell$. $\mathbf{S}_v[i]$ stores the best alignment score of the prefix $s_1 \cdots s_i$ ending at node $v$. $\mathbf{E}_v[i]$ and $\mathbf{F}_v[i]$ represent the best alignment scores of $s_1 \cdots s_i$ ending with an insertion and deletion at node $v$, respectively.

Let $v_S$ be the first node of the seed $S$. We define an alignment tree $T_S = (V_S, E_S)$ rooted at $v_S$ encoding all walks traversed during the search starting from $v_S$, where $V_S \subset V \times \mathbb{N}$ contains all the nodes of the paths originating at $v_S$ and $E_S \subset V_S \times V_S$ contains all the edges within these paths. $T_S$ is constructed on-the-fly during the seed extension process by extending it with new nodes and edges after each graph traversal step.

As the size of $T_S$ can grow exponentially if all paths are explored (and is, in fact, of infinite size if the graph is cyclic), we traverse the graph and update $T_S$ in a score-guided manner. For this, we maintain a priority queue graph nodes and corresponding score vectors to be traversed, prioritizing nodes whose traversal led to the best local score update[78]. We use several heuristics to restrict the alignment search space. First, we use the $X$-drop criterion[82,83], skipping an element if it is more than $X$ units lower than the current best-computed alignment score.

Moreover, we maintain an aggregated score column for each graph node storing the element-wise maximum score achieved among the score columns of each node in $T_s$. Using this, we discard nodes in $T_s$ from further consideration if their traversal did not update the aggregate score column. Finally, we apply a restriction on the total number of nodes which can be explored as a constant factor of $\ell$.

To find seeds of length $k' < k$ (by default, we use a seed length of 19) matching the suffixes of nodes in the canonical graph, a three-step approach is taken. First, seeds corresponding to the forward orientation of the query are found, which correspond to contiguous node ranges in the BOSS representation of the graph (a description of the node range matching algorithm is provided in Supplementary Information A.3.2). The next two steps then retrieve suffix matches which are represented in their reverse complement form in the graph. In the second step, the reverse complements of the query $k$-mers are searched to find node ranges corresponding to suffix matches of length $k'$. Finally, these ranges are traversed forwards $k - k'$ steps in the graph to make the prefixes of these nodes correspond to the sequence matched. The reverse complements of these nodes are then returned as the remaining suffix matches.

While primary graphs act as an efficient representation of canonical de Bruijn graphs, special considerations need to be made when aligning to these graphs to ensure that all paths that are present in the corresponding canonical graph are still reachable. For this, we introduce a further extension of the alignment algorithm to allow for alignment to an implicit canonical graph while only keeping a primary graph in memory. During seed extension, the children of a given node are determined simply by finding the children of that node in the primary graph, along with the parents of its reverse complement node. Finding exact matching seeds of length $k' \geq k$ can be achieved in a similar manner, searching for both the forward and reverse complement of each $k$-mer in the primary graph.

MetaGraph maintains three different alignment approaches that determine how the graph is traversed during seed extension, called MetaGraph-Align, SCA and TCG-Aligner, each applying different restrictions to traversal.

**MetaGraph-Align.** In this approach, the sequences are aligned against the joint de Bruijn graph to compute their respective closest walks in the graph. After computing a set of alignments, they are used in place of the original sequences to fetch their corresponding annotations. This approach allows for aligning to paths representing recombinations of sequences across annotation labels.

**Label-consistent graph alignment.** When label recombination is not desired, we support an alternative approach in which queries are aligned to subgraphs of the joint graph induced by single annotation labels (columns of the annotation matrix). We call this approach label-consistent graph alignment (or alignment to columns), and it is implemented by the SCA algorithm[6]. However, instead of aligning to all the subgraphs independently, we perform the alignment with a single search procedure while keeping track of the annotations corresponding to the alignments.

**Trace-consistent graph alignment (TCG-Aligner).** Finally, when input sequences are losslessly encoded in a MetaGraph index using the methodology introduced and evaluated previously[5], the alignment can be done against those original input sequences of which the respective walks in the graph are called traces. This method is called the trace-consistent graph aligner (TCG-Aligner)[5].

### Column transformations
In addition to the operations mentioned above, MetaGraph supports operations aggregating multiple annotation columns to compute statistics for the $k$-mers and their counts (abundances). In general, the following formula is used in the aggregation to compute the $i$th bit of the new annotation column:

$$a_{\min} \leq \sum_{j=1}^{m} 1[v_{\min} \leq c_{ij} \leq v_{\max}] \leq a_{\max},$$

where $c_{ij}$ is the count (abundance) of the $i$th $k$-mer in the $j$th label, and $1[A]$ is a Boolean predicate function that evaluates as 1 if the statement $A$ is true and as 0 otherwise. If no counts are associated with the column, we assume that $c_{ij} = 1$ for every set bit in the $j$th annotation column and 0 otherwise. If the sum $\sum_{j=1}^{m} 1[v_{\min} \leq c_{ij} \leq v_{\max}]$ falls within specified minimum and maximum abundance thresholds $a_{\min}$ and $a_{\max}$, the bit in the aggregated column for this $k$-mer is set to 1, and the value of the sum is written as the count associated with that bit. In other words, the resulting aggregated column is always supplemented with a count vector representing the number of original annotation columns with $k$-mer counts between $v_{\min}$ and $v_{\max}$, which can be used in downstream analyses as an ordinary count vector.

### Seamless distribution and interactive use
In addition to the single-machine use case, where the index is constructed and queried locally, MetaGraph also supports querying indexes provided on a remote server through a client-server architecture. In this approach, a set of graphs and annotations can easily be distributed across multiple machines. Each machine runs MetaGraph in server mode, hosting one or multiple indexes and awaiting queries on a pre-defined port (Extended Data Fig. 1b). This setup makes it straightforward to execute user queries across all indexes hosted on multiple servers. For easy integration of results and coordination of different MetaGraph instances, we provide client interfaces in Python (Extended Data Fig. 1a). Notably, our distribution approach can be used not only for hosting multiple indexes of distinct sources but also when indexing a single dataset of extremely large size, such as SRA. This distribution approach enables virtually unlimited scalability.

### MetaGraph API and Python client
For querying large graph indexes interactively, MetaGraph offers an API that allows clients to send requests to a single or multiple MetaGraph servers. When started in server mode, the MetaGraph index will be persistently present in server memory, which will accept HTTP requests on a pre-defined port. To make the querying more convenient, we have also implemented a Python API client as a Python package available at GitHub (https://github.com/ratschlab/metagraph/tree/master/metagraph/api/python).

### MetaGraph Online
The search engine MetaGraph Online has a clean and intuitive graphical web user interface (UI; Supplementary Fig. 11), enabling the user to paste an arbitrary sequence and search it against a selected index. Restrictions to search multiple sequences at once are only in place to limit hosting costs. By default, the search is performed through basic $k$-mer matching. For greater sensitivity, it is also possible for all indexes to additionally align the searched sequence to the annotated graph. If $k$-mer coordinates or counts are represented in the queried index, the web UI allows retrieving them for the query sequence as well. In addition to user interaction with the web interface, MetaGraph Online provides a web API that allows connecting to the respective servers via their endpoints. That is, any of the hosted indexes can be queried through Python API by connecting to the respective endpoint of the server hosting that index (Extended Data Fig. 1a).

### Indexing public read sets from the NCBI SRA
We have split the set of all read sets from SRA (excluding sequencing technologies with high read error rates, see more details in the sections below) into different groups of related samples and constructed

a separate MetaGraph index for each group. The groups were defined either using dataset definitions of previous work[32] or using the metadata provided by NCBI SRA. As a result, we constructed the following six datasets: SRA-MetaGut, SRA-Microbe, SRA-Fungi, SRA-Plants, SRA-Human and SRA-Metazoa. All of these datasets are listed in the supplementary tables available at GitHub (https://github.com/ratschlab/metagraph_paper_resources) and make up a total of 4.4 Pbp and 2.3 PB of gzip-compressed input sequences, while the indexes make up only 11.6 TB, which corresponds to the overall compression ratio of 193×, or 376 bp per byte.

For constructing the SRA-Microbe index, we used cleaned contigs downloaded from the European Bioinformatics Institute FTP file server provided as supplementary data to BIGSI[32]. Thus, no additional data preprocessing was needed for this dataset.

For all other datasets, each sample was either transferred and decompressed from NCBI's mirror on the Google Cloud Platform or, if not available on Google Cloud, downloaded from the ENA onto one of our cloud-compute servers and subjected to $k$-mer counting with KMC3 (ref. 62) to generate the full $k$-mer spectrum. If the median $k$-mer count on the spectrum was less than 2, the sample was further processed without any cleaning. Otherwise, the sample was subjected to cleaning with the standard graph cleaning procedure implemented in MetaGraph, with pruning tips shorter than $2k$ (for all these datasets $k$ was set to 31) and using an automatically computed $k$-mer abundance threshold for pruning low-coverage unitigs, with a fallback threshold value of 3. This cleaning procedure was applied for SRA-Fungi, SRA-Plants, SRA-Human and SRA-Metazoa.

For read sets of the SRA-MetaGut dataset, the sequencing depth was typically low, and we therefore applied a more lenient cleaning strategy. Namely, we switched off the singleton filtering (that is, we initially kept all $k$-mers that appear only once) on the $k$-mer spectrum and used a constant cleaning threshold of 2 during graph cleaning to remove all unitigs with a median $k$-mer abundance of 1.

For each dataset, we first constructed a joint canonical graph with $k = 31$ (including for each indexed $k$-mer its reverse complement) from the cleaned contigs and then transformed it into a primary graph (storing only one form of each $k$-mer and representing the other implicitly). Finally, using the same cleaned contigs, we annotated the joint primary graph with sample IDs to construct the annotation matrix. Each input sample thereby formed an individual column of the annotation matrix. The annotation matrix was then transformed to the RowDiff<Multi-BRWT> representation for higher compression and faster queries. The graph was, in turn, transformed to the small representation. The exact commands and scripts are available at GitHub (https://github.com/ratschlab/metagraph_paper_resources).

## SRA subset composition
Here we provide a detailed description of each of the 6 datasets.

**SRA-Microbe.** This dataset was first used to construct the BIGSI index[32]. Consisting of 446,506 microbial genome sequences, this dataset once posed the largest indexed set of raw sequencing data. However, at the time of performing our experiments, it represented only an outdated snapshot of the corresponding part of the SRA. Nevertheless, we decided to keep the same sequence set for this work to enable direct comparison and benchmarking. A complete list of SRA IDs contained in this set is available as file TableS1_SRA_Microbe.tsv.gz (with further information available in TableS10_SRA_Microbe_McCortex_logs.tsv.gz and TableS11_SRA_Microbe_no_logs.tsv) at GitHub (https://github.com/ratschlab/metagraph_paper_resources). For details on how the set of genomes was selected, we refer to the original publication[32].

**SRA-Fungi.** This dataset contains all samples from the SRA assigned to the taxonomic ID 4751 (Fungi) specifying the library sources GENOMIC and METAGENOMIC and excluding samples using platforms

PACBIO_SMRT or OXFORD_NANOPORE. In total, this amounts to 149,607 samples processed for cleaning. Out of these, 138,158 (92.3%) could be successfully cleaned and were used to assemble the final MetaGraph index. All sample metadata were requested from NCBI SRA on 25 September 2020 using the BigQuery tool on the Google Cloud Platform.

**SRA-Plants.** This dataset contains all samples from the SRA assigned to the taxonomic ID 33090 (Viridiplantae), specifying the library source GENOMIC and excluding samples using platforms PACBIO_SMRT or OXFORD_NANOPORE. In total, this amounts to 576,226 samples processed for cleaning. Out of these, 531,736 (92.3%) could be successfully cleaned and were used to assemble the final MetaGraph index. All sample metadata were requested from NCBI SRA on 17 August 2020 using the BigQuery tool on the Google Cloud Platform.

**SRA-Human.** This dataset contains all samples of assay type WGS, AMPLICON, WXS, WGA, WCS, CLONE, POOLCLONE, or FINISHING from the SRA assigned to the taxonomic ID 9606 (*Homo sapiens*) specifying the library source GENOMIC and excluding samples using platforms PACBIO_SMRT or OXFORD_NANOPORE. In total, this amounts to 454,252 samples processed for cleaning. Out of these, 436,502 (96.1%) could be successfully cleaned and were used to assemble the final MetaGraph index. All sample metadata were requested from NCBI SRA on 12 December 2020 using the BigQuery tool on the Google Cloud Platform.

**SRA-Metazoa.** This dataset contains all samples from the SRA assigned to the taxonomic ID 33208 (Metazoa) specifying the library source GENOMIC and excluding samples using platforms PACBIO_SMRT or OXFORD_NANOPORE. In total, this amounts to 906,401 samples processed for cleaning. Out of these, 805,239 (88.8%) could be successfully cleaned and were used to assemble the final MetaGraph index. All sample metadata were requested from NCBI SRA on 17 September 2020 using the BigQuery tool on the Google Cloud Platform.

**SRA-MetaGut (human gut microbiome).** This group contains all sequencing samples of the assay type WGS and AMPLICON from the SRA assigned to the taxonomic ID 408170 (human gut metagenome), excluding samples using platforms PACBIO_SMRT and OXFORD_NANOPORE. In total, this amounts to 242,619 samples, where 177,759 (73.3%) were AMPLICON and 64,860 (26.7%) were WGS samples. All these samples were successfully cleaned and were used to assemble the final MetaGraph index. All sample metadata were requested from NCBI SRA on 01 October 2020 using the BigQuery tool on the Google Cloud Platform.

The complete lists of all samples (including the list of successfully cleaned ones) for each subset are available at GitHub (TableS5_SRA_MetaGut.tsv.gz; https://github.com/ratschlab/metagraph_paper_resources).

## Indexing GTEx data
The 9,759 raw RNA-seq samples of the GTEx project have become a de facto reference set for the study of human transcriptomics[43]. All available RNA-seq samples that were part of the version 7 release of GTEx were downloaded through dbGaP to our compute cluster of ETH Zurich. A list of all of the samples used is available at GitHub (TableS7_GTEX.txt; https://github.com/ratschlab/metagraph_paper_resources).

Each sample was individually transformed into a graph using $k = 31$ and then cleaned with the standard graph cleaning algorithm implemented in MetaGraph, with trimming tips shorter than $2k$ and using an automatically computed coverage threshold with the fallback value of 2 for removing unitigs with low median $k$-mer abundance. All resulting cleaned contigs were assembled into a joint canonical de Bruijn graph and then transformed to the final primary graph. Using the typical workflow, the primary joint graph was annotated using the cleaned contigs extracted from each sample, generating one label per sample.

All individual annotation columns were finally collected into one matrix and transformed into the RowDiff<Multi-BRWT> representation.

When performing the indexing with $k$-mer counts (row 'GTEx with counts' in Table 1), we applied an additional smoothing of $k$-mer counts within cleaned unitigs to facilitate the compression. We used a smoothing window of size 60. That is, for each $k$-mer of a cleaned unitig, its count was replaced with the median abundance of 30 $k$-mers before it in that unitig and 30 after. This smoothing window is much smaller than the expected transcript length. However, it was sufficient to considerably reduce the annotation size (from 184 GB when indexing the original counts to 76 GB).

## Indexing the TCGA RNA-seq cohort

TCGA has collected RNA-seq samples on the same order of magnitude from primary tumours, spanning across more than 30 cancer types, constituting a central resource for cancer research[42]. We downloaded the data from the Genomic Data Commons Portal of the NCI. A list containing all processed samples is available as file TableS8_TCGA.tsv.gz at GitHub (https://github.com/ratschlab/metagraph_paper_resources). In total, the index contains 11,095 individual records spanning all available TCGA cancer types. We used the same indexing workflow as for GTEx. Similarly to GTEx, we have also constructed MetaGraph indexes with $k$-mer counts for TCGA (Table 1).

## Indexing environmental metagenome samples (MetaSUB)

This dataset contains 4,220 WMGS samples (the pilot dataset) collected from the environment through the MetaSUB consortium[8]. The swabs were collected at different locations and from different objects, where we also contributed to data collection by collecting swabs from benches, ticket machines and various other objects at different tram stops and train stations in Zurich. When sampling, each swab was annotated with additional data, including the location of sampling, the type of object from which the swab was collected, the material of that object, the elevation above or below sea level, and the station or line where the sample was collected. The swabs were then sent for further processing to the sequencing team. For more details about DNA extraction and sequencing, refer to the original publication[8].

The raw data (read sets) can be downloaded using the MetaSUB utils[84]. A list of all sample IDs used in this study is available as file TableS6_MetaSUB.csv.gz at GitHub (https://github.com/ratschlab/metagraph_paper_resources).

All input samples were directly assembled into canonical de Bruijn graphs (sample graphs) with $k = 41$. All graphs were then cleaned with the standard graph cleaning procedure implemented in MetaGraph, with pruning tips shorter than $2k$ and removing unitigs depending on coverage (automatically computed based on $k$-mer spectrum). If no threshold could be computed by the algorithm, we used 3 as a fallback value (an evaluation of this cleaning strategy is shown in Supplementary Fig. 15). The cleaned graphs were transformed into primary contigs, which were then used to assemble a joint graph and annotate it. We annotated the graph with sample IDs, which is the most fine-grained annotation we could construct. Thus, each sample was transformed into a single annotation column in the final MetaGraph index. All the annotation columns were finally aggregated into a joint annotation matrix compressed with the RowDiff<Multi-BRWT> representation. The additional metadata, such as the location, the object and the surface material, were written to a separate table and could easily be retrieved for any sample ID.

## Indexing the RefSeq and UniParc collections

The NCBI RefSeq database[9] contains a non-redundant collection of genomic DNA sequences (all assembled reference genome sequences), transcripts and proteins.

We indexed all 32,881,422 nucleotide sequences from release 97 of the RefSeq collection, a total of 1.7 Tbp, which takes 483 GB when compressed with gzip -9, using a de Bruijn graph $k$-mer index with $k = 31$. We annotated $k$-mer coordinates within buckets split by Taxonomy IDs (85,375 annotation columns with tuples of $k$-mer coordinates). The graph was constructed in basic mode (non-canonical, non-primary), as all the sequences of the collection are assemblies and are therefore of a determined orientation.

As expected, the compression ratio (the ratio between the compressed input and the index size) is lower than the raw sequencing read sets at 3.3 bp per byte (Table 1). Our index forms an alternative to the commonly used BLAST database[27,28,85] for competitive high-throughput search[5].

For amino acids, we indexed the UniParc collection of non-redundant sequences (release 2023_04), containing 210 gigaresidues, using a basic-mode graph and $k$-mer coordinate annotations to ensure lossless encodings. As expected, the compression ratio is low, at 1.7 amino acids/byte.

## Indexing global ocean microbiome (*Tara Oceans*) data

This collection (v.1.0) contains 34,815 genomes reconstructed from metagenomic datasets from major oceanographical surveys and time-series studies with high coverage of global ocean microbial communities across ocean basins, depth layers and time[11]. In addition to metagenome-assembled genomes (MAGs) constructed from 1,038 publicly available metagenomes extracted from ocean water samples collected at 215 globally distributed sampling sites, the collection includes a set of single amplified genomes and reference genome sequences of marine bacteria and archaea from other existing databases. For more details on the data composition, refer to the original publication[11].

We constructed an index for this collection (summary information is provided in Table 1) using a de Bruijn graph of order $k = 31$ constructed in the basic mode. This index encodes the coordinates of the $k$-mers within individual assembled genomes and therefore losslessly represents the input sequences. Notably, it still achieves a compression ratio of 4.2 bp per byte. We also indexed the raw assembled scaffolded contigs (360 gigabases) in an annotated graph with 318 million annotation labels. Owing to the very large number of annotation columns, in contrast to the annotation matrices in other indexes typically represented in the Multi-BRWT format, for this index, we represent the RowDiff-transformed annotation matrix in the RowFlat format for fast row queries.

## Indexing a subset of the Logan dataset

Building on the publicly available Logan resource[13], we grouped samples available in Logan (v.1.0) into subsets based on their phylogenetic relation. On the basis of the organism field in the metadata, we grouped samples together if at least 10,000 samples shared the same label. For the remainder, we mapped all samples to the NCBI taxonomy. Beginning at the species level, we aggregated samples into the same group, if they shared the same taxonomic assignment and formed a group of at least 1,000 samples. For the remaining samples yet ungrouped, we repeated the taxonomic grouping with the next-higher taxonomic levels (genus, family, order, superclass, kingdom, superkingdom) in the same manner. Yet ungrouped samples were assigned to a special group other. We performed the above procedure separately for DNA and RNA samples. Once a group was formed, we split it in subgroups based on the number of cumulative unique contig-level $k$-mers based on the Logan metadata. We processed the samples in order of reverse release date and started a new subgroup whenever the cumulative $k$-mer count of 500 billion was exceeded; except for groups metagenome and other, and groups above the superclass level. We indexed each sub-group independently, directly downloading the Logan contig-level samples from the cloud repository. If the contig-level sample was not available, we used the unitig-level sample. We tried downloading each sample up to ten times and marked the sample as not found if none of the attempts were successful. After download, we built a joint canonical graph from all inputs, without cleaning. We then transformed the canonical graph into its primary representation. We then built a joint annotation on the

primary graph from the same input samples and subsequently computed the RowDiff sparsification of all annotation columns. Finally, all annotation columns were compressed with Multi-BRWT compression using the default settings. Optionally, we transformed the primary graph from its default stat representation into small.

### Indexing the UHGG

We built two indexes using the assemblies from v.1.0 of the dataset. The first index, the UHGG catalogue, contains 4,644 reference genomes, while the other index contains 286,997 non-redundant genomes.

### Experiments

This section summarizes the experimental setup for the different results presented in this work.

**Benchmarking Mantis, Themisto, Bifrost and Fulgor.** For indexing and querying $k$-mer sets with these lossless indexing tools, we used Mantis (v.0.2.0), Themisto (v.3.2.2), Bifrost (v.1.3.5) and Fulgor (v.3.0.0).

**Benchmarking COBS and kmindex.** When indexing subsets of the collection of bacterial and viral genomic read sets[32] in our evaluation experiments, we used COBS[33] (commit 1cd6df2) with four hash functions and the target false-positive rate of 5%. For kmindex (v.0.5.3), we partitioned the samples into groups based on their $k$-mer counts, with one group per order of magnitude. We then constructed 28-mer indexes with false-positive rates of 5% for each group to leverage the findere algorithm[86] for querying 31-mers at a reduced false-positive rate.

**Experiment discovery on SRA graphs.** We evaluated each graph using 300 randomly selected samples from their respective input samples. To generate a query file for a graph, we randomly selected 100 reads (or the entire read set if fewer than 100 reads are available) from each of the 300 selected samples, resulting in query files of 30,000 reads per graph. To generate auxiliary reads with errors, we selected subsets of the original query sets such that 10 random reads would be selected from each sample. We then introduced substitution errors to these reads with probabilities 0.1%, 1%, 2%, 5% and 10% and insertions–deletions at 10% of the substitution probability using Mutation-Simulator[87] (v.3.0.1). Given these read sets, we then discarded all reads of which the median $k$-mer multiplicities were below the unitig cleaning thresholds determined by the cleaning procedure (see the 'Graph cleaning' section of the Methods).

We evaluated experiment discovery at two different levels of granularity: (1) mapping to individual sample graphs (Extended Data Fig. 3a); and (2) mapping to joint annotated graphs (Fig. 3a,b). When mapping to joint graphs, we considered only mapping results that retrieved the ground-truth label of each query read. For all granularities, we mapped the reads through both exact $k$-mer matching and label-consistent sequence-to-graph alignment using SCA[6]. We measure how well the reads aligned as the percentage of characters in the query that are covered by at least one reported mapping.

**Human gut resistome and phageome exploration.** We queried all AMR genes from the Comprehensive Antibiotic Resistance Database (CARD) database (v.3.2.7)[44] and all bacteriophages from RefSeq Release 218 (ref. 45). We selected bacteriophages by selecting all viral sequences with the term phage in their header. We mapped these sequences to all accessions in the SRA-MetaGut index representing WMGS samples. We recorded an accession as a match to a query if at least 80% of the query's $k$-mers exactly matched a $k$-mer in the accession. We then reduced the pools of AMR genes, SRA accessions and RefSeq bacteriophages to those for which at least one match was found.

To measure the degree of association for each AMR gene family–bacteriophage pair, we computed two binary vectors where each index represents a gut microbiome sample. The first vector indicates the presence of at least one gene match from the gene family and the second vector indicates a match to the phage. We then measure the association using the Matthews correlation coefficient (denoted by $Corr_{MCC}$) if both vectors indicate at least 5 present matches (value 1) and at least 5 absent matches (value 0).

When measuring the growth of resistance to antibiotics over time in each continent, we normalized the counts in the confusion matrix before computing $Corr_{MCC}$ to correct for differing numbers of samples deposited in each year and from each continent. If we denote the number of accessions from a continent $C$ by $n_C$ and the number of accessions from a year $Y$ by $n_Y$, we normalized the counts by letting each accession from continent $C$ and year $Y$ contribute a count of $c = \frac{n_N}{n_C \cdot n_Y}$ instead of 1. $n_N$ is a scaling factor applied to the four counts in the confusion matrix so that their sum equals the total number of accessions considered. We use the same scaling factor $c$ for each count when fitting a linear regression model to the match counts to determine drug-resistance growth over time for each continent.

We compute $P$ values for each gene-to-phage correlation $Corr_{MCC}$ and each drug resistance growth linear regression slope $s$ through permutation testing. For each analysis, we compute 100 permutations of the antibiotic and gene family indicator vectors, respectively, and compute $P$ values relative to the resulting null distributions.

In Fig. 4a, we only plot a gene family or a phage if it has at least one significant correlation with $Corr_{MCC} > 0.25$. In Fig. 4b, we report all antibiotics for which we measure statistically significant growth in at least one continent (modelled through a binomial GLM using the Python statsmodels package v.0.14.0). All $P$ values are corrected using the Benjamini–Yekutieli procedure to a family-wise error rate of 0.05 and are considered to be significant if they are $P < 0.05$ after correction (using the Python scipy package v.1.11.3).

**Survey of BSJs.** To generate the list of candidate trans-junctions, we iterated over all genes present in the GENCODE annotation (v.38) and generated for all transcripts a list of hypothetical BSJs. Assuming that a transcript consists of exons e1, e2, e3 and e4, we would connect the donor site for all exons starting from exon e2 with the acceptor site of all previous exons. The example transcript above would generate the following BSJ candidates: e2-e1, e3-e2, e3-e1, e4-e3, e4-e2, e4-e1. We included only exons with a length of at least 30 bp (which equals $k - 1$ for $k = 31$). For all junctions, we extracted the genomic sequences in a $\pm 1.5 \times k$ window around the junction, resulting in query sequences of 90 bp in length. We aligned all 4,052,768 candidate queries against the TCGA and GTEx MetaGraph indexes using the default metagraph align regime. Moreover, we also aligned all queries to the GENCODE (v.38) reference transcriptome using bwa-mem (v.0.7.17-r1188)[88] and against the hg38 human reference genome (GRCh38.p13, packaged with GENCODE v.38) using STAR (v.2.7.0 f)[89]. We conservatively retained only full-length matches against the graph, which removed most of our candidates, retaining 10,257 and 10,369 candidates for TCGA and GTEx, respectively. After making the candidate list unique over sequences and removing genome and transcriptome hits from the previous STAR and bwa-mem alignments, we retained 2,536 and 1,248 candidates for TCGA and GTEx, respectively. For our validation set, we downloaded accession GSE141693 from the Gene Expression Omnibus, containing experimentally confirmed circular RNAs[48]. We lifted over all annotations from hg19 to hg38 using the UCSC LiftOver tool[90], which could map all but 70 of the 87,555 circular RNA annotations. For all junctions, we extracted queries as windows $\pm 1.5 \times k$ around the junctions, resulting in 90-bp query sequences. Alignment against MetaGraph indexes, the reference genome and transcriptome, as well as subsequent filtering, was analogous to the in silico generated BSJ queries.

**Theoretical model for exact $k$-mer matching.** To derive a theoretically expected number of random matches to our 100-study subset (Fig. 5c),

we assume that a random sequence containing 100 $k$-mers (the average length in the query set used for Fig. 5a) is matched against a sample containing 252,631,773 $k$-mers (the average sample size in our 100-study subset). We use a hypergeometric distribution (that is, random sampling without replacement) to compute the probability of the query sequence having at least one matching $k$-mer in a sample. Using this as a success probability for a Bernoulli random variable, we multiply this probability by the number of samples in our 100-study subset to get the expected number of matches to the subset. We then scale this up to the entire SRA.

**Cloud-based query experiments.** Our index files computed from the Logan resource are available on AWS S3 storage. Analogous to the indexes formed from taxonomic grouping of input samples, we also generated 50 disjoint input sets, each aggregating the raw data of 100 randomly selected SRA studies. For 47 of the groups, we were able to construct a MetaGraph index using the same strategy as for the other Logan-based indexes. For three groupings, no index could be constructed because exceptionally large studies were selected, preventing the assembly of a single joint index. In addition to these sets, we also selected a subset of 21 index groups from the Logan set that contained only metagenome samples, as they represent the largest input diversity. Specifically, we selected metagenome groups 30, 50, 51, 57, 77, 85, 87, 110, 125, 133, 169, 184, 189, 192, 248, 253, 273, 316, 338, 376 and 385. Based on the total number of unique $k$-mers in each index group, we were able to extrapolate our query results to the entire index set.

We selected reads from the samples used to construct the 100-studies index described in Table 1 to construct query sets of varying sizes. First, we selected 300 random samples from this set. Then, given an integer $N$, we selected $N$ random reads from each of these 300 samples and filtered these by selecting reads of length at least 100 bp and at most 250 bp. We generated query sets for $N \in \{1, 50, 500, 5,000, 50,000, 5,000,000\}$. Moreover, we selected 1 out of every 20 reads from the $N = 1$ query set to generate our smallest query set.

We used an AWS Cloud Formation template (https://github.com/ratschlab/metagraph-open-data) to deploy the querying infrastructure, including AWS Batch compute environments and job queues on r6in, hpc7g and hpc7a EC2 instances, a Step Function and Lambdas for job scheduling and cost estimations. On this setup, we searched all query files in all 47 random study indexes. Smaller queries, for which the execution time is bounded by downloading the index, were processed on r6in instances, while the larger queries, for which the execution time is bounded by the actual runtime, were processed on hpc7g instances, or, when an index did not fit in the available RAM, on hpc7a. After processing all indexes, the total execution time of active EC2 instances was computed to estimate the EC2-related costs of the query. 'Mountpoint for S3' was used to avoid staging data on disk and load it directly into RAM instead. Below are the exact MetaGraph CLI parameters that were used for running the queries. Exact match queries: --batch-size 8000000 -p $(nproc) --threads-each 1, 32 or 48 --query-mode labels --min-kmers-fraction-label 0.7 --min-kmers-fraction-graph 0.0 --num-top-labels inf. Fast alignment queries: --batch-size 200000 -p $(nproc) --align-min-exact-match 0.85 --align-min-seed-length 27. Sensitive alignment queries: --batch-size 200000 -p $(nproc) --align-min-exact-match 0.0 --align-min-seed-length 27. A --threads-each parameter was chosen based on the instance type for r6in, hpc7g and hpc7a instances, and $(nproc) depends on the specific instance on which the query was processed. The GitHub repository provides further details on a reproducible deployment of the AWS querying infrastructure.

## The architecture and the implementation of MetaGraph Online
The architecture of the MetaGraph Online service is schematically shown in Supplementary Fig. 10. The backend server (Supplementary Fig. 10 (middle)), which is implemented as a Flask v.2.3.0 application, provides a web interface and generates dynamic web pages for submitting search queries and viewing the query results. The user queries are processed and transformed into a search query, which is then passed to the remote servers hosting the MetaGraph indexes (Supplementary Fig. 10 (right)). Once the query has been executed by the respective MetaGraph server, the results are sent back to the backend server and an aggregated summary is rendered to the user. Each MetaGraph index is hosted on a remote server by running the MetaGraph tool in server mode (Supplementary Fig. 10 (right)). The server applications run independently and are distributed across the available machines. Each MetaGraph server receives HTTP requests formed by the central backend server on user search requests. The communication between the central backend server and the other remote MetaGraph servers happens through the Python v.3.9 API. For seamless compatibility, we also made the backend server redirect user requests and provide the same web API for querying MetaGraph servers (for example, from Python or as a simple HTTP request) as if the MetaGraph tool is locally running in server mode. The backend of MetaGraph Online is implemented as a Flask application. This web application is deployed in a Docker container (v.1.13.1; API v.1.26) using the Nginx (v.1.16.1) server as a backend. For each search query from the user, it forms a request accordingly and sends it through the Python API client to the MetaGraph servers hosting the indexes. We run these MetaGraph servers in Docker containers on the same or other machines in a federated manner. Moreover, the web application emulates the usual MetaGraph API by redirecting all requests to the respective individual MetaGraph servers hosting the indexes.

### Reporting summary
Further information on research design is available in the Nature Portfolio Reporting Summary linked to this article.

## Data availability
Indexes of public sequencing data are available at S3 Bucket s3://metagraph, also accessible at https://metagraph.s3.amazonaws.com/index.html. Further details and instructions for accessing the data are provided at GitHub (https://github.com/ratschlab/metagraph-open-data). The web service is publicly available for web and API queries at https://metagraph.ethz.ch/search. Additional resources for this project, including sample metadata, interactive notebooks and analysis scripts are available in GitHub (https://github.com/ratschlab/metagraph_paper_resources). A list of all data sources and software used in context of this work is available as Supplementary Information.

## Code availability
The source code of the MetaGraph software is available under a GPLv3 License at GitHub (https://github.com/ratschlab/metagraph; commit 30f6280) Besides the core library in C++ and a Python API client, the project includes unit tests, benchmarks and integration tests for testing the APIs and the command line interface, a set of Snakemake workflows (compatible with versions ≥5) for simplified index construction and detailed documentation.

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

**Acknowledgements** This work was supported through the funds of ETH Zurich (to G.R.). M.K. and H.M. were supported by the Swiss National Science Foundation grant no. 407540_167331 "Scalable Genome Graph Data Structures for Metagenomics and Genome Annotation" as part of the Swiss National Research Programme (NRP) 75 'Big Data'. H.M. was also partially funded by the Personalized Health and Related Technologies (PHRT) Transition Postdoc Fellowship Project 2021/453. O.K. was partially funded through the Swiss National Science Foundation Grant No. 200550 to A.K. The project was supported by the Open Research Data Program of the ETH Board. Computations for this work were carried out on hardware financed through the SNF R'Equip program grant no. 213236 and the ETH Scientific Equipment Program to A.K. Parts of the computations were made possible through the SwissAI Initiative through a grant from the Swiss National Supercomputing Centre (CSCS) under project ID a02 on Alps. Computational data analysis was performed at the Leonhard Med secure trusted research environment at ETH Zurich. We thank the staff at Google for providing a package of free compute credits on the Google Cloud Platform; the staff at AWS Open Data Sponsorship program for providing free S3 cloud storage; the members of the MetaSUB international consortium for early access to raw sequencing data and their feedback on early versions of the geolocation DNA sequence search; the NCBI staff working on maintaining and developing the SRA for their support and interest in our project; the members of the Biomedical Informatics Group at ETH Zurich for constant feedback and input on the project; A. Bhatt, R. Chikhi, Y. W. Yu and the members of the research community for providing encouragement, feedback, components and resources that were important for this project, specifically, E. Birney, Z. Iqbal, B. Paten, R. Patro, R. Chikhi, C. Mason, D. Huson, R. Durbin, J.-P. Vert, E. Pamer, M. Carty, M. Stanke and T. Hoefler; the donors and their families who contributed data to the GTEx and TCGA projects. This study contains data gathered by the GTEx project available at dbGaP (phs000424.v7.p2). This study further contains data gathered by the TCGA project available at dbGaP (phs000178.v1.p1). The authors are also grateful for the ability to use the publicly available data gathered by the gnomAD project. Figure 1 (https://BioRender.com/3w8zkl3) and Supplementary Fig. 10 (https://BioRender.com/9lnzhrz) were created using BioRender.

**Author contributions** M.K., H.M., G.R. and A.K. conceived and designed the study. M.K., H.M., D.D., G.R. and A.K. developed algorithms, data structures and workflows. M.K. and H.M. implemented key software components. M.K., H.M. and A.K. designed and performed computational experiments. M.K., H.M., D.D. and A.K. generated large-scale sequence indexes. D.D. contributed to project scaling and supported the global indexing strategy. O.K. established the cloud query infrastructure and ensured reproducibility of experiments. M.Z. extended query interfaces, the website and API infrastructures. C.B. implemented optimizations and utility functions for core graph components. A.K. and G.R. supervised the work and provided project management and funding. M.K., H.M., G.R. and A.K. wrote the manuscript. All of the authors read and approved the final manuscript.

**Funding** Open access funding provided by Swiss Federal Institute of Technology Zurich.

**Competing interests** G.R. is a cofounder of Computomics and one of its shareholders. Computomics' products are not directly related to this publication. The other authors declare no competing interests.

**Additional information**
**Correspondence and requests for materials** should be addressed to Gunnar Rätsch or André Kahles.

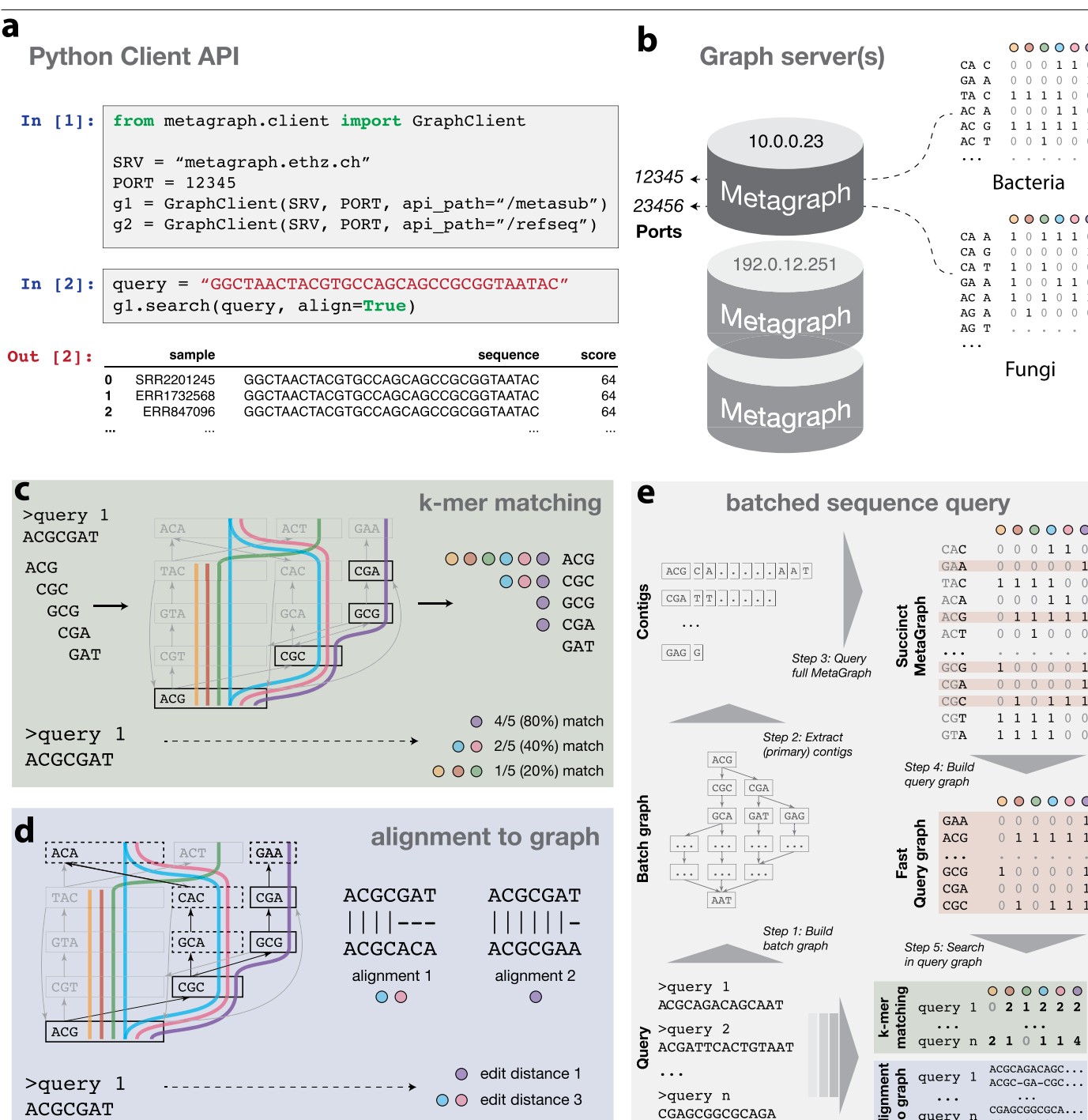

**Extended Data Fig. 1 | MetaGraph API and search approaches. a)** MetaGraph is designed to support a client-server infrastructure as exemplified here with a script in Python. In a few steps, several remote (or local) graph instances can be created and queried interactively. Results are returned as a data frame that can be used for further analyses. **b)** Conceptual overview of remote or local index distribution. Every graph index runs on a separate server, accepting queries via the client API. **c-e)** Schematic representation of two main approaches for sequence search. **c)** Counting exact k-mer matches between query and graph.

**d)** Alignment finds all closest paths within a given edit distance. **e)** Batched sequence search retrieves a decompressed subgraph (query graph) from the full compressed annotated graph for subsequent query. All query sequences are combined into an intermediate batch graph, which is then traversed to extract contigs to be queried against the full index. Hits and their corresponding annotations are aggregated to construct the final query graph, which is then searched against with the original query sequences.

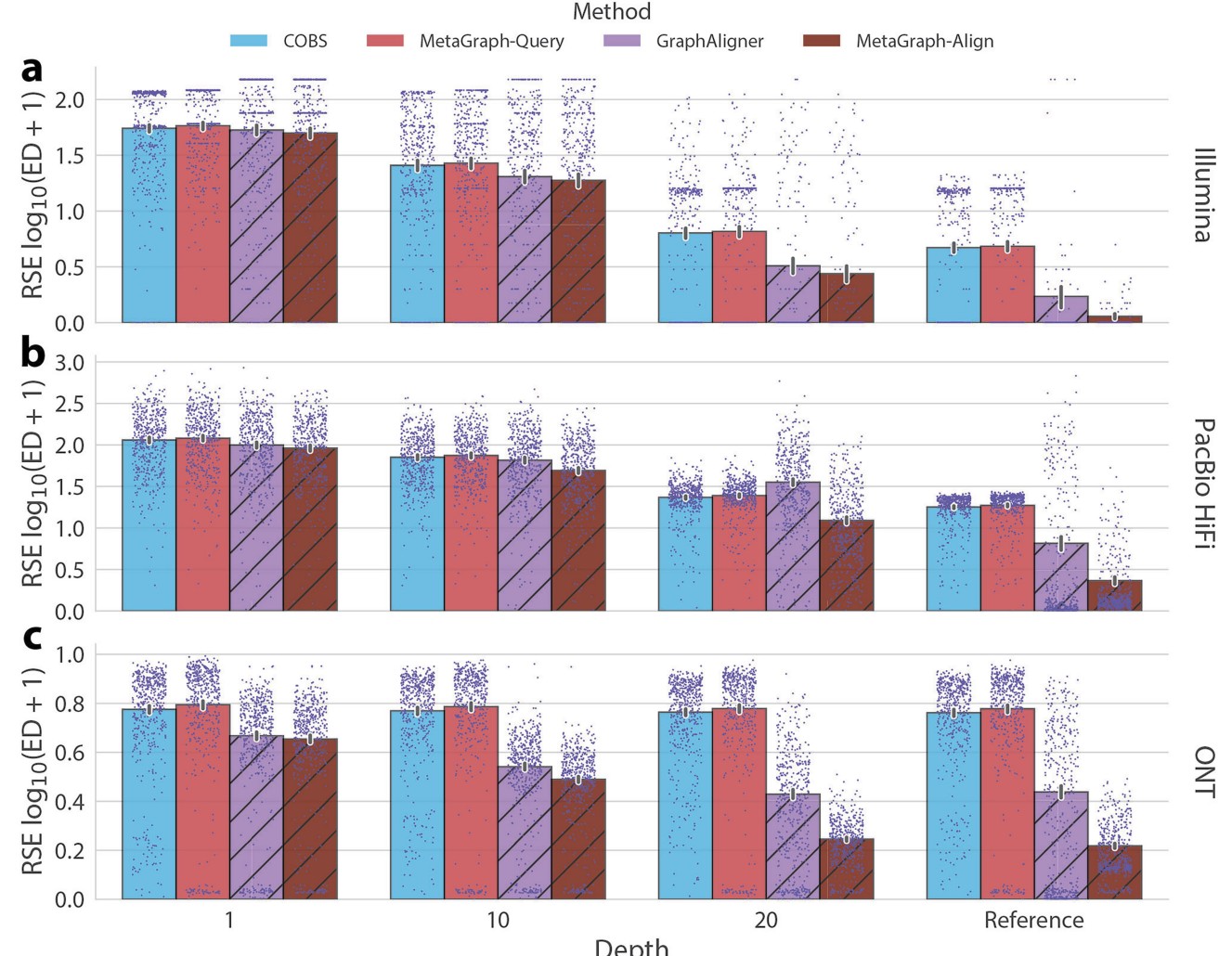

**Extended Data Fig. 2 | Accuracy of sequence search approaches.** Accuracy of sequence search approaches for queries of **a**) Illumina-type, **b**) PacBio HiFi-type, and **c**) ONT-type simulated reads. All graphs (indexes) were constructed from Chromosome 21 of the CHM13 v2.0 Homo sapiens reference genome, or its simulated Illumina-type reads at different sequencing depths, applying our usual indexing workflow with graph cleaning. Accuracy is measured as the mean RMSE between the logarithm of the edit distance computed by each method (COBS commit 1cd6df2 and GraphAligner v 1.0.17b) and gold-standard edit distances computed with edlib (commit 931be2b), measured across 1000 bootstrap samples of 500 simulated Chromosome 21 query reads. The query reads are simulated from all Chromosome 21 assemblies accessible on GenBank as of October 21, 2024. Error bars represent 95% confidence intervals of the mean. Bar hatching indicates a method that uses sequence-to-graph alignment instead of exact k-mer matching. Illumina-type reads were simulated using ART (v2.5.8). PacBio HiFi-type subreads and ONT-type reads were simulated using pbsim v3.0.0. HiFi reads were generated using PacBio CCS v6.4.0.

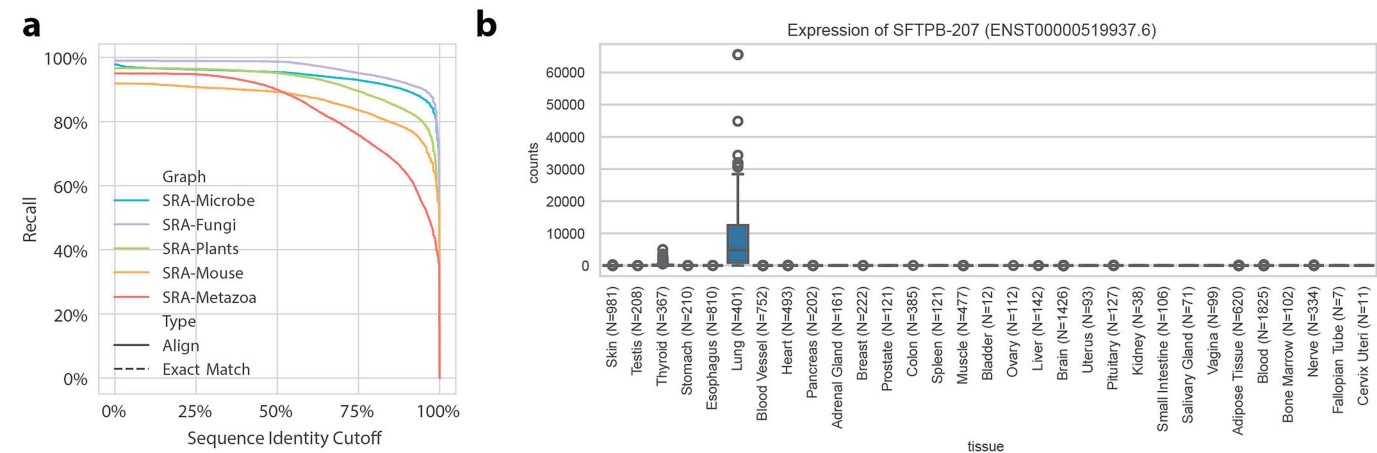

**Extended Data Fig. 3 | Completeness and specificity. a**) Complementary ECDF curve showing the fraction of reads that can be re-aligned to indexes of their respective original samples (y-axis) requiring at least x% matching positions (x-axis) for a range of indexed SRA datasets. **b**) Distribution of sample counts retrieved from the MetaGraph GTEx index, containing N = 11,036 samples, for the transcript ENST00000519937.6 of the gene SFTPB, known to be specifically expressed in lung and thyroid tissues. Boxes range from first to third quartile, the centre lines represent the median, and the whiskers represent the 1.5× inter-quartile-ranges, outliers are shown as empty circles.

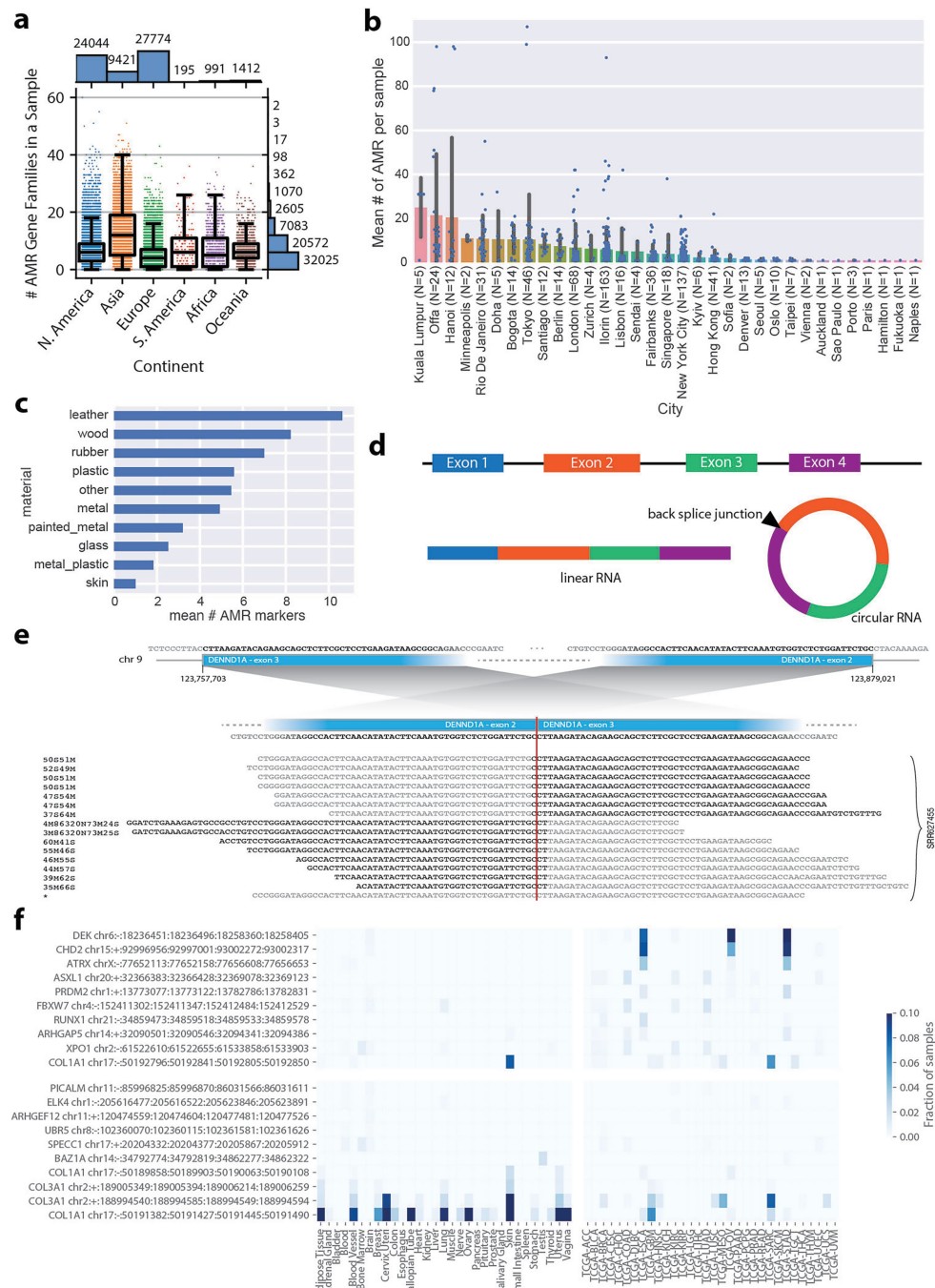

**Extended Data Fig. 4 | Additional applications. a)** Number of antimicrobial resistance (AMR) gene families per sample in the SRA-MetaGut human gut microbiome whole-genome sequencing samples (63,837 samples). Boxes range from first to third quartile, center line represents median, and whiskers represent 1.5× inter-quartile-range. **b)** Mean number of AMR markers per sample for different cities in the MetaSUB study. Error bars represent +/- one SD around the mean. **c)** Distribution of the mean number of AMR markers grouped by surface material based on all samples of the MetaSUB dataset **d)** Schematic of back-splice-junction formation in circular RNA. Exons 1 and 4 are connected in circular RNA (right) but not in linear RNA (left). **e)** Back-splice junction detection on the DENND1A gene in GTEx RNA-Seq sample SRR627455. **f)** Relative fraction of samples per tissue/project that show BSJ expression in GTEx (left panels) and TCGA (right panels) cohorts. Top/bottom panels show predominant TCGA/GTEx expression, respectively.

Gunnar Raetsch

# Reporting Summary

## Statistics

For all statistical analyses, confirm that the following items are present in the figure legend, table legend, main text, or Methods section.

| n/a | Confirmed | |
|---|---|---|
| ☐ | ☒ | The exact sample size (*n*) for each experimental group/condition, given as a discrete number and unit of measurement |
| ☒ | ☐ | A statement on whether measurements were taken from distinct samples or whether the same sample was measured repeatedly |
| ☐ | ☒ | The statistical test(s) used AND whether they are one- or two-sided<br>*Only common tests should be described solely by name; describe more complex techniques in the Methods section.* |
| ☐ | ☒ | A description of all covariates tested |
| ☐ | ☒ | A description of any assumptions or corrections, such as tests of normality and adjustment for multiple comparisons |
| ☐ | ☒ | A full description of the statistical parameters including central tendency (e.g. means) or other basic estimates (e.g. regression coefficient) AND variation (e.g. standard deviation) or associated estimates of uncertainty (e.g. confidence intervals) |
| ☐ | ☒ | For null hypothesis testing, the test statistic (e.g. *F*, *t*, *r*) with confidence intervals, effect sizes, degrees of freedom and *P* value noted<br>*Give P values as exact values whenever suitable.* |
| ☒ | ☐ | For Bayesian analysis, information on the choice of priors and Markov chain Monte Carlo settings |
| ☒ | ☐ | For hierarchical and complex designs, identification of the appropriate level for tests and full reporting of outcomes |
| ☒ | ☐ | Estimates of effect sizes (e.g. Cohen's *d*, Pearson's *r*), indicating how they were calculated |

*Our web collection on statistics for biologists contains articles on many of the points above.*

## Software and code

Policy information about availability of computer code

| Data collection | For data collection, we used the following software tools:<br>- sra_toolkit (v3.0.10)<br>- UCSC LiftOver tool (downloaded on 18.01.2021 from http://hgdownload.soe.ucsc.edu)<br>- KMC3 (https://github.com/karasikov/KMC; commit b163688)<br>- Google BigQuery (Service used; version N/A)<br>- ART (v2.5.8)<br>- pbsim (v3.0.0) |
|---|---|
| Data analysis | For the analysis of data, we have used the following software tools:<br>- Python (v3.9)-<br>- COBS (https://github.com/bingmann/cobs; commit 1cd6df2)<br>- kmindex (v0.5.3)<br>- Mantis (v0.2.0, https://github.com/splatlab/mantis; commit 0fb7dbb)<br>- Fulgor (v3.0.0)<br>- Themisto (v3.2.2)<br>- Bifrost (v1.3.5)<br>- BWA-MEM (v0.7.17-r1188)<br>- STAR (v2.7.0f)<br>- Snakemake (>= v5.0)<br>- GraphAligner (v1.0.17b)<br>- edlib (https://github.com/Martinsos/edlib; commit 931be2b) |

- Mutation-Simulator (v3.0.1)
- Flask (v2.3.0)
- statsmodels (v0.14.0)
- scipy (v1.11.3)
- Nginx (v1.16.1)
- Docker (v1.13.1; API v1.26)
- PacBio CCS (v6.4.0)

All code for the MetaGraph framework and the accompanying analysis scripts is publicly available on GitHub;
https://github.com/ratschlab/metagraph; commit 30f6280
https://github.com/ratschlab/metagraph_paper_resources
https://github.com/ratschlab/metagraph-open-data

For manuscripts utilizing custom algorithms or software that are central to the research but not yet described in published literature, software must be made available to editors and reviewers. We strongly encourage code deposition in a community repository (e.g. GitHub). See the Nature Portfolio guidelines for submitting code & software for further information.

# Data

Policy information about availability of data

All manuscripts must include a data availability statement. This statement should provide the following information, where applicable:
- Accession codes, unique identifiers, or web links for publicly available datasets
- A description of any restrictions on data availability
- For clinical datasets or third party data, please ensure that the statement adheres to our policy

Input data was collected from public sequence read archives, such as NCBI SRA, EBI ENA, or Genomic Data Commons using standard retrieval methods such as the sra toolkit or wget. In addition, we used public data provided in AWS S3 storage via the Logan project.

Following data sources have been used:
- NCBI Sequence Read Archive/ENA (samples available on January 11, 2025)
- RefSeq (release 97)
- UniParc (version 2023_04)
- UHGG (Unified Human Gut Genome, v1.0)
- Tara Oceans (v1.0)
- CARD (Comprehensive Antibiotic Resistance Database, v3.2.7)
- GENCODE annotation (v38)
- hg38 human reference genome (GENCODE v38)
- CHM13 human reference genome (v2.0)
- IsoCirc dataset (circular RNAs, GEO accession GSE141693)
- GTEx (Genotype Tissue Expression project, dbGaP phs000424.v7.p1)

Indexes of public sequence data are available at S3 Bucket s3://metagraph, also accessible via https://metagraph.s3.amazonaws.com/index.html. Further details and instructions for accessing the data can be found at https://github.com/ratschlab/metagraph-open-data. The web service is publicly available for web and API queries at https://metagraph.ethz.ch/search. Additional resources for this project, including sample metadata, interactive notebooks and analysis scripts are available in GitHub at https://github.com/ratschlab/metagraph_paper_resources. A list of all data sources and software used in context of this work is available as supplementary information.

# Research involving human participants, their data, or biological material

Policy information about studies with human participants or human data. See also policy information about sex, gender (identity/presentation), and sexual orientation and race, ethnicity and racism.

| Reporting on sex and gender | n/a |
|---|---|
| Reporting on race, ethnicity, or other socially relevant groupings | n/a |
| Population characteristics | n/a |
| Recruitment | n/a |
| Ethics oversight | n/a |

Note that full information on the approval of the study protocol must also be provided in the manuscript.

# Field-specific reporting

Please select the one below that is the best fit for your research. If you are not sure, read the appropriate sections before making your selection.

☒ Life sciences          ☐ Behavioural & social sciences          ☐ Ecological, evolutionary & environmental sciences

For a reference copy of the document with all sections, see nature.com/documents/nr-reporting-summary-flat.pdf

# Life sciences study design

All studies must disclose on these points even when the disclosure is negative.

| | |
|---|---|
| Sample size | The goal of our study was to build an index for biological sequencing samples at the petabase scale. For the SRA cohorts, we have selected all samples that were publicly available at the time of data access. For other cohorts (TCGA, GTEx, etc) we have included all available samples. That is, sample size was only limited through technical factors (e.g., availability for download). |
| Data exclusions | We have selected data to be indexed based on a pre-determined set of metadata values. For instance, data originating from 3rd generation long-read sequencing technologies has been excluded from parts of our study, as the error correction quality was not sufficient. All inclusion and exclusion criteria are clearly stated in the text. As the main point of our paper is that of scalability, the exclusion of a subset of inputs does not invalidate our findings. |
| Replication | We have replicated the key points of our manuscript (compressability, scalability) across 8 different input cohorts (all data is summarised in Tabl1 of the manuscript). |
| Randomization | For the extrapolation of index growth, we have subsampled all available input data uniformly at random. |
| Blinding | This is not applicable to our study as no subjects were recruited and only publicly available data was downloaded either exhaustively (comprising the full data sets) or as a subset selected uniformly at random. |

# Reporting for specific materials, systems and methods

We require information from authors about some types of materials, experimental systems and methods used in many studies. Here, indicate whether each material, system or method listed is relevant to your study. If you are not sure if a list item applies to your research, read the appropriate section before selecting a response.

## Materials & experimental systems

| n/a | Involved in the study |
|---|---|
| ☒ | ☐ Antibodies |
| ☒ | ☐ Eukaryotic cell lines |
| ☒ | ☐ Palaeontology and archaeology |
| ☒ | ☐ Animals and other organisms |
| ☒ | ☐ Clinical data |
| ☒ | ☐ Dual use research of concern |
| ☒ | ☐ Plants |

## Methods

| n/a | Involved in the study |
|---|---|
| ☒ | ☐ ChIP-seq |
| ☒ | ☐ Flow cytometry |
| ☒ | ☐ MRI-based neuroimaging |

## Plants

| | |
|---|---|
| Seed stocks | n/a |
| Novel plant genotypes | n/a |
| Authentication | n/a |

