## [Peer Review file · Nature]

Efficient and Accurate Search in Petabase-Scale Sequence Repositories

Corresponding Author: Dr Andre Kahles

Version 0:

Reviewer comments:

Referee #1

(Remarks to the Author)

Comments for Author

In the manuscript "Indexing All Life's Known Biological Sequences" by Karasikov et al., the authors make a valiant effort to broaden the ****searchable**** set of biological sequences. The splashiest contribution of this article is the indexing of "all life's known biological sequences" for search; they make petabases available for searching over and provide algorithms/webservices/software for doing so.

The second contribution are the algorithms for indexing and searching, which is in my mind technically much more challenging and also just as important of a contribution. This manuscript appears to be culmination of multiple prior efforts, some of which were previously published whereas others were only preprinted by not published.

- The MetaGraph algorithm was preprinted in 2020

- Alignment was improved upon: MetaGraph-MLA by Mustafa et al., 2022 bioRxiv and TCG Aligner by Karasikov et al, 2022 Genome Research)

- Better color compression (Karasikov et al., 2020, Journal of Computational Biology)

The engineering + algorithms are extremely impressive, and to me the main highlight of the paper, though because of the complicated prior history, it's a little hard to discern which algorithmic contributions should be considered novel to this manuscript.

Overall, this work is a very strong methods paper that has the potential to substantially move the field forward. However, there are many non-trivial caveats, especially with respect to presentation, experiments, benchmarking (or lack thereof). In its current form, I don't think this paper is suitable for Nature, but I could certainly see a revision making that cut.

Specific major points:

Point 1: The paper's title is far too suggestive. While the author's do not explicitly claim this work indexes ****all of life's known sequences****, the title strongly suggests just that. A claim that strong needs to be substantiated. To pick just one example, metagenomes are vastly underindexed. Given that microbes make up a huge diversity of life and that most have not/cannot be cultured, this is an important issue. The authors index SRA-MetaGut (all human gut metagenomes) and MetaSub (4000+ urban environmental metagenomes) as the two sets of metagenomes fully indexed. This excludes non-gut, non-human, host-associated microbiomes as well as terrestrial microbiomes such as soil and ocean (although TARA **assemblies** are indexed).

On line 153, I note that the authors collected their SRA dataset in April 2020, which is four years old at this point. Another recent paper Pebblescout (Nature Methods 2024) indexes all SRA metagenomes (and all SRA RNA-Seq experiments) up to ~2022, arguably more a comprehensive database than what is presented in this paper. My sense is that Pebblescout's indices are much less comprehensive and impressive than MetaGraph, but given the emphasis on exponential growth in sequencing capacity in the introduction, a gap of 4 years to the present seems like sufficient time for the dataset sizes to have more than doubled.

Point 2: Many of the figures (3C, 4A-C) focus on biological discovery, but these show proof-of-concept results rather than significant biological discoveries that are substantiated with follow-up. For example, 3C suggests that one can use MetaGraph to do large-scale querying of transcript abundances, but no benchmarking of this functionality is shown. This would be OK for a methods-based paper (e.g. in Nature Methods), but seems a bit limited for a Nature paper (though I defer to the Editor on whether this is actually an issue).

Point 3: As a follow-up to point 2, given that the most significant contributions are methods-based, there is a lack of comprehensive benchmarking, especially of recent methods, many of which themselves benchmark against the MetaGraph preprint.

Indexing:

- Fig 2A – Some newer methods seem to be omitted from benchmarking, including kindex (Lemane et al, 2024), Thjemisto (Alanko et al, 2023). Both methods claim better performance than MetaGraph in their papers. GGCAT is cited, but not compared against, either.

- Index building time is very relevant in practice. I do not see benchmarks for index building time, nor for continuous updating. The usefulness of a webserver like BLAST is partially in that it is kept up-to-date. How long would it take to rebuild (or update) the indexes on say a monthly basis? Would the index have to be built from scratch, or could it be delta'd?

Graph cleaning:

- Graph cleaning creates a **lossy** index, not a lossless index as claimed. This is an important point that needs to be cleaned up. The lossiness of MetaGraph is remarked upon by some of the competitor more recent papers, such as in the Pebblescout manuscript (Nature Methods 2024). Yes, it is technically true that MetaGraph provides a lossless representation of the **indexed** k-mer sets, but the data cleaning prior to choosing k-mers to index makes the overall process lossy.

- Furthermore, I would like to see benchmarks showing how graph cleaning contributes to the efficiency. Is the graph cleaning a major reason for MetaGraph's efficiency?

- Graph cleaning seems to be relatively aggressive (≤ 3 -coverage k-mers are pruned, but possibly more, according to Methods). This will remove a lot of low-abundance organisms' reads in Metagenomes, or lowly abundant transcripts. This point needs to be addressed. Fig. 3A does not adequately address this because it calculates the overall recall, but the recall of low abundance transcripts/organisms is interesting by itself. This is especially important for more careful interrogations of low-frequency bacterial species in a metagenome, rare isoforms in RNA-seq abundance quantification, or structural variations supported by a single high-quality long-read in say cancer biology.

Alignment/sequence search:

- More precision/sensitivity tests for alignment are likely needed. There is not test of precision that I am aware of. Figure 3A shows how the recall varies for reads mapping **to its own sample**, but there should probably be a test of how reads map to the combined metagraph index. Some notion of "false positive" should be used as a benchmarking metric. This is especially important since, unlike a BLAST search, k-mer matching does not give as obvious metrics for evaluating alignment uncertainty or goodness.

- On the same note, Extended Data Fig. 2 shows read mapping to an e-coli with MetaGraph's alignment algorithms. E. coli genomes are very simple compared to a complex eukaryotic genome. Could this experiment be repeated with Human or Wheat genomes?

- Additionally, although it is impressive to have a single software that indexes close to "all" biological sequence, nucleotide sequence search and protein sequence search do actually often have substantial differences. The k-mer based exact match alignment for nucleotide sequences works less well for protein sequences, where identity between similar protein sequences may be lower. Could the authors provide a benchmark of this?

Specific minor points:

The abstract claims that the compressed representation fits on a single consumer hard drive (~100 USD), without specifying exactly which databases fit. Given the setup, the implication is that "all" sequences in this compressed format will fit on a \$100 drive. At present, the highest capacity HDD I can find for around \$100 only stores 12 TB of data.

(<https://diskprices.com/>) However, in Table 1, the authors themselves say that SRA-Public to 01.01.2023 would take 130 TB to store the index. Even excluding that, the sum of all the other indices in the table seems to be just above 12 TB. Please clarify.

Although it is common folklore in the field that sequencing data has been growing exponentially (and it certainly was for a while), this appears to have leveled off in recent years. The authors cite a 2015 paper to substantiate their exponential claim. But, in the ENA statistics the authors cite, the run doubling time has been **increasing** over the last decade, from a

doubling time of around 10-15 months back in 2012, to nearly 45 months currently. Were growth actually exponential, the run doubling time would be constant over time. Thus, although sequencing data still grows quickly, the exponential claim appears to be outdated.

Line 49: the authors have made a distinction between (i) sketching techniques and (ii) methods employing Bloom filters. Taxonomies are hard in any field, but certainly many computer scientists would include Bloom filters as a type of early probabilistic sketch in this day and age. The authors choice of taxonomy is justifiable, but certainly should be made explicit.

Line 137: saying that 4.8 petabases = 2.5 petabytes is a bit confusing without reading the supplement on how those numbers are counted. Not sure what if anything can be done here to clarify things, but it was a bit confusing on first reading.

Line 285: "with higher error rates, such as PacBio's SMRT" PacBio HiFi reads have a lower error than short reads nowadays

Line 317: How could MetaGraph "facilitate large-scale learning tasks" and train LLMs?

Line 321: the authors have not demonstrated how sensitive MetaGraph sequence alignment can be before failing. I would like to know when it fails. Fig. 3A shows 2% mutation rate, but higher mutation rates should be tested. BLAST works with quite dissimilar query sequences, for example, and although MetaGraph should not be expected to be as sensitive, users should know when it fails.

Line 479: how is the gamma-distributed mean relevant

Line 780: what default value of seeds is used for alignment, $< k$?

Micro points:

Line 56: extra space before reference "alignment 34".

Line 60: this sentence is a bit awkward. Consider rewording.

Line 411: "one from4". Reference should be expanded here.

Line 1025: incorrect smart quotes. In latex need `and`.

Reference 48 is a duplicate of reference 39.

Reference 61 does not properly capitalize "rna"

Table 1: is "MGS" defined anywhere?

(Remarks on code availability)

Software comments

Overall, the software is extremely high-quality and well packaged for a bioinformatics software. There seems to have been dedicated software engineers for the software, which is extremely impressive. I had no problem running and installing the software. Documentation is well done. The web search is an impressive feat of engineering. It is a bit rough, but I think it has tremendous potential.

Web search suite

The web interface is decent looking, but quite simple. I think it could benefit from a few tune-ups. In general, I think more documentation on the website itself would be good.

- Alignment to graph option – this is not very interpretable for a general user. When should they use alignment vs normal k-mer querying?

- Performance is weird. Sometimes it's fast, sometimes it's quite slow and hangs.

- There should be an option to show only the best "X" number of hits. My browser ran out of memory.

- I think the authors could look at the Pebblescout (NCBI service) interface for inspiration on documentation, which I found a bit easier to use.

Error in the "Metazoa" index: (see attached doc)

The visualization of De Bruijn graphs seems to have an error (see attached doc):

Software online documentation:

- Documentation is very good where present, but could be a bit more complete. Docs on sequence-to-graph alignment, which seems like an important aspect, is missing.

- The quick start page is a bit overwhelming. Perhaps an (interactive) flow chart would help.

Referee #2

(Remarks to the Author)

The exponentially increasing, publicly available sequencing data hosted at the SRA is an invaluable resource for

researchers. However current methods for accessing and searching this data are limited, often requiring excessive time and computational resources. Karasikov/Mustafa and colleagues address this challenge with their innovative MetaGraph framework. They leverage advanced text data structures to create highly compressed and searchable indices of sequencing data.

The core of MetaGraph is an annotated de Bruijn graph, which represents large sets of reads efficiently through k-mers with custom associated data (dataset identifier, abundance, or genomic coordinates). The framework supports aligning query sequences to the graph, allowing higher-sensitivity alignments hence increasing the utility beyond providing a simple k-mer index. A web server for querying some of the indices is also provided.

The manuscript shows applications of MetaGraph to three use cases: association of AMR genes to bacteriophages, tracking AMR in human gut metagenomes across time, and searching back-splice-junctions in GTEX data.

From a computational perspective, the achievement is impressive. To the best of my knowledge, apart from NCBI PebbleScout which appeared years later and compresses data less well, no other combination of algorithmic methods and software architecture has been applied to indexing petabase-scale sequence data.

The main text is overall well-written and the Methods section is easy to follow (for a specialist).

Remarks in main text:

1. Title: I am unsure what is the stance of Nature for aspirational titles. 8.9% of the SRA (as of early 2024) has been indexed by MetaGraph (offline, in Table 1) and 1.6% of the SRA is searchable in MetaGraph Online. This contrasts with “all Life’s known biological sequences” that implies numbers close to 100%.
2. PebbleScout, an online web server which has indexed 7.3% of the SRA (all the metagenomes) and was published recently in Nature Methods, is absent from the manuscript yet should be included and its features/performance compared against MetaGraph.
3. Several other indexing schemes have since appeared, e.g. Themisto, Fulgor and kmindex. Even though the original 2020 MetaGraph article predates these methods, they should be benchmarked in this manuscript.
4. Regarding the sentence “Our indexes are freely available to the research community” in the Abstract, some are missing from the Indexes pages at <https://metagraph.ethz.ch/>: Uniparc, SRA-Metazoa mouse and SRA-Metazoa w/o human.
5. I remain unconvinced of the claim that the method can scale to the entire SRA at the projected costs, given that the most challenging datasets, raw environmental metagenomes, were excluded. How much time/RAM/disk was needed to construct the largest indices? (MetaGut, Plant, Metazoa)
6. Have SRA transcriptomic data been indexed? Given the library selection method (“GENOMIC”), it appears that this is not the case, but this is unnoted in the main text.
7. In the “back-splice junctions in GTEX and TCGA” section, TCGA was not surveyed.

Minor remarks in main text:

8. “When indexing the SRA as a whole, the final MetaGraph index constructed from it exceeds the amount of memory available on a single machine.” This sentence, while likely true, should be reformulated to clarify the SRA has not yet been indexed as a whole.
9. Regarding the sentence “storing the MetaGraph indexes even in their most performant representation”: what is meant by “most performant” here?
10. The sentence “Processing the remaining 90%, as well as all future incoming data, can be easily solved via scaling through parallelisation” greatly downplays the practical difficulties of processing the remaining 90% and the adjective “easily” should be removed.
11. The technical sentence “[.] the alternative approach of chaining together unitigs to form longer contigs is a clear target for future work” is out of place in the main text due to its lack of context (contigs are de facto chains of unitigs).

Remarks in the Methods section:

12. Regarding the sentence: “traversal in contig mode extends a traversed path until no further outgoing edge is present or if all the next outgoing edges have already been traversed”, it appears from this definition that contigs follow arbitrary branching hence can contain chimeric sequences that are biologically unsound. Is this what the authors meant?
13. Regarding: “Depending on the structure of the query data, this algorithm achieves a 10- to 100-fold speedup” over which baseline?
14. Regarding: “First, seeds corresponding to the forward orientation of the query are found according to the algorithm

described above.” Which algorithm? immediately above was the extension algorithm.

15. Regarding: “when input sequences are losslessly encoded in a MetaGraph index”: could you expand on this? It is also mentioned in the caption of Table 1, but the capability and trade-offs of MetaGraph to losslessly encode sequences by storing genome coordinates is not much developed in the manuscript.

16. Regarding: “used a constant cleaning threshold of 2 during graph cleaning to remove all unitigs with a median k-mer abundance of 1” even when the unitig is not a tip nor isolated? Also, what is “singleton filtering”?

17. Regarding SRA-Metazoa: if taxid 33208 was used to select samples, how were human samples excluded?

18. Missing section number in “in supplemental resources given in Section)”.

19. Supplementary material, “indexing costs” section: the projection implicitly assumes linearity in computing costs w.r.t input accession size. Is this the case? I would assume some datasets (metagenomes, low-coverage isolates) to be significantly more challenging to index. Please also add that the projected costs are lower bounds that do not take into account several cloud computing overheads: scratch storage costs, latency of the batch system, data download and input file conversion.

Code:

20. I could compile and test the code locally from sources with only minor issues. The compilation process requires an Internet connection as the Makefile downloads components.

21. I could not construct an index with counts: “To construct a MetaGraph index with k-mer counts (Counting de Bruijn graph), construct a de Bruijn graph as usual (see Construct graph) and then add --count-kmers to the annotation command, e.g.:" this step did not work, I get the error “[warning] No k-mer counts found at

'./tests/data/transcripts_1000.fa.kmer_counts.gz'. Every input k-mer will have count 1.”. My command lines were:

“./metagraph build -v -p 4 -k 31 -o graph \$DATA” (to “construct a de Bruijn graph as usual”) then “./metagraph annotate -v -i graph.dbg --anno-filename --count-kmers -p 4 -o annotation \$DATA”.

22. I recommend to the authors that they include additional tutorials that use the test data provided in the repository (./tests/data/transcripts_1000.fa) instead of data that is not provided (e.g. SRR403017) as presented in the documentation: tutorials for construction of an index with k-mer counts, construction of an index with genomic coordinates, sequence-to-graph alignment.

23. The sequence search page is incomplete: https://metagraph.ethz.ch/static/docs/sequence_search.html#align-sequences-to-the-graph

Typos: In abstract: “Mpb”. In Extended Figure 4: “representas”. In supplementary material: “RowDiff;RowFlat;”

(Remarks on code availability)

Referee #3

(Remarks to the Author)

In this manuscript, Karasikov et al develop and apply a method to improve compression of DNA sequencing data. They create a compressed representation of a large subset (but not comprehensive collection) of the data available from the NCBI SRA. They also present approaches to perform cost-efficient querying the graph-based compressed data. Overall, the presented work is clear, the MetaGraph approach is useful, and it does provide a benchmarked advantage over existing approaches. Limitations of this work aren't as well described as they could be, and while this is certainly a modest advance in computing, it is unclear to me how this represents a major advance over the existing graph based lossless compression algorithms.

Undeniably, the rate at which data are being generated is rapid and accelerating, with the decreasing costs of sequencing. Thus, finding ways for lossless compression of the data is critical for sustainability. The approach they describe compresses data variably based on type (human RNAseq data, for example, is compressed very efficiently; metagenomic data is much less well compressed).

The authors note limitations of existing approaches for compression (such as VG) and also note the challenges of alignment approaches (e.g. BLAST). They then explore the use of k-mer based approaches, but indicate that these types of approaches, while efficient, are inadequately sensitive. They present MetaGraph as an approach to address current limitations in data size (through compression) and analysis (through a graph-based alignment approach).

Overall, the manuscript is clear and relatively well written. It is certainly written for a relatively specialized audience, and for me, felt like a manuscript one would read in Bioinformatics or perhaps PLoS Computational Biology. I think the work they present is of quite high importance, but given that Nature is a general journal with a very broad readership, the manuscript could benefit from slightly more general framing.

I appreciate the benchmarking of MetaGraph against available tools for compression, such as BIGSI, COBS, etc. On one hand, the 16-38 times smaller representation from MetaGraph is a definite improvement compared to the other approaches; on the other hand, I do wonder if improving compression by this amount is an advance sufficiently exciting to describe this approach as truly transformational. With the rate of data generation, a much more significant compression will be needed to make a substantial dent in the challenges that we face as a computational community. Furthermore, while some of the

compression algorithms as lossy (BIGSI and COBS), not all of them are. All that being said, this is an advance and is notable.

Fig 2. In terms of performance against a difficult task, like a human gut microbiome query, MetaGraph performs very similarly to Mantis (another lossless approach), but it seems that the construction of Mantis indexes could not be done on all of the subsets due to a timeout. This is only discussed in the figure legend – given the excellent performance of Mantis, I think it would be useful to discuss the findings and then the potential reasons for the timeout for Mantis in the main text.

Table 1. I think this is a very useful table. I wonder if the authors can contextualize the utility of their tools given the expected ‘increase’ in SRA deposits over the coming years. If there is some reference/resource that has made estimations, it might help bolster their argument that this is a highly useful tool. On one hand, those who are doing metagenomics might find this tool to be incremental in its utility. On the other hand, those who are doing scRNAseq might find great utility in this tool. To that end, given that quite a lot of scRNAseq data is now being generated, it would be valuable to see the utility of this tool in the context of that data type.

One thing that isn’t immediately obvious from the way the manuscript is written is how alignments will fare in the context of indels or SNPs (or even larger structural variations) in the context of graph alignment. These important issues should be addressed in the manuscript to maximize utility of the findings.

The BSJ analysis is quite interesting. It would be helpful to have a supplementary figure that shows a toy example of how this works.

The web-based GUI is intuitive and easy to use. Can it handle batch queries? As best as I could tell, it cannot. This makes me wonder what the context for its use would be, as the throughput is currently limited to one sequence at a time.

While it is undeniable that storing all available data on a commercial cloud would be costly, one of the major advantages of cloud computing is that such references can be centrally stored and accessed by hundreds, thousands, or even more groups all over the world. I would presume that eventually, most will not want or need create their own ‘local’ copies of such data in the cloud computing environment. The way this section of the manuscript is written doesn’t acknowledge this.

(Remarks on code availability)

Version 1:

Reviewer comments:

Referee #1

(Remarks to the Author)

I thank for the authors for the detailed point-by-point response to the issues I raised. Most of them appear to have been addressed in the most recent revision, making the manuscript more suitable as a version of record for the MetaGraph project (and associated development), which as I detailed in my previous review, is a substantial contribution to the literature.

First, I thank the authors for having changed the paper title. The work is impressive, but not over-selling is just as important.

Having said that, I do still worry that random subsetting to build extrapolations might fail in subtle ways. To pick one example in the authors' favor, compression ratios are likely to improve as more similar data is acquired. An example in the other direction though: indexes at a fixed k-mer size might get more crowded increasing query runtimes. My intuition is that the cost and scalability estimates are reasonable, and the authors do discuss some of these subtleties, but the simple linear scaling still feels suboptimal, as scaling is never as easy as it initially looks.

I would like to explicitly thank the authors for the updated benchmarking against new tools, as I know how much work that can be.

I do still think using the terminology of "lossless" is a bit confusing. I agree with the Authors that the MetaGraph approach is perhaps "less lossy" than other "lossy" competitors, and that it is unclear that researchers would ever even want a fully lossless index of unfiltered data--personally, I tend to come down on the side of saying that a fully lossless index of unfiltered data is unnecessary. However, as a matter of principle and to avoid confusion, I do think it needs to be made clear to a non-expert reader what exactly the work purports to do. This is largely a matter of semantics, and I do not think changing the terminology is by any means needed before publication, but I do exhort the authors to carefully consider the potential confusion this choice of terminology may cause a general audience.

Overall though, I think the manuscript is substantially improved since the last time I read it. My one final thought is that the manuscript should make even more explicit how this work benefits the scientific community in practice, given the general audience of Nature. Having said that, I believe the manuscript is likely suitable for publication.

(Remarks on code availability)

Referee #2

(Remarks to the Author)

The authors revised their manuscript presenting MetaGraph, with a slightly updated title that fixes the scope, and many new supplementary analyses supporting the reviewers' remarks. I still have remarks mainly on the software side, yet comparatively to the magnitude of the contribution, these remarks are minor. One additional point that I develop below is on the extrapolation to full SRA, that still appears to be uncertain.

Major remark:

0. I appreciate that the authors provided supplementary table S3, to give an overview of the 100 studies test for indexing the entire SRA, but I remain unconvinced that the approach scales. (What is "WMGS" in that table?) I will refine my remark by asking the authors to do a cost projection based on indexing the metagenomes alone (an increasing number of them), to see if the previous extrapolation still holds up.

Questions on new figure 5:

1. "despite our indexing approach being lossless" as noted by previous a reviewer comment, this is a misleading claim as
2. Fig 5c and also Fig 3, I do not understand what the metric "Sequence Identity Cutoff" represents, could you please explain how it is defined?

Software/docs:

3. In my previous review, on remark 21 I had issues running a provided example. I revisited it. It is worth mentioning that a few tools are further needed as compilation dependencies: `aclocal` from `automake`, `autoreconf` from `autoconf`, and `libdeflate`; but this is a minor point. I could compile the new version of the software.

4. For testing the beginning of quick start (https://metagraph.ethz.ch/static/docs/quick_start.html), please add information on which folder to run the tests (supposedly it is `tests/data`).

5. The quick start tutorial "Index k-mer counts" still needs fixing. The authors mention in their response to my remark 21 that: "It seems that the mentioned file `../tests/data/transcripts_1000.fa.kmer_counts.gz` was created by a previous command `metagraph build --count-kmers ...` + `metagraph transform --to-fasta [..]`" however these steps do not appear in the "Index k-mer counts" part of the tutorial, nor elsewhere. To be clear, I am trying to run the quick start tests starting from https://metagraph.ethz.ch/static/docs/quick_start.html#index-k-mer-counts, so I went back to the "Construct graph" section as instructed there.

6. In response to my remark 22, the authors write "we have instead updated the tutorial to include a download step for SRR403017" however I do not find this download step anywhere in https://metagraph.ethz.ch/static/docs/quick_start.html. Is it elsewhere?

7. My remark 23 either was not addressed: "The sequence search page is incomplete: https://metagraph.ethz.ch/static/docs/sequence_search.html#align-sequences-to-the-graph We thank the reviewer for bringing up this oversight. We have updated the documentation to include a description of sequence-to-graph alignment and its output format". I still see an empty section at that URL.

Typos in updated text: "Sequence Read Archive (DRA)" should be SRA.

(Remarks on code availability)

Referee #3

(Remarks to the Author)

Overall, the manuscript is improved, which I appreciate. I continue to wonder if this will be accessible to the 'general genomics researcher'; however, it certainly will appeal to the highly CS-focused portion of the genomics community, and does represent an important step forward. Assuming MetaGraph is maintained and remains competitive (in terms of performance and usability) compared to other tools that can perform similar tasks, it has the potential to be a useful tool for the genomics community in the years to come.

1. It would have been nice to have a version of the manuscript with changes highlighted.

2. Some limitations are pointed out - it might be easier for the reader to follow if there is a dedicated limitations paragraph.

(Remarks on code availability)

Version 2:

Reviewer comments:

Referee #1

(Remarks to the Author)

The authors have addressed all of the issues I raised previously, and have improved the discussion. I especially appreciate the additional indexing of further samples, which does indeed demonstrate that the MetaGraph approach's scaling behavior.

I thank the authors for their hard work. I consider this manuscript suitable for publication.

(Remarks on code availability)

(I have previously the software, but did not do a deep dive this round, because the software/code was already good quality in a previous round of reviews)

Referee #2

(Remarks to the Author)

The authors revised their manuscript presenting MetaGraph, changing the biological sequence data index, which is the central focus of the paper, from an index of "SRA sequence data" to an index of pre-processed contig data, "Logan contigs".

The revised manuscript's methods and claims regarding MetaGraph's scalability depend on Logan contigs, therefore several methods and novelty claims throughout the manuscript are now overstated. As it stands, MetaGraph is not shown to be sufficient for pan-SRA index scalability without a major external pre-processing dependency. This is not a critical flaw to the method, but the manuscript should provide a fair and complete description of the essential prior work in the Logan project, which it now depends upon. At the Logan index snapshot (December 2023), the SRA contained 50 petabases of sequence data, the Logan contigs are a compressed representation of that data of only 2 petabases which are now being indexed and presented as MetaGraph's index of the SRA sequence data.

The Logan contigs are publicly available since July 2024, and the index of all Logan contigs, Logan Search was made publicly available alongside that data since November 2024, albeit neither is yet published. While the Logan contig dependency is cited, the Logan Search index released alongside it was not mentioned (<https://logan-search.org/>). In my view the manuscript should more fairly represent the state of the art and add a qualitative description and URL reference to Logan Search, as to provide complete attribution to prior publicly available work, Logan contigs and Logan Search.

The new experiments asked by the reviewers also have clarified the scalability claims. Small queries which were previously projected to cost 14 USD, in the new revision are projected to cost 100 USD, an order of magnitude higher, showing the limits of previous extrapolations from smaller sets.

Recommendation: Accept after minor revision.

signed,

Rayan Chikhi

Competing interests: I am a co-developer of Logan.

Major points:

1. Novelty claims vs. Logan Search: Please (i) cite Logan Search (<https://logan-search.org/>) alongside Logan contigs wherever feasibility is discussed, and (ii) adjust wording in Abstract, Introduction and Discussion that currently frames MetaGraph as the first or only demonstration of full-archive indexing and search:

- a. in the Abstract: "We demonstrate the feasibility of indexing the full extent of all public biological sequences". The authors' claim of first demonstration is overstated, since Logan Search has been publicly available for months.
- b. in the Introduction "[...] to demonstrate the practical feasibility of full-text indexing all known biological sequences" (similar)
- c. "However, the raw sequencing data itself remain inaccessible for full-text search" It is false now that Logan Search exists.
- d. "[...] to demonstrate that indexing repositories such as public sequence archives as a whole is not only theoretically possible, but practically feasible": It has already been shown to be practically feasible, at a greater volume than achieved in the present manuscript.
- e. in the Discussion: "As we set out to demonstrate the practical feasibility of indexing entire sequence archives" (similar)
- f. "Notably, this is an analysis that would otherwise have required access to hundreds of terabytes of raw sequencing data." Using Logan contigs or unitigs would also enable this analysis, with a volume of data 10-100x smaller.

g. in the Discussion: “a crucial milestone in computational modelling, solving the problem of making all existing biological sequence data searchable and easily accessible” Regardless of Logan Search, this claim needs to be toned down, as per previous reviewer remarks.

h. in the Discussion: “For the first time, it is now feasible to efficiently search” needs to be amended.

i. In the Discussion: “Processing the remaining 90%, as well as all future incoming data, can be achieved via scaling through parallelization and should be quite feasible for institutions such as EBI or NCBI” It was achieved already in Logan Search for SRA data until 2023.

2. Cost projections: The projected per-query cost rose from USD 14 to ~USD 100 (Figure 5a). Please clarify the source of this revision and its implications for scalability.

3. Index construction strategy: The manuscript moved from monolithic per-clade indexes to > 1300 per-sample indexes for metagenomes (public folder s3://metagraph/all_sra/metagenomes/). In principle this should affect the linear extrapolations in Figure 3, given that lower compression is achieved when using subindexes instead of a single large one. Could the authors explain this?

4. Losslessness wording: The text still states the indexing approach is “lossless”; since noisy k-mers are dropped during cleaning this is inaccurate. Please rephrase.

5. Throughout the manuscript, make the distinction between “SRA Sequence Data” and “Logan Contigs” explicit, as these are distinct datasets.

Minor points:

1. The sentence “All indexes constructed from public-access data are available in the cloud under s3://metagraph” needs to instead point to the Github documentation repository explaining what indexes this s3 contains.

2. “In total, we have processed almost 5 petabases [..]” in the Discussion. In the abstract, it is written that 16 petabases were processed.

(Remarks on code availability)

Referee #3

(Remarks to the Author)

I thank the authors for their revised manuscript and attention to the reviewer comments. I have no further comments and feel that the manuscript is appropriate for publication.

(Remarks on code availability)

Version 3:

Reviewer comments:

Referee #2

(Remarks to the Author)

The revised manuscript is now substantially improved, I have no further comments, and recommend acceptance.

(Remarks on code availability)

Response to Reviewer Comments

Introduction

We would like to thank all reviewers for their time and effort spent on reviewing our manuscript. The provided criticism has been very constructive and has inspired several new experiments and data figures. We are confident that the changes we implemented have significantly improved the manuscript and clarified the advantages of MetaGraph for large-scale sequence indexing.

We have structured our response to the reviews as follows. First, we address two general points that have been brought up by several reviewers. After that, we provide a point-by-point response to all remarks made by the individual reviewers. The reviewers' text is provided in black, while all our responses are shown in blue italic.

Responses to general comments

C1: Scope of the manuscript/title and data snapshots

We appreciate the reviewers' concern regarding the suggestiveness of our manuscript's title. In the following, we would like to outline the motivation behind our choice.

With this work, we aimed to demonstrate the feasibility of creating a sensitive index for the entire SRA—one that can not only be deployed centrally and support rapid full-text searches, but also exhibit a certain degree of portability. To achieve this, we developed the MetaGraph framework, which we describe in detail in our manuscript.

From the outset, we recognized that generating and maintaining a full SRA index would likely exceed the capabilities of a single research laboratory, especially given the repository's continual super-linear growth. Such an undertaking should be left to internationally supported institutions. Our goal, therefore, was not to construct a complete index ourselves, but rather to establish scalability at a technical level—both in index construction and querying—and to benchmark our methods against the current state of the art.

To further validate feasibility, we selected a recent (June 2024) representative random subset of SRA samples and extrapolated the costs associated with different indexing scenarios based on the current metadata (January 2025). Our analysis indicates that, compared to other large-scale scientific efforts and relative to the original cost of producing the underlying data, the investment required to build a full SRA index using MetaGraph is very moderate. We also show that the search over the whole set of publicly available sequences is within reach for performing research on this invaluable resource.

Finally, we sought to demonstrate MetaGraph's capacity to process petabytes of input data by creating a proof-of-concept index set. We processed millions of SRA samples, comprising more than 4.8 Petabytes of input. We made these indexes publicly available in the cloud as a resource for the scientific community, thereby showcasing the portability of our approach—an attribute not provided by competing indexing strategies. We have continued creating indexes and will continue uploading them once additional parts are completed.

In summary, our approach represents a significant step forward in making all public sequencing data fully searchable, thereby opening new avenues for large-scale biological data analysis. However, to address the concerns on the wording of our title and to better focus our scope, we have renamed the manuscript to “Towards Indexing All Life’s Known Biological Sequences”. In addition, we are in the process of indexing a wider variety of RNA-Seq samples to better demonstrate MetaGraph’s applicability for large-scale transcriptomics.

C2: Comparisons to more recent tools

We thank the reviewers for their suggestions to extend our evaluation to additional tools. We have extended the scalability evaluation in the manuscript to also compare to Themisto, kmindex, Fulgor, and meta-diff Fulgor. We would like to note that, despite new algorithmic developments, MetaGraph is still the only framework that demonstrates scalability to indexes on petabase-scale inputs while preserving the capability to sensitively align query sequences and losslessly represent the input k-mer sets. Our evaluation demonstrates that, after selecting appropriate parameters and matching their values across all tools, MetaGraph continues to provide the smallest index and very competitive index load + query times for exact matches and can perform sensitive sequence-to-graph alignment with an overhead of a small constant factor. For this evaluation, we have removed BIGSI since its authors have deprecated it in favour of COBS and have asked users to switch to the latter tool.

Referee #1 (Remarks to the Author):

Comments for Author

In the manuscript “Indexing All Life’s Known Biological Sequences” by Karasikov et al., the authors make a valiant effort to broaden the **searchable** set of biological sequences. The splashiest contribution of this article is the indexing of “all life’s known biological sequences” for search; they make petabases available for searching over and provide algorithms/webservices/software for doing so.

The second contribution are the algorithms for indexing and searching, which is in my mind technically much more challenging and also just as important of a contribution. This manuscript appears to be culmination of multiple prior efforts, some of which were previously published whereas others were only preprinted by not published.

- The MetaGraph algorithm was preprinted in 2020
- Alignment was improved upon: MetaGraph-MLA by Mustafa et al., 2022 bioRxiv and TCG Aligner by Karasikov et al, 2022 Genome Research)
- Better color compression (Karasikov et al., 2020, Journal of Computational Biology)

The engineering + algorithms are extremely impressive, and to me the main highlight of the paper, though because of the complicated prior history, it’s a little hard to discern which algorithmic contributions should be considered novel to this manuscript.

We appreciate the reviewer’s concern in this regard. The novelty in our manuscript lies mainly in the composition of these technical contributions into a software framework, the design of a scalable workflow for large-scale sequence indexing, the cost estimate of indexing the entirety of the SRA, and the demonstration of how such a resource can be used to derive new scientific insights. In addition, we provide all indexes generated on public data as a community resource. With this, we demonstrate the feasibility of indexing the SRA and the portability of the indexes. Our prior publications focussed on specific algorithmic developments but did not attempt large-scale analyses of all publicly available biological sequences.

Overall, this work is a very strong methods paper that has the potential to substantially move the field forward. However, there are many non-trivial caveats, especially with respect to presentation, experiments, benchmarking (or lack thereof). In its current form, I don't think this paper is suitable for Nature, but I could certainly see a revision making that cut.

Specific major points:

Point 1: The paper's title is far too suggestive. While the author's do not explicitly claim this work indexes ****all of life's known sequences****, the title strongly suggests just that. A claim that strong needs to be substantiated.

We thank the reviewer for bringing up this point. Please refer to our response C1. To more precisely frame the scope and message of our manuscript, we have renamed it to "Towards Indexing All Life's Known Biological Sequences". Moreover, we'd like to note that computations are running to eventually make this index much more complete over time.

To pick just one example, metagenomes are vastly underindexed. Given that microbes make up a huge diversity of life and that most have not/cannot be cultured, this is an important issue. The authors index SRA-MetaGut (all human gut metagenomes) and MetaSub (4000+ urban environmental metagenomes) as the two sets of metagenomes fully indexed. This excludes non-gut, non-human, host-associated microbiomes as well as terrestrial microbiomes such as soil and ocean (although TARA *assemblies* are indexed).

We acknowledge the assessment of the reviewer that losslessly indexing all metagenome samples of SRA within a joint index is a formidable task. To demonstrate feasibility, we have selected two metagenome subsets and provide the according indexes. In addition, our random subset of SRA, that forms the base of our cost extrapolations, contains metagenome samples at the expected fraction (20%). Thus, our cost and scalability estimates contain the metagenome part of SRA. In addition, we have prioritized indexing of metagenomes in the ongoing indexing efforts and a growing fraction of metagenomes will be indexed and available soon.

On line 153, I note that the authors collected their SRA dataset in April 2020, which is four years old at this point. Another recent paper Pebblescout (Nature Methods 2024) indexes all SRA metagenomes (and all SRA RNA-Seq experiments) up to ~2022, arguably more a comprehensive database than what is presented in this paper. My sense is that Pebblescout's indices are much less comprehensive and impressive than MetaGraph, but given the emphasis on exponential growth in sequencing capacity in the introduction, a gap of 4 years to the present seems like sufficient time for the dataset sizes to have more than doubled.

We appreciate the reviewer's feedback on the comprehensiveness of our processed dataset. For this first point, we would like to refer the reviewer to our response C1. Specifically for PebbleScout, we would like to add that the approach is conceptually quite different from the one we use in the MetaGraph framework. As already outlined in the PebbleScout paper (Shiryev & Agarwala, 2024; Table 2), the PebbleScout approach generates orders of magnitudes more false positive hits than MetaGraph. In addition, due to its design and k-mer centric approach, the PebbleScout method is not able to perform alignments as is possible with MetaGraph. That said, we acknowledge that the presented work by Shiryev and Agarwala is an important milestone and indexes a larger number of RNA-Seq samples. We have added the appropriate reference to our text and highlight the difference to MetaGraph. To address the relative underrepresentation of RNA-Seq data sets in our index set, we have started the process of indexing additional RNA-Seq samples for human and mouse and will continue with the remaining available samples.

Regarding the mentioned exponential growth of the data, we would like to point out that, while over the past years the global sequencing capacity has retained its exponential growth, the actual amount of sequences uploaded yearly to the SRA has shown slower growth since 2020 (cf. Supplementary Figure S-13). Thus, we think that our proposed approach will be able to keep up with

the growth of SRA for the foreseeable future. Our extrapolations use a representative random subset of all SRA samples, drawn from the complete metadata table in June 2024. The cloud cost estimates for creating the indexes for data from 2025 and providing search/alignment functionality are given in Supplementary Table S-4.

Point 2: Many of the figures (3C, 4A-C) focus on biological discovery, but these show proof-of-concept results rather than significant biological discoveries that are substantiated with follow-up. For example, 3C suggests that one can use MetaGraph to do large-scale querying of transcript abundances, but no benchmarking of this functionality is shown. This would be OK for a methods-based paper (e.g. in Nature Methods), but seems a bit limited for a Nature paper (though I defer to the Editor on whether this is actually an issue).

We thank the reviewer for this comment. Our vision for MetaGraph, as laid out in our manuscript, is similar to that of a search engine like Google that researchers can use for their own research (e.g., in the cloud using the provided tools). Towards this goal, we address the technical challenges of this endeavour, and in the process, demonstrate that MetaGraph can not only be used for large-scale biological analysis, but also that the indexing itself can be used to derive novel insights from sequence data.

For example, Figure 1 (and its corresponding Table 1) and Figure 3a are studies of the diversities of the indexed sequence sets, measuring the inter- and intra-sample similarities of the sequencing sets, and the distinguishability of different samples within an index based on read re-alignability.

The presented transcriptome analysis is to showcase the ability to preserve count information within the indexes while maintaining a very high compression rate. We use two large available cohorts (TCGA and GTEx) as a proof of concept. We have extended our analysis on the back-splice junctions to now contain a comparison of both cohorts (Figure 4c) and the overlap to experimentally validated data (Figure 4d). In addition, we discuss the tissue-specific signal of back-splice junctions and subset the presented data to the COSMIC Cancer Census gene set (Extended Figure 4f) to demonstrate how a large number of synthetic sequence features can be assessed comprehensively in the indexed data in an unbiased manner without requiring re-analysis of the full raw data. We have revised the discussion section to better highlight the insights drawn from these experiments.

Point 3: As a follow-up to point 2, given that the most significant contributions are methods-based, there is a lack of comprehensive benchmarking, especially of recent methods, many of which themselves benchmark against the MetaGraph preprint.

We thank the reviewer for raising this point. We would like to note that, in addition to the technical contributions, one of the main points of our manuscript is to demonstrate the feasibility of indexing a full public sequencing archive, which includes cost estimates for full-text search over all publicly available sequences and showing that the raw data is very compressible. For further details, please refer also to our response C2. We have expanded our benchmarking in the current version of the manuscript (see comment below).

Indexing:

- Fig 2A – Some newer methods seem to be omitted from benchmarking, including kmindex (Lemane et al, 2024), Thjemisto (Alanko et al, 2023). Both methods claim better performance than MetaGraph in their papers. GGCAT is cited, but not compared against, either.

We thank the reviewer for bringing up this concern that has also been echoed in another review. For a general comment on this matter, please refer to our response C2. In particular, we have extended our benchmarks to also profile kmindex, Themisto, and different variants of Fulgor. Please see the updated Figure 2. The main conclusions remain the same.

- Index building time is very relevant in practice. I do not see benchmarks for index building time, nor for continuous updating. The usefulness of a webserver like BLAST is partially in that it is kept

up-to-date. How long would it take to rebuild (or update) the indexes on say a monthly basis? Would the index have to be built from scratch, or could it be delta'd?

We thank the reviewer for mentioning this important point. With the 100-study SRA-Public data set, we showed that a high compression ratio can still be achieved with a random selection of sequencing studies. This motivates a database growth strategy where updates are made by constructing new indexes to cover accessions that were added after the last update. Additionally, a less frequent database rebuild to consolidate existing samples into a smaller number of indexes would improve compressibility. We have updated the Discussion section to describe this growth strategy. Based on the number of sequences added yearly over the past 5 years (cf. Supplementary Figure 12), this strategy appears realistic and feasible for the foreseeable future.

Graph cleaning:

- Graph cleaning creates a **lossy** index, not a lossless index as claimed. This is an important point that needs to be cleaned up. The lossiness of MetaGraph is remarked upon by some of the competitor more recent papers, such as in the Pebblescout manuscript (Nature Methods 2024). Yes, it is technically true that MetaGraph provides a lossless representation of the **indexed** k-mer sets, but the data cleaning prior to choosing k-mers to index makes the overall process lossy.

We thank the reviewer for their comments and concerns regarding the nature of our indexing strategy. It is indeed true that, while MetaGraph provides a lossless index of an annotated k-mer set, our indexes constructed from raw sequencing data sets do not provide lossless representations of the inputs (our indexes of assembled sequences, such as reference genomes and MAGs, on the other hand, are fully lossless). Most existing indexing strategies (except PebbleScout) use some form of input preprocessing in order to reduce sequencing noise from the data. Even PebbleScout uses minimizer indexing (selecting only one representative 25-mer in a local context of 42), thus reducing the entities that are being indexed and creating possible false-positive matches. We do note, however, that unlike lossy representation schemes, a match that is found when searching a MetaGraph index is indeed a true match. The lossy approaches also need the noise reduction by preprocessing but introduce an additional kind of “noise” to the search results (namely false positives).

Lastly, we would like to point out that the decision to filter raw inputs is not due to a technical limitation of MetaGraph but an active choice to optimize the cost/benefit ratio when indexing large data sets. For instance, building the index on the MetaGut data set without filtering compared to singleton-filtering would increase its size by a factor of approx. 4.5X, but would only provide marginal improvements of overall matching sensitivity (see new Supplemental Figure S-8). Interestingly, PebbleScout indexes are approx. 5-10X larger compared to MetaGraph indexes (Table 1 of the PebbleScout paper - <https://doi.org/10.1038/s41592-024-02280-z>)

Even though it is technically possible to omit the cleaning procedure in MetaGraph while indexing, determining the utility of an index of completely unfiltered data is still an active area of research. There has been particular interest in assessing the impact of contamination in reference databases, and such work can also be extended to indexes of unassembled sequence data.

- Furthermore, I would like to see benchmarks showing how graph cleaning contributes to the efficiency. Is the graph cleaning a major reason for MetaGraph's efficiency?

The achieved efficiency of our indexing approach comes from two factors. First, the data pre-processing that includes error correction (cleaning) and k-mer deduplication significantly reduced the actual amount of data that needs to be indexed without substantial signal loss (measured in the ability to realign 80 to 90% of the input reads, see Figure 3). This data reduction is measured quantitatively and is shown in column Data redundancy ('Redund.') in Table 1. Second, the cleaned data is indexed with efficient data structures that achieve a high compression ratio, while our search algorithms ensure that we maintain fast queries. This efficiency is quantitatively shown in column 'Efficiency' in Table 1. In our scaling experiments (Figure 2), when run on identical input data, MetaGraph performs orders of magnitude better than several of the comparison methods. This improvement is due to technical contributions such as an efficient implementation of a succinct k-mer

dictionary and the compressed representation of graph annotations. For the specific case of cleaning metagenomics samples, please see our comment below regarding our additional experiment benchmarking our cleaning strategy.

- Graph cleaning seems to be relatively aggressive (≤ 3 -coverage k-mers are pruned, but possibly more, according to Methods). This will remove a lot of low-abundance organisms' reads in Metagenomes, or lowly abundant transcripts. This point needs to be addressed. Fig. 3A does not adequately address this because it calculates the overall recall, but the recall of low abundance transcripts/organisms is interesting by itself. This is especially important for more careful interrogations of low-frequency bacterial species in a metagenome, rare isoforms in RNA-seq abundance quantification, or structural variations supported by a single high-quality long-read in say cancer biology.

We thank the reviewer for raising this concern. For the metagenome indexes, our cleaning strategies were adapted over time and are different for each. We would like to point out that in our cleaning procedure, the threshold is applied to the median abundance of k-mers within each unitig in the graph instead of filtering all k-mers independently, hence, k-mers with very low abundance may still be admitted. For the SRA-MetaGut metagenomic read index we imposed a universal per-unitig median count threshold of 2 (i.e., keep all unitigs where at least half of the constituent k-mers occur more than once).

To benchmark our cleaning strategy more generally, we have included an additional experiment on SRA-MetaGut where we have compared our cleaning strategy to a simpler strategy of removing singleton k-mers from each sample (i.e., discard all k-mers that occur only once). For each index, we then align a random subset of reads from ten random MetaGut samples to each index (the same reads used in the recall evaluation). After using Kraken to determine species abundances and classify each read, we measure read label recall at different abundance levels. We find that our cleaning strategy recovers the correct label for 85%-99% of reads at different abundance levels ranging from 0.0001% to 10% (see Supplementary Figure S-8). These results are comparable to singleton filtering, with no major differences at any abundance level. Again, there is no technical limitation in MetaGraph to index a given dataset without filtering, but the resulting indexes will be a small constant factor larger, which can increase hosting cost.

For MetaSUB, we re-evaluated our cleaning strategy and found that the one we used for MetaGut is superior to the one we previously applied (see Supplementary Figure S-15). We are now in the process of re-indexing the MetaSUB data using that same filtering strategy.

Alignment/sequence search:

- More precision/sensitivity tests for alignment are likely needed. There is not test of precision that I am aware of. Figure 3A shows how the recall varies for reads mapping ****to its own sample****, but there should probably be a test of how reads map to the combined metagraph index. Some notion of "false positive" should be used as a benchmarking metric. This is especially important since, unlike a BLAST search, k-mer matching does not give as obvious metrics for evaluating alignment uncertainty or goodness.

We thank the reviewer for this suggestion. We have included additional evaluations to study the prevalence of random matches to our indexes. After generating random nucleotide sequences with the same length distribution as our query reads, we mapped them to each graph via exact k-mer matching and alignment. For sequence identity cut-offs down to 32 bp, we observe no random matches (see Figure 5c). We have also added an evaluation comparing the random match rates to kindex (see Supplementary Figure S-14).

Another possibility for measuring alignment uncertainty would be the estimation of expect-values (E values) for alignments. Although beyond the scope of this work, an interesting avenue for future work would be to generalise Karlin-Altschul statistics for pangenome graphs to enable these calculations for local and semi-global alignments (as partially explored by Schultz, et al. 2021. PMID: 33532821).

- On the same note, Extended Data Fig. 2 shows read mapping to an e-coli with MetaGraph's alignment algorithms. E. coli genomes are very simple compared to a complex eukaryotic genome. Could this experiment be repeated with Human or Wheat genomes?

We thank the reviewer for their suggestion. We have replaced this evaluation with one done using all assemblies of Human chromosome 21 in GenBank as of Oct. 21, 2024. In this new experiment, we also found that sequence-to-graph alignment offers substantially better approximations of alignment scores for long reads than exact k-mer matching, with MetaGraph's alignment algorithm performing significantly better than the state-of-the-art GraphAligner tool (cf. Extended Data Figure 2).

- Additionally, although it is impressive to have a single software that indexes close to "all" biological sequence, nucleotide sequence search and protein sequence search do actually often have substantial differences. The k-mer based exact match alignment for nucleotide sequences works less well for protein sequences, where identity between similar protein sequences may be lower. Could the authors provide a benchmark of this?

We thank the reviewer for this comment. To begin, we would like to acknowledge that, given a fixed evolutionary distance, one would expect an amino acid sequence to be better conserved than its corresponding DNA sequence due to the redundancy of the codon table in the translation process. However, we acknowledge that the contexts in which amino acid searches are performed more often involve searches for remote homology, in which a lower similarity is acceptable or desired.

We would also like to bring attention to the fact that many amino acid alignment techniques (including BLAST tools) like MetaGraph typically use a k-mer index in the seeding step of a seed-and-extend alignment paradigm. Our focus in this work was on indexing scalability, and thus, our primary focus was on indexing the much vaster amounts of DNA sequencing data, and consequently, our alignment algorithm was optimised primarily for nucleotide sequence alignments. Although we do provide a BLOSUM62 matrix as the default scoring matrix for alignments on amino acid graphs and a suite of unit tests for these alignments to guarantee correctness, our alignment algorithm has not been designed to compete with state-of-the-art tools such as DIAMOND. We leave refining our algorithm for this specific context as future work.

Specific minor points:

The abstract claims that the compressed representation fits on a single consumer hard drive (~100 USD), without specifying exactly which databases fit. Given the setup, the implication is that "all" sequences in this compressed format will fit on a \$100 drive. At present, the highest capacity HDD I can find for around \$100 only stores 12 TB of data. (<https://diskprices.com/>) However, in Table 1, the authors themselves say that SRA-Public to 01.01.2023 would take 130 TB to store the index. Even excluding that, the sum of all the other indices in the table seems to be just above 12 TB. Please clarify.

We thank the reviewer for this comment. The statement we made in the abstract referred to the size of the SRA fraction we actually indexed. We have revised the abstract and generalized the statement to explicitly refer to an all-SRA index that may use several consumer hard drives.

Although it is common folklore in the field that sequencing data has been growing exponentially (and it certainly was for a while), this appears to have leveled off in recent years. The authors cite a 2015 paper to substantiate their exponential claim. But, in the ENA statistics the authors cite, the run doubling time has been **increasing** over the last decade, from a doubling time of around 10-15 months back in 2012, to nearly 45 months currently. Were growth actually exponential, the run doubling time would be constant over time. Thus, although sequencing data still grows quickly, the exponential claim appears to be outdated.

We thank the reviewer for this comment. We agree that the growth of data deposited in publicly-funded databases such as the SRA has been decreasing in the last few years. At the same time, the drop in costs for sequencing continues to be exponential according to the National Human Genome Research Institute, indicating an exponential growth in sequencing capacity. Much of this sequencing is done for non-research or private use, such as for industry or medical applications. We have included an additional citation to support this statement. Also, we have added a new Supplemental Figure (S-14) to show the difference between public and non-public data available via SRA.

Line 49: the authors have made a distinction between (i) sketching techniques and (ii) methods employing Bloom filters. Taxonomies are hard in any field, but certainly many computer scientists would include Bloom filters as a type of early probabilistic sketch in this day and age. The authors choice of taxonomy is justifiable, but certainly should be made explicit.

We thank the reviewer for this comment. We have expanded the descriptions to more clearly differentiate (i) and (ii), distinguishing sketching techniques intended for distance estimation and those intended for approximate membership queries:

“(i) methods based on sketching techniques for approximate set similarity and containment queries, which produce small hash-based summaries of the input data and then use these summaries (i.e., sketches) to estimate distances between query sequence sets and a target; (ii) methods employing Bloom filter-based data structures to allow for approximate membership queries of individual k-mers”

Line 137: saying that 4.8 petabases = 2.5 petabytes is a bit confusing without reading the supplement on how those numbers are counted. Not sure what if anything can be done here to clarify things, but it was a bit confusing on first reading.

Thank you for pointing out this oversight. We have clarified in the manuscript that the 2.5 petabyte figure refers to the total size of the gzipped FASTQ files used as input.

Line 285: “with higher error rates, such as PacBio’s SMRT” PacBio HiFi reads have a lower error than short reads nowadays

This originally referred to PacBio CLR (whose use has been declining recently). We have revised this sentence to solely refer to ONT reads.

Line 317: How could MetaGraph “facilitate large-scale learning tasks” and train LLMs?

We have expanded this section of the Discussion to provide more insight into the utility of MetaGraph indexes for LLM training: “For example, MetaGraph indexes can act as a database for efficiently generating sequences for model training, including the generation of previously observed and de novo sequence recombinations.”

Line 321: the authors have not demonstrated how sensitive MetaGraph sequence alignment can be before failing. I would like to know when it fails. Fig. 3A shows 2% mutation rate, but higher mutation rates should be tested. BLAST works with quite dissimilar query sequences, for example, and although MetaGraph should not be expected to be as sensitive, users should know when it fails.

We thank the reviewer for this comment. We have added Supplementary Figure S-5 extending this evaluation to also include mutation rates of 5% and 10%. We observe a wider gap between the average recall of our sequence-to-graph alignment and exact k-mer matching queries, noting that the average recall for sequence-to-graph alignment drops to 0.18–0.3 at a mutation rate of 10%.

Line 479: how is the gamma-distributed mean relevant

We have clarified this section. Since a Poisson distribution with a gamma-distributed mean is essentially a negative binomial distribution (with the added potential for the dispersion parameter to be

real-valued instead of integer), we have simplified the text to say that we fit a negative binomial. We have also revised the subsection to provide more details on the histogram fitting.

Line 780: what default value of seeds is used for alignment, $< k$?

We have mentioned in the text that the default seed length for alignment is 19.

Micro points:

- Line 56: extra space before reference “alignment 34”.
- Line 60: this sentence is a bit awkward. Consider rewording.
- Line 411: “one from 4”. Reference should be expanded here.
- Line 1025: incorrect smart quotes. In latex need `and`.
- Reference 48 is a duplicate of reference 39.
- Reference 61 does not properly capitalize “rna”
- Table 1: is “MGS” defined anywhere?

We thank the reviewer for these suggestions, we have addressed these points, and in the process, corrected a number of other issues, such as updating outdated and removing duplicate references.

We have revised line 60 to now read: “A major challenge faced by all discussed methods is finding a desirable balance between efficient operation on petabase-scale sequence collections and support for a versatile array of query operations, such as exact k-mer lookup, sequence-to-graph alignment, and experiment discovery.”

Software comments

Overall, the software is extremely high-quality and well packaged for a bioinformatics software. There seems to have been dedicated software engineers for the software, which is extremely impressive. I had no problem running and installing the software. Documentation is well done. The web search is an impressive feat of engineering. It is a bit rough, but I think it has tremendous potential.

Web search suite

The web interface is decent looking, but quite simple. I think it could benefit from a few tune-ups. In general, I think more documentation on the website itself would be good.

- Alignment to graph option – this is not very interpretable for a general user. When should they use alignment vs normal k-mer querying?

- Performance is weird. Sometimes it's fast, sometimes it's quite slow and hangs.

- There should be an option to show only the best “X” number of hits. My browser ran out of memory.

- I think the authors could look at the Pebblescout (NCBI service) interface for inspiration on documentation, which I found a bit easier to use.

Error in the “Metazoa” index: (see attached doc)

The visualization of De Bruijn graphs seems to have an error (see attached doc):

We thank the reviewer for pointing this out. Our De Bruijn graph visualizer was previously maintained for private use, was not officially supported or documented, and has since been deprecated. We removed its link from our website during the review process. However, for users that still stumble upon this functionality, we made sure that it now properly works.

Software online documentation:

- Documentation is very good where present, but could be a bit more complete. Docs on sequence-to-graph alignment, which seems like an important aspect, is missing.

We thank the reviewer for pointing out this oversight. We have extended the documentation to provide a description of sequence-to-graph alignment, its features, and its output format.

- The quick start page is a bit overwhelming. Perhaps an (interactive) flow chart would help.

We thank the reviewer for bringing this up. Further improving the user experience is a constant goal for us. We will keep working on simplifying the workflow and providing more supportive material to allow users to create their own indexes.

Referee #2 (Remarks to the Author):

Comments for the author

The exponentially increasing, publicly available sequencing data hosted at the SRA is an invaluable resource for researchers. However current methods for accessing and searching this data are limited, often requiring excessive time and computational resources. Karasikov/Mustafa and colleagues address this challenge with their innovative MetaGraph framework. They leverage advanced text data structures to create highly compressed and searchable indices of sequencing data.

The core of MetaGraph is an annotated de Bruijn graph, which represents large sets of reads efficiently through k-mers with custom associated data (dataset identifier, abundance, or genomic coordinates). The framework supports aligning query sequences to the graph, allowing higher-sensitivity alignments hence increasing the utility beyond providing a simple k-mer index. A web server for querying some of the indices is also provided.

The manuscript shows applications of MetaGraph to three use cases: association of AMR genes to bacteriophages, tracking AMR in human gut metagenomes across time, and searching back-splice-junctions in GTex data.

From a computational perspective, the achievement is impressive. To the best of my knowledge, apart from NCBI PebbleScout which appeared years later and compresses data less well, no other combination of algorithmic methods and software architecture has been applied to indexing petabase-scale sequence data.

We thank the reviewer for the positive assessment of our work.

The main text is overall well-written and the Methods section is easy to follow (for a specialist).

Remarks in main text:

1. Title: I am unsure what is the stance of Nature for aspirational titles. 8.9% of the SRA (as of early 2024) has been indexed by MetaGraph (offline, in Table 1) and 1.6% of the SRA is searchable in MetaGraph Online. This contrasts with “all Life’s known biological sequences” that implies numbers close to 100%.

We thank the reviewer for bringing up this possible confusion. We have addressed it in our response C1. To more precisely frame the scope and message of our manuscript, we have renamed it to “Towards Indexing All Life’s Known Biological Sequences”.

2. PebbleScout, an online web server which has indexed 7.3% of the SRA (all the metagenomes) and was published recently in Nature Methods, is absent from the manuscript yet should be included and its features/performance compared against MetaGraph.

We thank the reviewer for bringing up this oversight. As mentioned in our response to Reviewer #1, we have referenced PebbleScout in the current version of the manuscript. We noted to the reviewer that a comparison to MetaGraph is already present in the PebbleScout publication, showcasing their greater false positive rate compared to MetaGraph.

3. Several other indexing schemes have since appeared, e.g. Themisto, Fulgor and kmindex. Even though the original 2020 MetaGraph article predates these methods, they should be benchmarked in this manuscript.

Please refer to our response C2. We have updated our benchmarks to include Themisto, kmindex, and different variants of Fulgor.

4. Regarding the sentence “Our indexes are freely available to the research community” in the Abstract, some are missing from the Indexes pages at <https://metagraph.ethz.ch/>: Uniparc, SRA-Metazoa mouse and SRA-Metazoa w/o human.

We thank the reviewer for bringing up this oversight. We have uploaded all published graphs for download (except TCGA and GTEx, where data access regulations legally prevent us from sharing the data), and are in the process of setting up additional hardware to allow us to serve the remaining public-data graphs through our web service. All graphs can be downloaded from s3://metagraph-data-public for performing local (or cloud-based) analyses.

5. I remain unconvinced of the claim that the method can scale to the entire SRA at the projected costs, given that the most challenging datasets, raw environmental metagenomes, were excluded. How much time/RAM/disk was needed to construct the largest indices? (MetaGut, Plant, Metazoa)

We thank the reviewer for their comment. The subset of the SRA from which we extrapolated our cost projections consists of all samples from 100 randomly-selected studies. We have chosen this to be a representative subsample of the SRA, including whole-genome, amplicon, and environmental metagenomics sequencing samples. To better highlight the diversity of this data set, we have added Supplementary Table S-3 (referenced in Table 1), tabulating the distribution of samples in this subset.

Additionally, we have improved the presentation of our cost projections by including a new Figure 5, plotting the cost estimates for both exact k-mer matching and sequence-to-graph alignment using MetaGraph. We have also included an extension of this plot as Supplementary Figure S-14 comparing these costs to kindex and meta-Fulgor.

6. Have SRA transcriptomic data been indexed? Given the library selection method (“GENOMIC”), it appears that this is not the case, but this is unnoted in the main text.

At the moment, we have created indexes for GTEx and TCGA. In addition, 19.6% of the accessions in our 100-study subset of the SRA were RNA-Seq samples. To represent a wider range of transcriptomics data, we are currently in the process of indexing subsets of RNA fraction the SRA, which will be released in the near future.

7. In the “back-splice junctions in GTEx and TCGA” section, TCGA was not surveyed.

We thank the reviewer for bringing up this oversight. We, in fact, did perform this analysis and released parts of it in a previous version of this manuscript. We have added it back to our updated manuscript. We have extended Figures 4 and Ext Figure 4 with additional panels and modified the text to now include both cohorts.

Minor remarks in main text:

8. “When indexing the SRA as a whole, the final MetaGraph index constructed from it exceeds the amount of memory available on a single machine.” This sentence, while likely true, should be reformulated to clarify the SRA has not yet been indexed as a whole.

We thank the reviewer for pointing out this oversight. We have revised the sentence as follows: “For indexing the SRA as a whole, a single MetaGraph index would most certainly exceed the amount of memory available on a single machine.”

9. Regarding the sentence “storing the MetaGraph indexes even in their most performant representation”: what is meant by “most performant” here?

We thank the reviewer for bringing up this oversight. We have extended the text to clarify that “most performant” refers to “fastest query time”.

10. The sentence “Processing the remaining 90%, as well as all future incoming data, can be easily solved via scaling through parallelisation” greatly downplays the practical difficulties of processing the remaining 90% and the adjective “easily” should be removed.

We appreciate and agree with the reviewer's concern regarding the practical difficulties of indexing the SRA in its entirety. We have revised the text to now read "can be achieved via scaling". When processing the samples included in the current MetaGraph indexes the biggest limitation was actually reliable transfer of the input data onto the processing server for processing. Oftentimes samples were not available in the cloud or timed out when using direct download from SRA or ENA. Once we had access to a sample, we were able to index it with MetaGraph.

11. The technical sentence "[...] the alternative approach of chaining together unitigs to form longer contigs is a clear target for future work" is out of place in the main text due to its lack of context (contigs are de facto chains of unitigs).

We agree with the reviewer that this sentence appears without the proper context. We have replaced it with a discussion of MetaGraph's cleaning strategy and how future work can focus on improvements inspired by de novo genome assembly.

Remarks in the Methods section:

12. Regarding the sentence: "traversal in contig mode extends a traversed path until no further outgoing edge is present or if all the next outgoing edges have already been traversed", it appears from this definition that contigs follow arbitrary branching hence can contain chimeric sequences that are biologically unsound. Is this what the authors meant?

We thank the reviewer for this comment. Indeed, the sentence was interpreted correctly. As clarified in our response to your question 11, the sequence assembly operations in MetaGraph are designed to produce a sequence set representing each k-mer exactly once (i.e., a spectrum-preserving string set) without the intention to produce a strictly biologically-accurate set of contigs. Thus, chimeric sequences are a possible outcome of this design choice. As noted above, the task of using MetaGraph as a basis for genome assembly is left for future work.

13. Regarding: "Depending on the structure of the query data, this algorithm achieves a 10- to 100-fold speedup" over which baseline?

We thank the reviewer for this comment. The comparison here was done against the basic unbatched version of the query algorithm. We have revised the manuscript to clarify our baseline: "...compared to unbatched queries."

14. Regarding: "First, seeds corresponding to the forward orientation of the query are found according to the algorithm described above." Which algorithm? immediately above was the extension algorithm.

We thank the reviewer for pointing out this oversight. We have revised the section as follows: "First, seeds corresponding to the forward orientation of the query are found, which correspond to contiguous node ranges in the BOSS representation of the graph (see Supplementary Section A.3.2 for a description of the node range matching algorithm)."

15. Regarding: "when input sequences are losslessly encoded in a MetaGraph index": could you expand on this? It is also mentioned in the caption of Table 1, but the capability and trade-offs of MetaGraph to losslessly encode sequences by storing genome coordinates is not much developed in the manuscript.

We have extended this sentence to better clarify that lossless encoding of sequences using De Bruijn graphs was introduced and evaluated in a separate technical publication: "Finally, when input sequences are losslessly encoded in a MetaGraph index using the methodology introduced and evaluated by Karasikov, et al., the alignment can be done against those original input sequences whose respective walks in the graph are called "traces."

16. Regarding: “used a constant cleaning threshold of 2 during graph cleaning to remove all unitigs with a median k-mer abundance of 1” even when the unitig is not a tip nor isolated? Also, what is “singleton filtering”?

We have revised the Graph cleaning subsection of the methods to further motivate our cleaning strategy. In particular, we note that an internal unitig with low abundance is one branch of a graph topological structure called a “bubble”, typically created by sequencing errors. Discarding such unitigs is referred to as “bubble popping” and is a standard assembly graph cleaning strategy.

We have also clarified the definition of singleton filtering in the text as follows: “Namely, we have switched off the singleton filtering (i.e., we initially kept all k-mers that appear only once)”

17. Regarding SRA-Metazoa: if taxid 33208 was used to select samples, how were human samples excluded?

We thank the reviewer for this comment. Specifically, for the SRA-Metazoa index we excluded all retrieved records with taxid 33208 that also had the “organism” field populated with “Homo Sapiens”. As this field contains controlled vocabulary, we believe this is an effective way to remove human samples from that subset. We have made the queries available alongside the other metadata: https://github.com/ratschlab/metagraph_paper_resources/tree/master/metadata_queries

18. Missing section number in “in supplemental resources given in Section)”.

We thank the reviewer for pointing out this oversight. Since this subsection is unnumbered, its reference was rendered empty. We have inserted the text of the subsection header to guide the reader.

19. Supplementary material, “indexing costs” section: the projection implicitly assumes linearity in computing costs w.r.t input accession size. Is this the case? I would assume some datasets (metagenomes, low-coverage isolates) to be significantly more challenging to index. Please also add that the projected costs are lower bounds that do not take into account several cloud computing overheads: scratch storage costs, latency of the batch system, data download and input file conversion.

As noted in our response to your comment 5, we have included a Supplementary Table S-3 detailing the composition of the random subset of the SRA we used for our cost estimation. Being a representative subset, challenging data sets, such as metagenomes, are indeed considered in our cost estimates. Additionally, the estimates for the indexing cost fully include the overhead for the scratch storage costs, downloading the raw input files, and the necessary file conversions as they are based on the actual processing we did for that subset.

Code:

20. I could compile and test the code locally from sources with only minor issues. The compilation process requires an Internet connection as the Makefile downloads components.

We thank the reviewer for this suggestion. We are in the process of improving the compilation process to no longer require an active internet connection.

21. I could not construct an index with counts: “To construct a MetaGraph index with k-mer counts (Counting de Bruijn graph), construct a de Bruijn graph as usual (see Construct graph) and then add --count-kmers to the annotation command, e.g.:" this step did not work, I get the error “[warning] No k-mer counts found at './tests/data/transcripts_1000.fa.kmer_counts.gz'. Every input k-mer will have count 1.”. My command lines were: “./metagraph build -v -p 4 -k 31 -o graph \$DATA” (to “construct a de Bruijn graph as usual”) then “./metagraph annotate -v -i graph.dbg --anno-filename --count-kmers -p 4 -o annotation \$DATA”.

It seems that the mentioned file `./tests/data/transcripts_1000.fa.kmer_counts.gz` was created by a previous command `metagraph build --count-kmers ...` + `metagraph transform --to-fasta

...), which generates two files: The first file `../tests/data/transcripts_1000.fa.kmer_counts.gz` is the compressed array of the k-mers counts, while the second file contains the sequences extracted from the graph `../tests/data/transcripts_1000.fa.fasta.gz`. It is the second file `*.fasta.gz` that should be passed as input for graph construction and annotation, or the original file with sequences (here `../tests/data/transcripts_1000.fa`). In other words, DATA in the reviewer's command above should be not `../tests/data/transcripts_1000.fa.kmer_counts.gz` but `../tests/data/transcripts_1000.fa.fasta.gz` or simply `../tests/data/transcripts_1000.fa`.

22. I recommend to the authors that they include additional tutorials that use the test data provided in the repository (`../tests/data/transcripts_1000.fa`) instead of data that is not provided (e.g. SRR403017) as presented in the documentation: tutorials for construction of an index with k-mer counts, construction of an index with genomic coordinates, sequence-to-graph alignment.

We thank the reviewer for this suggestion. Since a typical indexing workflow involves downloading sequence data from an external source, we have instead updated the tutorial to include a download step for SRR403017.

23. The sequence search page is incomplete:

https://metagraph.ethz.ch/static/docs/sequence_search.html#align-sequences-to-the-graph

We thank the reviewer for bringing up this oversight. We have updated the documentation to include a description of sequence-to-graph alignment and its output format.

Typos:

- In abstract: "Mbp".
- In Extended Figure 4: "representas".
- In supplementary material: "RowDiff;RowFlat;"

We thank the reviewer for these comments, we have fixed them.

Referee #3 (Remarks to the Author):

Comments to the author

In this manuscript, Karasikov et al develop and apply a method to improve compression of DNA sequencing data. They create a compressed representation of a large subset (but not comprehensive collection) of the data available from the NCBI SRA. They also present approaches to perform cost-efficient querying the graph-based compressed data. Overall, the presented work is clear, the MetaGraph approach is useful, and it does provide a benchmarked advantage over existing approaches. Limitations of this work aren't as well described as they could be, and while this is certainly a modest advance in computing, it is unclear to me how this represents a major advance over the existing graph based lossless compression algorithms.

We thank the reviewer for this comment and agree that the improvements in compression of the data are modest compared to other specialised DNA-compressors. However, the core message of our manuscript is the feasibility of SRA-scale full-text sequence indexing. Consequently, maintaining searchability is a central component of our endeavour. We would like to emphasise that improving compression is necessary but on its own not sufficient for scaling up sequence search. Thus, compression has to be viewed in the context of data layout, data representation and storage, index construction, as well as query performance. Balancing these components against each other is essential for creating a scalable and yet affordable full-text indexing system. We also would like to point out that MetaGraph has shaped and still is strongly shaping the field of population-scale sequence search. While the idea of full-text searching the full SRA is not revolutionary, we would claim that demonstrating its feasibility is and we were the first ones to do so (within the given limitations of a single lab).

Several of the technical contributions used in the MetaGraph journal have been described in more detail in specialised technical venues. We find that this allows us to focus on the overall scalability of search in this manuscript, which is addressed to a much broader audience. Thus, alongside the brief review of the algorithmic aspects of MetaGraph, our work presents the first demonstration of feasibility and cost estimation for indexing the entire SRA using our proposed strategy. We also provide pre-computed indexes of important subsets of the SRA as a community resource. Although some of the core algorithmic work was presented in other venues, we designed distributed and scalable implementations to enable our indexing efforts. In addition, we would like to refer you to our common response C1 for a more expanded discussion of MetaGraph's contribution.

Regarding the limitations of our strategy - we have expanded our Discussion to describe future work for improving the sample cleaning strategy we use during index construction.

Undeniably, the rate at which data are being generated is rapid and accelerating, with the decreasing costs of sequencing. Thus, finding ways for lossless compression of the data is critical for sustainability. The approach they describe compresses data variably based on type (human RNAseq data, for example, is compressed very efficiently; metagenomic data is much less well compressed).

The authors note limitations of existing approaches for compression (such as VG) and also note the challenges of alignment approaches (e.g. BLAST). They then explore the use of k-mer based approaches, but indicate that these types of approaches, while efficient, are inadequately sensitive. They present MetaGraph as an approach to address current limitations in data size (through compression) and analysis (through a graph-based alignment approach).

We thank the reviewer for their positive assessment and support of our endeavors. We agree that lossless compression of these large databases is critical.

Overall, the manuscript is clear and relatively well written. It is certainly written for a relatively specialized audience, and for me, felt like a manuscript one would read in Bioinformatics or perhaps PLoS Computational Biology. I think the work they present is of quite high importance, but given that

Nature is a general journal with a very broad readership, the manuscript could benefit from slightly more general framing.

We thank the reviewer for this suggestion. We have expanded the introduction to provide a broader overview of the field, including the available public databases for DNA sequencing data, and more detailed descriptions of DNA sequencing and the kinds of queries we wish to enable with our work.

I appreciate the benchmarking of MetaGraph against available tools for compression, such as BIGSI, COBS, etc. On one hand, the 16-38 times smaller representation from MetaGraph is a definite improvement compared to the other approaches; on the other hand, I do wonder if improving compression by this amount is an advance sufficiently exciting to describe this approach as truly transformational. With the rate of data generation, a much more significant compression will be needed to make a substantial dent in the challenges that we face as a computational community. Furthermore, while some of the compression algorithms are lossy (BIGSI and COBS), not all of them are. All that being said, this is an advance and is notable.

We thank the reviewer for this comment. Our argument is that both our scalable indexing strategy and our large-scale demonstration of its efficacy and applications are transformational. We agree that the compression in itself is not transformational, but it is a fundamental contributor to our demonstration of the feasibility of this indexing endeavor. In addition, MetaGraph is designed as a framework that can benefit from continued advancement in algorithms and data structures. That is, the same indexing strategy and setup can be used with new k-mer coordinate schemes (such as minimal perfect hashing) or further improved label compression. Lastly, echoing our statements made in our reply to the reviewer's first comment, our focus is not on compression alone but on demonstrating feasibility of SRA-scale full-text search. Here, improving compression is a necessary but not a sufficient component of realising feasibility. Our approach, for the first time, allows analyses that can integrate information across millions of samples, possibly uncovering many new answers to burning questions about the diversity of the encoding of life. All this additional functionality is achieved at a considerably smaller memory footprint than representing the (already compressed) version of the raw SRA data.

Fig 2. In terms of performance against a difficult task, like a human gut microbiome query, MetaGraph performs very similarly to Mantis (another lossless approach), but it seems that the construction of Mantis indexes could not be done on all of the subsets due to a timeout. This is only discussed in the figure legend – given the excellent performance of Mantis, I think it would be useful to discuss the findings and then the potential reasons for the timeout for Mantis in the main text.

We thank the reviewer for this comment. After re-evaluating our Mantis benchmark on a system with a different configuration (in particular, with access to a local NVMe SSD), we have concluded that our previous execution time issues resulted from our use of a network filesystem for storing intermediate files during index construction. We have extended Figure 2 to include the missing data points and extended the Discussion to note that some index representations rely on fast local storage for optimal performance.

Table 1. I think this is a very useful table. I wonder if the authors can contextualize the utility of their tools given the expected 'increase' in SRA deposits over the coming years. If there is some reference/resource that has made estimations, it might help bolster their argument that this is a highly useful tool. On one hand, those who are doing metagenomics might find this tool to be incremental in its utility. On the other hand, those who are doing scRNAseq might find great utility in this tool. To that end, given that quite a lot of scRNAseq data is now being generated, it would be valuable to see the utility of this tool in the context of that data type.

We agree that scRNA-Seq is an important emerging data type. Using our counting De Bruijn graph indexing strategy (Karasikov, et al. 2022), the MetaGraph framework can index scRNA-Seq

sequences and corresponding k -mer abundances. Depending on the specific single-cell sequencing technology, our approach might be well or less well suited. Specifically, the broadly used 10x genomics employs approach uses only end tags to quantify transcript abundances, resulting in a markedly lower diversity in these sequences compared to whole-genome or whole-metagenome sequencing reads. In this case, generating a searchable full-text index might be less suitable than just tabulating the data. Instead, a search by count could be implemented. However, we feel that this functionality is beyond the scope of this manuscript, but is nonetheless a worthwhile pursuit for future work.

On the other hand, when using other technologies, such as long-read single cell sequencing, the data is well suited to be represented with a MetaGraph index. This type of data only starts to appear in the public databases and was not the primary target for our feasibility study. However, this will be an interesting future direction.

One thing that isn't immediately obvious from the way the manuscript is written is how alignments will fare in the context of indels or SNPs (or even larger structural variations) in the context of graph alignment. These important issues should be addressed in the manuscript to maximize utility of the findings.

We thank the reviewer for this comment. Our alignment algorithm is a generalisation of the Smith-Waterman-Gotoh algorithm to a sequence graph setting, allowing for indels as edits. As part of the alignment evaluation done in Fig 3b, we aligned sets of reads with simulated errors. As described in our Methods section, our simulation included both SNPs and indels, where indels occurred at 1/10th of the rate of substitutions.

To account for larger structural variations, our algorithm supports outputting multiple alignments per query. Thus, a chimeric alignment of a long read (or a discordant alignment of a short read pair) can be used as evidence for SV discovery. We have extended the "Sequence search with alignment" section of the Methods to clarify this point: "The user can choose to report multiple alignments for each query, which may be found if seeds to multiple locations in the graph are discovered."

The BSJ analysis is quite interesting. It would be helpful to have a supplementary figure that shows a toy example of how this works.

We thank the reviewer for this suggestion. We have extended our description of the BSJ results and also added a schematic outlining the formation of BSJ in circular RNAs (Extended Data Figure 4).

The web-based GUI is intuitive and easy to use. Can it handle batch queries? As best as I could tell, it cannot. This makes me wonder what the context for its use would be, as the throughput is currently limited to one sequence at a time.

Currently, the command-line interface for MetaGraph has a more extensive feature set than the web interface since it targets larger-scale local querying. We are currently in the process of extending the web API to incorporate more features from this API.

While it is undeniable that storing all available data on a commercial cloud would be costly, one of the major advantages of cloud computing is that such references can be centrally stored and accessed by hundreds, thousands, or even more groups all over the world. I would presume that eventually, most will not want or need create their own 'local' copies of such data in the cloud computing environment. The way this section of the manuscript is written doesn't acknowledge this.

We thank the reviewer for this comment. We would like to note that there is, indeed, no requirement for each user to have a copy of a MetaGraph index, and that the indexes can be stored centrally for many to use. Since the indexing cost has to be paid at least once (by an institution), we reflect this in the cost table. We have reformatted our cost table to reflect this use case, reporting the query costs incurred by an end-user accessing a centrally-stored index.

Response to Reviewer Comments

Introduction

We would like to express our gratitude to all reviewers for providing constructive feedback on our manuscript and the MetaGraph framework in general. The revision process has triggered several fruitful new developments that we believe have made MetaGraph a more valuable resource for a wider community.

Similar to the previous revision round, we have structured our response in two parts. First, we provide general comments on the current state of indexing and our evaluation of query costs – topics that were raised by several reviewers. Second, we provide a point-by-point response to all remarks made by the individual reviewers. The reviewers' text is provided in black, while all our responses are shown in blue italics.

Comment on current state of indexing

A major concern shared by several reviewers throughout the revision process was MetaGraph's capability to practically index vast amounts of diverse data (especially metagenomics), as is present in public archives such as the Sequence Read Archive (SRA).

To demonstrate that there is no technical limitation that prevents MetaGraph from indexing at such a scale, we are happy to report that we were able to greatly increase the number of samples available as a MetaGraph index. Including indexes generated from a subset of the publicly available Logan dataset, we have now completed indexing of more than 69% of the 27 million samples available in the Logan dataset and aim to complete the remaining 31% in the coming weeks and months. We have selected the contig-level samples from Logan as they are most comparable to the cleaning strategy we have applied in our MetaGraph indexes compiled so far.

Recently, our group has been approved to participate in the AWS Open Data Sponsorship program. As a result, all currently computed public SRA indexes (that is for more than 18 million SRA accessions) are publicly available to the research community in the cloud on AWS fast S3 storage (s3://metagraph) for online processing or download. We are in the process of procuring further compute resources to not only complete the remaining fraction of SRA, but also to keep up indexing of newly submitted datasets. If there is need in the community, we are also prepared to index the unitig-level assemblies from Logan, which will increase the final index size by a factor of 2-3, as there is slightly more signal in the data, but also considerably more noise in the sequences.

Evaluation of our cloud compute query cost model

A major concern raised by Referees #1 and #2 was the accuracy of our querying model within a cloud compute environment. To address this, we have measured the query costs directly on a large set of accessions (more than 230,000) already present in the AWS cloud and updated Figure 5a to present our empirical measurements. We have also updated Supplementary Figure S-14a based on an updated querying model reflecting our new insights about our cloud compute environment.

In our new Figure 5a, we report a projected cost of 0.74 USD per Mbp of query against the entire set of 33 million public SRA samples (as counted in January 2025) for large queries and ~100 USD for

small queries up to 1 Mbp. We attribute the increase (by a factor of 1.7) in cost for large queries compared to our previous estimates to the sub-linear scaling of query throughput with increasing physical CPU count. We attribute the more substantial increase in the cost for small- and intermediate-sized queries to (i) greater index transfer times due to lower real-world network bandwidth speeds, and (ii) the substantial cost of container image spin-up during query job initiation. We believe that both factors can be addressed, at least partially, through further engineering efforts. Eventually, the lower cost for high throughput queries can always be achieved through aggregating multiple small queries into larger batches.

We have updated Supplementary Figure S-14a to reflect these new insights. Along with incorporating more accurate network bandwidth speeds (increasing the amount of time taken to transfer indexes from cloud storage to the compute environment), we now also factor in the time taken to initialize the container when starting a query.

Referee #1 (Remarks to the Author):

Comments for Author

I thank for the authors for the detailed point-by-point response to the issues I raised. Most of them appear to have been addressed in the most recent revision, making the manuscript more suitable as a version of record for the MetaGraph project (and associated development), which as I detailed in my previous review, is a substantial contribution to the literature.

First, I thank the authors for having changed the paper title. The work is impressive, but not over-selling is just as important.

We would like to thank the reviewer for the positive assessment of our work and for acknowledging the improvements made in the revision process.

Having said that, I do still worry that random subsetting to build extrapolations might fail in subtle ways. To pick one example in the authors' favor, compression ratios are likely to improve as more similar data is acquired. An example in the other direction though: indexes at a fixed k-mer size might get more crowded increasing query runtimes. My intuition is that the cost and scalability estimates are reasonable, and the authors do discuss some of these subtleties, but the simple linear scaling still feels suboptimal, as scaling is never as easy as it initially looks.

We understand the reviewer's concerns and have thus begun to index a much larger dataset to demonstrate that our projections, as described in the manuscript, indeed hold for the existing data. As outlined in our general response above, we have used the Logan data set to index additional samples. While completion of the effort is still ongoing, we are happy to report that, at this point in time, we have indexed more than 69% of all Logan samples (18,796,454 out of a total of 27,186,967 SRA accessions). Specifically, we have prioritized whole-metagenome sequencing samples. All built indexes are available publicly in the cloud under s3://metagraph.

To address the second aspect of the comment, concerning the scalability of query, we have tested our model directly on more than 230,000 accessions already present in the cloud. Please refer to our common response for an explanation of our findings.

I would like to explicitly thank the authors for the updated benchmarking against new tools, as I know how much work that can be.

Thank you very much for recognizing the effort.

I do still think using the terminology of "lossless" is a bit confusing. I agree with the Authors that the MetaGraph approach is perhaps "less lossy" than other "lossy" competitors, and that it is unclear that researchers would ever even want a fully lossless index of unfiltered data---personally, I tend to come down on the side of saying that a fully lossless index of unfiltered data is unnecessary. However, as a matter of principle and to avoid confusion, I do think it needs to be made clear to a non-expert reader what exactly the work purports to do. This is largely a matter of semantics, and I do not think changing the terminology is by any means needed before publication, but I do exhort the authors to carefully consider the potential confusion this choice of terminology may cause a general audience.

We thank the reviewer for bringing up this topic again. Indeed, this is a technical detail that could easily confuse a reader. We have extended the discussion to better emphasise that our use of the term "lossless" refers only to the subset of k-mers chosen from each sample to be represented, and that this indeed results in a technically lossy index of the input sequences and their k-mers for some indexes. We have noted that a fully lossless encoding is currently done for indexes of assembled sequences (e.g., RefSeq) and that we have chosen not to use this approach when indexing read sets (although we have previously demonstrated our ability to do so [1]) to improve the query performance, and hence accessibility, of these indexes.

*[1] Karasikov, M., Mustafa, H., Ratsch, G., & Kahles, A. (2022). Lossless indexing with counting de Bruijn graphs. *Genome Research*, 32(9), 1754-1764.*

Overall though, I think the manuscript is substantially improved since the last time I read it. My one final thought is that the manuscript should make even more explicit how this work benefits the scientific community in practice, given the general audience of Nature. Having said that, I believe the manuscript is likely suitable for publication.

We again thank the reviewer for recognizing our work. To make MetaGraph and the generated indexes accessible to a wider audience, we have now deposited all data in the cloud (under `s3://metagraph`) and have developed a query framework [2] that allows users to formulate their own queries and run them across the whole dataset indexed so far. Eventually, even queries of a few megabases in size can be run against all indexes of 33 million accessions for a cost of just around 100 USD. For larger queries, the effective throughput increases, and the price goes down to just around 0.74 USD per Mbp of query (see Figure 5a).

[2] <https://github.com/ratschlab/metagraph-open-data?tab=readme-ov-file#usage-within-aws>

Referee #2 (Remarks to the Author):

Comments for the author

The authors revised their manuscript presenting MetaGraph, with a slightly updated title that fixes the scope, and many new supplementary analyses supporting the reviewers' remarks. I still have remarks mainly on the software side, yet comparatively to the magnitude of the contribution, these remarks are minor. One additional point that I develop below is on the extrapolation to full SRA, that still appears to be uncertain.

We would like to thank the reviewer for the constructive feedback and appreciate the positive recognition of our work.

Major remark:

0. I appreciate that the authors provided supplementary table S3, to give an overview of the 100 studies test for indexing the entire SRA, but I remain unconvinced that the approach scales. (What is “WMGS” in that table?) I will refine my remark by asking the authors to do a cost projection based on indexing the metagenomes alone (an increasing number of them), to see if the previous extrapolation still holds up.

We appreciate the reviewer’s concerns. As mentioned in our response to Reviewer #1, we have indexed a much larger proportion of SRA accessions and empirically evaluated our SRA querying workflow on the AWS cloud. We have also updated our querying model to reflect new insights drawn from these experiments. Please refer to our common response for a detailed explanation of our findings.

We have also included a cost projection for querying only metagenome chunks as Supplementary Figure S-17, where we observe that querying chunks indexing metagenomes incurs similar costs to querying chunks indexing random studies.

To address the minor comment, we have clarified in Table S3 that “WMGS” stands for “whole metagenome sequencing”.

Questions on new Figure 5:

1. “despite our indexing approach being lossless” as noted by previous a reviewer comment, this is a misleading claim as

Unfortunately, the reviewer’s comment was truncated. We assume that the question is related to possible misconceptions of the term “lossless”. Aligning with our response to the second point of Reviewer 3, we have added a “Limitations” section to the manuscript that provides further details on how to understand losslessness in this context.

2. Fig 5c and also Fig 3, I do not understand what the metric “Sequence Identity Cutoff” represents, could you please explain how it is defined?

Thank you for bringing up this point. We have extended these captions to provide a definition of “Sequence Identity Cutoff”. Given a mapping of a query read to a sequence in the graph (i.e., the sequence spelled by a graph walk) from a single (not necessarily unique) annotation track (e.g., label, accession, etc.), the “sequence identity” is the percentage of read nucleotides that match the graph sequence exactly. For a given “sequence identity cutoff”, only mappings with sequence identities above that cutoff are considered when calculating recall.

Software/docs:

3. In my previous review, on remark 21 I had issues running a provided example. I revisited it. It is worth mentioning that a few tools are further needed as compilation dependencies: aclocal from automake, autoreconf from autoconf, and libdeflate; but this is a minor point. I could compile the new version of the software.

We thank the reviewer for evaluating our installation and quick start instructions. Although our instructions already included automake and autoconf as prerequisites, we have now added libdeflate

to the list. In addition, we have updated our Dockerfile with these new prerequisites and included a Docker testing step to our continuous integration workflow to ensure its continuing reliability.

4. For testing the beginning of quick start (https://metagraph.ethz.ch/static/docs/quick_start.html), please add information on which folder to run the tests (supposedly it is tests/data).

We have updated the quick start instructions to include this information. It is indeed easiest to execute the instructions from the tests/data subdirectory, assuming that the metagraph executable is included in the \$PATH environment variable (the updated instructions also mention how to proceed if this is not the case).

5. The quick start tutorial “Index k-mer counts” still needs fixing. The authors mention in their response to my remark 21 that: “It seems that the mentioned file `../tests/data/transcripts_1000.fa.kmer_counts.gz` was created by a previous command `metagraph build -count-kmers ... + metagraph transform -to-fasta [...]`” however these steps do not appear in the “Index k-mer counts” part of the tutorial, nor elsewhere. To be clear, I am trying to run the quick start tests starting from https://metagraph.ethz.ch/static/docs/quick_start.html#index-k-mer-counts, so I went back to the “Construct graph” section as instructed there.

We have updated the count indexing instructions. We have included a forward reference to the “Annotate with pre-counting” subsection describing the creation of the “.kmer_counts.gz” file.

6. In response to my remark 22, the authors write “we have instead updated the tutorial to include a download step for SRR403017” however I do not find this download step anywhere in https://metagraph.ethz.ch/static/docs/quick_start.html. Is it elsewhere?

Please accept our apologies for this oversight. We indeed had updated our documentation resources prior to resubmission (<https://github.com/ratschlab/metagraph/commit/a1e49c0bb3dcdd0c12876509d4857486df9a965a>). Unfortunately, these changes were not properly synchronized to the online documentation available at metagraph.ethz.ch. We have fixed this now and the documentation should be accessible here: https://metagraph.ethz.ch/static/docs/quick_start.html#construct-from-kmc-counters

7. My remark 23 either was not addressed: “The sequence search page is incomplete: https://metagraph.ethz.ch/static/docs/sequence_search.html#align-sequences-to-the-graph We thank the reviewer for bringing up this oversight. We have updated the documentation to include a description of sequence-to-graph alignment and its output format”. I still see an empty section at that URL.

Please accept our apologies for this oversight. We indeed had updated our documentation resources prior to resubmission (please refer to commits [e040d5a](https://github.com/ratschlab/metagraph/commit/e040d5a) and [89a2ee5](https://github.com/ratschlab/metagraph/commit/89a2ee5) here: <https://github.com/ratschlab/metagraph/commits/master/metagraph/docs>). Unfortunately, these changes were not properly synchronized to the online documentation available at metagraph.ethz.ch. We have fixed this now and the documentation should be accessible here: https://metagraph.ethz.ch/static/docs/sequence_search.html#align-sequences-to-the-graph

Typos in updated text:

“Sequence Read Archive (DRA)” should be SRA.

We thank the reviewer for highlighting this possible point of confusion. As noted in the manuscript, the Sequence Read Archive maintained by the DNA Data Bank of Japan (DDBJ) is abbreviated as "DRA". To avoid confusion, we have updated the manuscript to clarify the acronym.

Referee #3 (Remarks to the Author):

Comments to the author

Overall, the manuscript is improved, which I appreciate. I continue to wonder if this will be accessible to the 'general genomics researcher'; however, it certainly will appeal to the highly CS-focused portion of the genomics community, and does represent an important step forward. Assuming MetaGraph is maintained and remains competitive (in terms of performance and usability) compared to other tools that can perform similar tasks, it has the potential to be a useful tool for the genomics community in the years to come.

We thank the reviewer for the positive feedback. We have been developing MetaGraph for the past 7 years and are committed to supporting further improvements. We have secured funding to extend MetaGraph's usability and make more indexing data available to the public.

1. It would have been nice to have a version of the manuscript with changes highlighted.

We thank the reviewer for making us aware of this. We have double-checked our past submission and the pdf with all changes highlighted has correctly been uploaded. We have reached out to the editor to inquire whether there has been a technical problem in the journal's submission system that prevented the reviewer from accessing the pdf.

2. Some limitations are pointed out - it might be easier for the reader to follow if there is a dedicated limitations paragraph.

We thank the reviewer for this suggestion. We have reformatted the discussion to include a dedicated "Limitations" subsection. We have expanded this subsection to further detail our use of the term "lossless" to describe the nature of our indexes.

Response to Reviewer Comments

Introduction

We thank all reviewers for the constructive feedback and the continuing effort towards improving our work. In the following, we provide a point-by-point response to all remarks made by the individual reviewers. The reviewers' text is provided in black, while all our responses are shown in blue italics.

Editorial Comment

In line with prior comment from referee 3, we also feel your article would benefit from being more upfront about the limitations in the discussions section.

We have extended the "Limitations" section in our Discussion to more broadly reflect known shortcomings of the current MetaGraph method. We hope that this will make it easier for the readers to understand the scope of our work.

Editorially, we also feel that "All Life's Known Biological Sequences" is inappropriately grandiose for an article title, so we kindly request that you change that.

*We thank the editor for this constructive feedback. We have revised our title as follows: **"Efficient and Accurate Search in Petabase-scale Sequence Repositories"***

Referee #1 (Remarks to the Author):

The authors have addressed all of the issues I raised previously, and have improved the discussion. I especially appreciate the additional indexing of further samples, which does indeed demonstrate that the MetaGraph approach's scaling behavior.

I thank the authors for their hard work. I consider this manuscript suitable for publication.

(I have previously used the software, but did not do a deep dive this round, because the software/code was already good quality in a previous round of reviews)

We thank the reviewer for their positive assessment of our work.

Referee #2 (Remarks to the Author):

The authors revised their manuscript presenting MetaGraph, changing the biological sequence data index, which is the central focus of the paper, from an index of "SRA sequence data" to an index of pre-processed contig data, "Logan contigs".

We thank the reviewer for his assessment. However, we would like to point out that none of the previous data has been discarded. Instead, as requested by the reviewers, we have expanded the demonstration of scalability of our approach by indexing further sources of publicly available data (e.g., Logan contigs). The main claim of the manuscript remains valid (feasibility of efficient, accurate search) and our results show this in different ways (e.g., based on 100 random studies, indexing a significant fraction of SRA and now, in addition, indexing a good fraction of Logan contigs)

The revised manuscript's methods and claims regarding MetaGraph's scalability depend on Logan contigs, therefore several methods and novelty claims throughout the manuscript are now overstated. As it stands, MetaGraph is not shown to be sufficient for pan-SRA index scalability without a major external pre-processing dependency. This is not a critical flaw to the method, but the manuscript should provide a fair and complete description of the essential prior work in the Logan project, which it now depends upon.

We truly appreciate the public availability of the Logan resource. However, we believe that the contexts in which we used Logan data do not weaken the substantiality of our efforts and results. The main claim of our study is that efficient and accurate search in large sequence repositories is feasible. This is because of advanced succinct graph and bitmap algorithms that exploit redundancies in the sequence data. We state the feasibility and ambition, but we do not claim global completeness for all public sequences (which is anyways hard, as the data set continuously grows and in January 2025 already exceeded 66.9 Pbp, 33% larger than Logan's December 2023 snapshot). In particular, in the previous manuscript version (and still in the current manuscript), we have shown that MetaGraph is indeed "sufficient for pan-SRA index scalability without a major external pre-processing dependency." As demonstrated with our previous data sets (~5PB), completing an equivalent index without the Logan resource would have mainly incurred a higher one-time construction cost and was not prohibited through methodological limitations (as the reviewer correctly states). We have extended the discussion to elaborate more on the relation to the Logan resource.

At the Logan index snapshot (December 2023), the SRA contained 50 petabases of sequence data, the Logan contigs are a compressed representation of that data of only 2 petabases which are now being indexed and presented as MetaGraph's index of the SRA sequence data.

We thank the reviewer for pointing this out as it provides us with the opportunity to clarify a potential misconception. We did not intend to present our indexes of the Logan contigs as an index of the full SRA, but rather use them as grounds for extrapolation to indexing the complete SRA repository. We have now made this more clear in Table 1. The final section of this table (SRA-Public) uses two different sample sets as grounds for extrapolation (up to the cut-off date 11.01.2025): i) a random set of 100 SRA studies completely processed with the MetaGraph pipeline, and ii) a set of 4700 random studies pre-processed by Logan. Both extrapolations lead to comparable compression estimates and we get a predicted final index size between 172 TB (using the 4700-study Logan set) and 224 TB (using the 100-study SRA set). In addition, we provide a line in Table 1 that also records the total number of Logan contigs indexed so far. The compression ratio observed for this subset of more than 16 million accessions is nearly identical to the one estimated from the earlier subsets. This

shows that the larger data sets are not necessarily needed to make accurate estimates on compression and cost.

The Logan contigs are publicly available since July 2024, and the index of all Logan contigs, Logan Search was made publicly available alongside that data since November 2024, albeit neither is yet published. While the Logan contig dependency is cited, the Logan Search index released alongside it was not mentioned (<https://logan-search.org/>). In my view the manuscript should more fairly represent the state of the art and add a qualitative description and URL reference to Logan Search, as to provide complete attribution to prior publicly available work, Logan contigs and Logan Search.

As the reviewer correctly states, we have cited the pre-print that has been made available for the publicly shared Logan data set. The cited resource provides details on how the data has been generated and allows for reproducing the data sets (if sufficient computational resources are available). For Logan Search, we were unable to find a pre-print or technical report detailing how the indexes are constructed (of particular importance are the choices of Bloom filter parameters, discussed below), which data they contain, whether any filtering of the inputs takes place, how the search is currently deployed, which search modes it supports (k-mer matching vs. sequence alignment), how accurate the search is, and how much the queries cost. Without these details, we did not see a reasonable way for a direct comparison. Since Logan Search is driven by a kmindex engine, we indexed a subset of the data using kmindex and included it into our scalability comparison. In addition, we studied the profound effects on random sample discovery rates for different choices of Bloom filter false positive rate (see Supplementary Figure S-14). However, we did not have sufficient information for a direct comparison with the Logan Search portal.

In absence of a published manuscript describing Logan Search, we cannot make direct and meaningful comparisons of our methodology to Logan Search. Also, the accuracy, efficiency and cost of the Logan Search are not publicly available, which are among the key factors in our manuscript. We have included and referenced Logan Search in the Discussion section as additional ongoing work. We hope this sufficiently addresses the reviewer's concern.

The new experiments asked by the reviewers also have clarified the scalability claims. Small queries which were previously projected to cost 14 USD, in the new revision are projected to cost 100 USD, an order of magnitude higher, showing the limits of previous extrapolations from smaller sets.

We thank the reviewer for bringing up this point. While several factors contributed to the final correction of our estimate, the sample size used as a basis for extrapolation only had a small impact. A much larger contribution came from correcting assumptions we have made for cloud resource efficiency and scheduling. In the prior version, we assumed theoretical throughput numbers as specified by the cloud provider, while these newly provided estimates were based on realistic measurements. For instance, while network connectivity in the cloud provider specifications was given as 40 Gigabit, the actual sustained throughput was almost a factor of 4 lower. The inclusion of larger data sets for estimation and carrying out queries using cloud infrastructure has made our estimates more robust. We would also like to point out that overall, the relative advantage of MetaGraph over other methods has not changed. We have used the 100-study subset initially and later also a small fraction of

the Logan resource (4,700 random studies) for the cost estimates. So, the claims in the paper do not depend on the whole Logan resource or on Logan Search.

Recommendation: Accept after minor revision.

signed,

Rayan Chikhi

Competing interests: I am a co-developer of Logan.

Major points:

1. Novelty claims vs. Logan Search: Please (i) cite Logan Search (<https://logan-search.org/>) alongside Logan contigs wherever feasibility is discussed, and (ii) adjust wording in Abstract, Introduction and Discussion that currently frames MetaGraph as the first or only demonstration of full-archive indexing and search:

a. in the Abstract: “We demonstrate the feasibility of indexing the full extent of all public biological sequences”. The authors’ claim of first demonstration is overstated, since Logan Search has been publicly available for months.

b. in the Introduction “[...] to demonstrate the practical feasibility of full-text indexing all known biological sequences” (similar)

Re. a,b: One of the main aims and claims of this work is that we show that indexing and searching large petabase-scale sequence repositories is feasible. We show this in different ways in the manuscript. We have also provided accuracy and cost estimates to confirm the feasibility of sequence search for genomics researchers in the community. For Logan Search, no such results have been made publicly available yet. However, we have added a reference to Logan Search in the Discussion and have adjusted our wording where applicable.

In particular, to avoid possible confusion with other ongoing efforts, we have changed the wording as follows:

a) Changed to: “We demonstrate the feasibility of efficient, accurate, and cost-effective full-text search in large public biological sequence repositories”

b) Changed to: “to demonstrate the practical feasibility of economical and accurate full-text indexing of biological sequence repositories”

c. “However, the raw sequencing data itself remain inaccessible for full-text search” It is false now that Logan Search exists.

We thank the reviewer for bringing this up. We agree that the landscape of computational solutions addressing the topic of raw data access and navigation is currently under very active development. We address this by changing our formulation to past tense. The sentence now reads:

“For the longest time, the raw sequencing data itself remained inaccessible for full-text search [...]”

d. “[...] to demonstrate that indexing repositories such as public sequence archives as a whole is not only theoretically possible, but practically feasible”: It has already been shown to be practically feasible, at a greater volume than achieved in the present manuscript.

We have changed the wording to more accurately reflect our claim that our work not only addresses feasibility, but also presents analyses on representation accuracy and query cost under different hosting and query scenarios. To the best of our knowledge, such an analysis has not been presented for any other solution so far. The corresponding sentence now reads:

“[...] to demonstrate that searching public sequence archives as a whole is not only theoretically possible, but practically feasible at high accuracy and affordable costs.”

We would also like to note that Logan Search has not been published and also not been fully peer reviewed, and, hence, the mentioned claims about Logan Search have not been validated by others (at least not to our knowledge).

e. in the Discussion: “As we set out to demonstrate the practical feasibility of indexing entire sequence archives” (similar)

In line with the changes suggested in the prior responses, we have altered the sentence as follows:

“As we set out to demonstrate the practical feasibility of accurate and cost-effective search in entire sequence archives [...]”

f. “Notably, this is an analysis that would otherwise have required access to hundreds of terabytes of raw sequencing data.” Using Logan contigs or unitigs would also enable this analysis, with a volume of data 10-100x smaller.

We have formulated the statement more generally. The text now reads:

“Using classical approaches, this analysis would have required access to hundreds of terabytes of raw sequencing data. With compressed sequence representations it can be done on a single compute node in about an hour.”

g. in the Discussion: “a crucial milestone in computational modelling, solving the problem of making all existing biological sequence data searchable and easily accessible” Regardless of Logan Search, this claim needs to be toned down, as per previous reviewer remarks.

We thank the reviewer for raising his concern about the alignment of this claim with the remainder of the manuscript. We have rephrased the wording as follows:

“The results presented in this work are an important milestone in computational genomics, demonstrating the feasibility of accurate and cost-effective search in petabase-scale biological sequence repositories [...]”

h. in the Discussion: “For the first time, it is now feasible to efficiently search” needs to be amended.

In agreement with the previous changes, we have also rephrased this statement to better align with the claims made in our manuscript. Specifically, we focus on the provided results regarding accuracy and maintenance/search cost. The new sentence reads: “[...] we have demonstrated that it is feasible to accurately and cost-efficiently search for and align nucleotide sequences in all available raw and assembled sequencing data across the tree of life.”

i. In the Discussion: “Processing the remaining 90%, as well as all future incoming data, can be achieved via scaling through parallelization and should be quite feasible for institutions such as EBI or NCBI” It was achieved already in Logan Search for SRA data until 2023.

We have rewritten this part of the discussion to more clearly reflect the relationship of Logan data and the existing MetaGraph indexes, acknowledging the contributions of the Logan project.

2. Cost projections: The projected per-query cost rose from USD 14 to ~USD 100 (Figure 5a). Please clarify the source of this revision and its implications for scalability.

We appreciate the reviewer’s concern regarding the change of our estimate. In the reviewer responses submitted alongside the previous version of our manuscript, we outlined that the increase in cost derives from

- a) the increase in the size of the SRA since our last reference point,*
- b) switching from theoretical throughput numbers to measured throughputs that take infrastructure limitations (network bandwidth, job scheduling overhead, instance boot time, etc.) of the cloud environment into account, and*
- c) the price changes for AWS instances since our initial submission.*

The size of the subset used to get to the estimates did not change the results in any substantial way.

3. Index construction strategy: The manuscript moved from monolithic per-clade indexes to > 1300 per-sample indexes for metagenomes (public folder s3://metagraph/all_sra/metagenomes/). In principle this should affect the linear extrapolations in Figure 3, given that lower compression is achieved when using subindexes instead of a single large one. Could the authors explain this?

Please note that we have added an additional way to index genomes: in addition to the “monolithic” indexes we also use chunk-based indexes (each corresponding to thousands of studies). The compression is slightly lower for metagenomes if split into chunks. However, the linear extrapolation is valid as it correctly uses the lower compression ratio for this case (see Table 1 last 4 rows).

4. Losslessness wording: The text still states the indexing approach is “lossless”; since noisy k-mers are dropped during cleaning this is inaccurate. Please rephrase.

In response to this concern, raised by multiple reviewers, we have included a dedicated "Limitations" section to the Discussion describing the context in which we feel that the term "lossless" applies, while clarifying that we are not presenting MetaGraph indexes as lossless representations of the raw read sets.

5. Throughout the manuscript, make the distinction between "SRA Sequence Data" and "Logan Contigs" explicit, as these are distinct datasets.

We thank the reviewer for highlighting this possible point of confusion. In the manuscript there are only a few places that make use of Logan contigs. Specifically, we use a random 4,700-study subset of Logan contigs as an additional base for extrapolating the full index size. Specifically, this result is shown in rows 21 and 22 of Table 1. The same 4,700-study subset is used for the query cost experiments presented in Figures 5a and S16a. While the reference to Logan was already present in the caption of Table 1 and in the Methods describing the 4,700-study subset, we have now also added a reference in the captions of Figures 5a and S16a.

Minor points:

1. The sentence "All indexes constructed from public-access data are available in the cloud under s3://metagraph" needs to instead point to the Github documentation repository explaining what indexes this s3 contains.

We thank the reviewer for pointing this out. We have extended the sentence to now also include a link to the online repository describing how to access the data.

2. "In total, we have processed almost 5 petabases [..]" in the Discussion. In the abstract, it is written that 16 petabases were processed.

We thank the reviewer for pointing out this discrepancy. We have updated the corresponding sections of the text. We indeed processed 5 petabases of raw input data directly from SRA. In addition, we also processed more than 16 million samples from the Logan contig set. We have removed any statements from the text that could lead to confusion.

Referee #3 (Remarks to the Author):

I thank the authors for their revised manuscript and attention to the reviewer comments. I have no further comments and feel that the manuscript is appropriate for publication.

We thank the reviewer for their positive assessment of our work.

Comments for Author

In the manuscript “Indexing All Life’s Known Biological Sequences” by Karasikov et al., the authors make a valiant effort to broaden the ****searchable**** set of biological sequences. The splashiest contribution of this article is the indexing of “all life’s known biological sequences” for search; they make petabases available for searching over and provide algorithms/webservices/software for doing so.

The second contribution are the algorithms for indexing and searching, which is in my mind technically much more challenging and also just as important of a contribution. This manuscript appears to be culmination of multiple prior efforts, some of which were previously published whereas others were only preprinted by not published.

- The MetaGraph algorithm was preprinted in 2020
- Alignment was improved upon: MetaGraph-MLA by Mustafa et al., 2022 bioRxiv and TCG Aligner by Karasikov et al, 2022 Genome Research)
- Better color compression (Karasikov et al., 2020, Journal of Computational Biology)

The engineering + algorithms are extremely impressive, and to me the main highlight of the paper, though because of the complicated prior history, it’s a little hard to discern which algorithmic contributions should be considered novel to this manuscript.

Overall, this work is a very strong methods paper that has the potential to substantially move the field forward. However, there are many non-trivial caveats, especially with respect to presentation, experiments, benchmarking (or lack thereof). In its current form, I don’t think this paper is suitable for Nature, but I could certainly see a revision making that cut.

Specific major points:

Point 1: The paper’s title is far too suggestive. While the author’s do not explicitly claim this work indexes ****all of life’s known sequences****, the title strongly suggests just that. A claim that strong needs to be substantiated. To pick just one example, metagenomes are vastly underindexed. Given that microbes make up a huge diversity of life and that most have not/cannot be cultured, this is an important issue. The authors index SRA-MetaGut (all human gut metagenomes) and MetaSub (4000+ urban environmental metagenomes) as the two sets of metagenomes fully indexed. This excludes non-gut, non-human, host-associated microbiomes as well as terrestrial microbiomes such as soil and ocean (although TARA *assemblies* are indexed).

On line 153, I note that the authors collected their SRA dataset in April 2020, which is four years old at this point. Another recent paper Pebblescout (Nature Methods 2024) indexes all SRA metagenomes (and all SRA RNA-Seq experiments) up to ~2022, arguably more a comprehensive database than what is presented in this paper. My sense is that Pebblescout’s indices are much less comprehensive and impressive than MetaGraph, but given the emphasis on exponential

growth in sequencing capacity in the introduction, a gap of 4 years to the present seems like sufficient time for the dataset sizes to have more than doubled.

Point 2: Many of the figures (3C, 4A-C) focus on biological discovery, but these show proof-of-concept results rather than significant biological discoveries that are substantiated with follow-up. For example, 3C suggests that one can use MetaGraph to do large-scale querying of transcript abundances, but no benchmarking of this functionality is shown. This would be OK for a methods-based paper (e.g. in Nature Methods), but seems a bit limited for a Nature paper (though I defer to the Editor on whether this is actually an issue).

Point 3: As a follow-up to point 2, given that the most significant contributions are methods-based, there is a lack of comprehensive benchmarking, especially of recent methods, many of which themselves benchmark against the MetaGraph preprint.

Indexing:

- Fig 2A – Some newer methods seem to be omitted from benchmarking, including kmindex (Lemane et al, 2024), Thjemisto (Alanko et al, 2023). Both methods claim better performance than MetaGraph in their papers. GGCAT is cited, but not compared against, either.
- Index building time is very relevant in practice. I do not see benchmarks for index building time, nor for continuous updating. The usefulness of a webserver like BLAST is partially in that it is kept up-to-date. How long would it take to rebuild (or update) the indexes on say a monthly basis? Would the index have to be built from scratch, or could it be delta'd?

Graph cleaning:

- Graph cleaning creates a **lossy** index, not a lossless index as claimed. This is an important point that needs to be cleaned up. The lossiness of MetaGraph is remarked upon by some of the competitor more recent papers, such as in the Pebblescout manuscript (Nature Methods 2024). Yes, it is technically true that MetaGraph provides a lossless representation of the **indexed** k-mer sets, but the data cleaning prior to choosing k-mers to index makes the overall process lossy.
- Furthermore, I would like to see benchmarks showing how graph cleaning contributes to the efficiency. Is the graph cleaning a major reason for MetaGraph's efficiency?
- Graph cleaning seems to be relatively aggressive (≤ 3 -coverage k-mers are pruned, but possibly more, according to Methods). This will remove a lot of low-abundance organisms' reads in Metagenomes, or lowly abundant transcripts. This point needs to be addressed. Fig. 3A does not adequately address this because it calculates the overall recall, but the recall of low abundance transcripts/organisms is interesting by itself. This is especially important for more careful interrogations of low-frequency bacterial species in a metagenome, rare isoforms in RNA-seq abundance quantification, or structural variations supported by a single high-quality long-read in say cancer biology.

Alignment/sequence search:

- More precision/sensitivity tests for alignment are likely needed. There is not test of precision that I am aware of. Figure 3A shows how the recall varies for reads mapping **to its own**

sample**, but there should probably be a test of how reads map to the combined metagraph index. Some notion of “false positive” should be used as a benchmarking metric. This is especially important since, unlike a BLAST search, k-mer matching does not give as obvious metrics for evaluating alignment uncertainty or goodness.

- On the same note, Extended Data Fig. 2 shows read mapping to an e-coli with MetaGraph’s alignment algorithms. E. coli genomes are very simple compared to a complex eukaryotic genome. Could this experiment be repeated with Human or Wheat genomes?
- Additionally, although it is impressive to have a single software that indexes close to “all” biological sequence, nucleotide sequence search and protein sequence search do actually often have substantial differences. The k-mer based exact match alignment for nucleotide sequences works less well for protein sequences, where identity between similar protein sequences may be lower. Could the authors provide a benchmark of this?

Specific minor points:

The abstract claims that the compressed representation fits on a single consumer hard drive (~100 USD), without specifying exactly which databases fit. Given the setup, the implication is that “all” sequences in this compressed format will fit on a \$100 drive. At present, the highest capacity HDD I can find for around \$100 only stores 12 TB of data. (<https://diskprices.com/>) However, in Table 1, the authors themselves say that SRA-Public to 01.01.2023 would take 130 TB to store the index. Even excluding that, the sum of all the other indices in the table seems to be just above 12 TB. Please clarify.

Although it is common folklore in the field that sequencing data has been growing exponentially (and it certainly was for a while), this appears to have leveled off in recent years. The authors cite a 2015 paper to substantiate their exponential claim. But, in the ENA statistics the authors cite, the run doubling time has been ****increasing**** over the last decade, from a doubling time of around 10-15 months back in 2012, to nearly 45 months currently. Were growth actually exponential, the run doubling time would be constant over time. Thus, although sequencing data still grows quickly, the exponential claim appears to be outdated.

Line 49: the authors have made a distinction between (i) sketching techniques and (ii) methods employing Bloom filters. Taxonomies are hard in any field, but certainly many computer scientists would include Bloom filters as a type of early probabilistic sketch in this day and age. The authors choice of taxonomy is justifiable, but certainly should be made explicit.

Line 137: saying that 4.8 petabases = 2.5 petabytes is a bit confusing without reading the supplement on how those numbers are counted. Not sure what if anything can be done here to clarify things, but it was a bit confusing on first reading.

Line 285: “with higher error rates, such as PacBio’s SMRT” PacBio HiFi reads have a lower error than short reads nowadays

Line 317: How could MetaGraph “facilitate large-scale learning tasks” and train LLMs?

Line 321: the authors have not demonstrated how sensitive MetaGraph sequence alignment can be before failing. I would like to know when it fails. Fig. 3A shows 2% mutation rate, but higher mutation rates should be tested. BLAST works with quite dissimilar query sequences, for example, and although MetaGraph should not be expected to be as sensitive, users should know when it fails.

Line 479: how is the gamma-distributed mean relevant

Line 780: what default value of seeds is used for alignment, k ?

Micro points:

Line 56: extra space before reference “alignment 34”.

Line 60: this sentence is a bit awkward. Consider rewording.

Line 411: “one from4”. Reference should be expanded here.

Line 1025: incorrect smart quotes. In latex need ` and’.

Reference 48 is a duplicate of reference 39.

Reference 61 does not properly capitalize “rna”

Table 1: is “MGS” defined anywhere?

Software comments

Overall, the software is extremely high-quality and well packaged for a bioinformatics software. There seems to have been dedicated software engineers for the software, which is extremely impressive. I had no problem running and installing the software. Documentation is well done. The web search is an impressive feat of engineering. It is a bit rough, but I think it has tremendous potential.

Web search suite

The web interface is decent looking, but quite simple. I think it could benefit from a few tune-ups. In general, I think more documentation on the website itself would be good.

- Alignment to graph option – this is not very interpretable for a general user. When should they use alignment vs normal k-mer querying?
- Performance is weird. Sometimes it's fast, sometimes it's quite slow and hangs.
- There should be an option to show only the best “X” number of hits. My browser ran out of memory.
- I think the authors could look at the Pebblescout (NCBI service) interface for inspiration on documentation, which I found a bit easier to use.

Error in the “Metazoa” index: (see attached doc)

TGAA

Select graph:
SRA-Metazoa (1,000 studies)

Minimum k-mer matches: 60%

Search with alignment

Search SRA-Metazoa (1,000 studies)

Search results

Show 10 entries

#	sample	k-mer matches
1	/cluster/work/grlab/projects/metagenome/data/cloudcompute/metazoa/clean/srr/srr110/srr11037758/srr11037758.fasta.gz	1431

Showing 1 to 1 of 1 entries 1 row selected

National Center for Biotechnology Information

Sequence Read Archive

Run Browser > /cluster/work/grlab/projects/metagenome/data/cloudcompute/metazoa/clean/srr/srr110/srr11037758/srr11037758

Run Browser

/cluster/work/grlab/projects/metagenome/data/cloudcompute/metazoa/clean/srr/srr110/srr11037758/srr11037758 is not found

Search and browse data for a single run

/cluster/work/grlab/project Search

What can be entered in this field?

The visualization of De Bruijn graphs seems to have an error:

Home Search Align Visualize Graphs

De Bruijn Graph Visualization Demo

4
ACTAGCTAGCTAGCTAGCTAGC

Build

De Bruijn Graph Visualization Demo

4
ACTAGCTAGCTAGCTAGC

Build

metagraph: /lib/x86_64-linux-gnu/libhts.so.3: no version information available (required by metagraph)

Software online documentation:

- Documentation is very good where present, but could be a bit more complete. Docs on sequence-to-graph alignment, which seems like an important aspect, is missing.
- The quick start page is a bit overwhelming. Perhaps an (interactive) flow chart would help.